# Amyloid-β disrupts APP-regulated protein aggregation and dissociation from recycling endosomal membranes

Preman J Singh [1,2,7], Bhavna Verma [1,7], Adam Wells[1], Cláudia C Mendes [1], Dali Dunn [1], Ying-Ni Chen [3], Jade Oh [1], Lewis Blincowe [1], S Mark Wainwright[1], Roman Fischer [4], Shih-Jung Fan [1,3], Adrian L Harris [5], Deborah C I Goberdhan [1,6,8] & Clive Wilson [1,8 ✉]

## Abstract

**Secretory proteins aggregate into non-soluble dense-core granules in recycling endosome-like compartments prior to regulated release. By contrast, aberrantly processed, secreted amyloid-β (Aβ) peptides derived from amyloid precursor protein (APP) form pathological extracellular amyloidogenic aggregations in late-stage Alzheimer's disease (AD). By examining living *Drosophila* prostate-like secondary cells, we show that both APP and Aβ peptides affect normal biogenesis of dense-core granules. These cells generate dense-core granules and secreted nanovesicles called Rab11-exosomes via evolutionarily conserved mechanisms within highly enlarged secretory compartments with recycling endosomal identity. The fly APP homologue, APP-like (APPL), associates with these vesicles and the compartmental limiting membrane, from where its extracellular domain modulates protein aggregation. Proteolytic release of this domain permits mini-aggregates to coalesce into a large central dense-core granule. Mutant Aβ expression disrupts this process and compartment motility, and increases aberrant lysosomal targeting, mirroring previously unexplained early-stage pathological events in AD. It also promotes cell-to-cell propagation of these endolysosomal defects, again phenocopying changes observed in AD. Our data therefore demonstrate physiological roles for APP in membrane-dependent protein aggregation, involving molecular mechanisms, which when disrupted by Aβ peptides, trigger Alzheimer's disease-relevant pathologies.**

**Keywords** Alzheimer's Disease; Dense-Core Granules; Exosomes; Rab11; Transforming Growth Factor-Beta-Induced
**Subject Categories** Membranes & Trafficking; Molecular Biology of Disease; Neuroscience

## Introduction

Alzheimer's Disease (AD) is a progressive neurodegenerative disorder, affecting an increasing proportion of the world's ageing population (Gustavsson et al, 2023). Post-mortem brains of AD patients typically present with extracellular amyloid plaques, primarily containing organised β-strand assemblies of Aβ-peptides generated from aberrant cleavage of the transmembrane amyloid precursor protein (APP) (Thal et al, 2002; Acquasaliente and De Filippis, 2022). Extracellular neuronal APP cleavage products are generated by at least three proteases, α-, β- and γ-secretase, with β- and γ-secretase required to produce amyloidogenic Aβ-peptides (O'Brien and Wong, 2011; Zhang et al, 2019).

Although late-stage amyloid plaque pathology can compromise brain function, plaque formation is probably not a key initiator of neurodegeneration in AD (Selkoe and Hardy, 2016). Therapeutic antibodies that disrupt amyloid plaques slow progression, but neither block nor reverse disease (Travis, 2023). One hypothesis is that early AD-associated defects in secretory pathways, which release cleaved forms of APP including Aβ-peptides, disrupt neuronal cell biology, altering processes such as endolysosomal trafficking, thus progressively leading to cell death (Kimura and Yanagisawa, 2018; Cataldo et al, 2000; Weglinski and Jeans, 2023). Indeed, some studies suggest that Aβ-peptides can induce *APP* loss-of-function phenotypes, to generate early pathological effects (Bignante et al, 2013; Kepp, 2016). APP is proposed to have multiple physiological roles, regulating brain development, memory and synaptic functions (Nalivaeva and Turner, 2013), displaying potential trophic activities (Dawkins and Small, 2014), as well as acting as a receptor for Wnts (Liu et al, 2021) and a ligand for heterotrimeric G-proteins (Copenhaver and Kögel, 2017). However, it remains unclear how Aβ-peptides might interfere with any of these functions.

Humans express three APP-like molecules, APP, APP-like protein 1 (APLP1) and APLP2. By contrast, the fruit fly, *Drosophila melanogaster*, has a single APP homologue, APP-like (APPL),

[1]Department of Physiology Anatomy and Genetics, University of Oxford, Oxford, UK. [2]US Department of Veterans Affairs, Veterans Affairs Medical Center, Fayetteville, AR, USA. [3]Department of Life Sciences, National Central University, Taoyuan City, Taiwan. [4]Target Discovery Institute, University of Oxford, Oxford, UK. [5]Department of Oncology, University of Oxford, Oxford, UK. [6]Nuffield Department of Women's and Reproductive Health, University of Oxford, Oxford, UK. [7]These authors contributed equally: Preman J Singh, Bhavna Verma. [8]These authors contributed equally: Deborah C I Goberdhan and Clive Wilson. ✉E-mail: clive.wilson@dpag.ox.ac.uk

facilitating loss-of-function studies. APPL contains α-, β- and γ-secretase cleavage sites, which also generate extracellular APP cleavage products, including Aβ-like peptides that can induce neurodegeneration (Carmine-Simmen et al, 2009; Wentzell and Kretzschmar, 2010). In addition, *Appl* loss-of-function behavioural phenotypes can be rescued by human APP, suggesting functional conservation (Luo et al, 1992). Although the specific molecular and subcellular functions of APPL remain unclear, it appears to control the balance of regulated secretion and retrograde transport at the synapse (Penserga et al, 2019).

Regulated secretion of proteins from neurons and glands typically involves secretory compartments in which proteins aggregate into non-soluble dense-core granules (DCGs), which dissipate upon extracellular release (Gondré-Lewis et al, 2012). For some hormones, like pituitary hormones, amyloid assembly drives DCG biogenesis (Maji et al, 2009), while for others, such as insulin in pancreatic β-cells, the hormone crystallises (Li, 2014) and amyloidogenesis is only observed in pathology (Nedumpully-Govindan and Ding, 2015). DCG compartment biogenesis within the *trans*-Golgi network (TGN) requires the monomeric G-protein Arf1 (Stamnes and Rothman, 1993; Traub et al, 1993), the adaptor protein complex, AP-1 (Bonnemaison et al, 2013) and members of the Rab family of monomeric G-proteins, which control compartment identity and vesicle trafficking. Rab6 coats compartments emerging from the TGN (Miserey-Lenkei et al, 2010), while mature DCG compartments often interact with the recycling endosomal marker Rab11 (Sugawara et al, 2009).

We have analysed DCG biogenesis in the prostate-like secondary cells (SCs) of the male accessory gland (AG) in *Drosophila melanogaster* (Fig. 1A). They form DCG compartments that are thousands of times larger in volume than typical secretory cells (Redhai et al, 2016), enabling dynamic intra-compartmental maturation events to be analysed ex vivo in real-time by fluorescence microscopy. Golgi-derived Rab6-labelled, precursor compartments must receive Rab11-mediated input and transition to recycling endosomal identity to trigger formation of DCGs and intraluminal vesicles (ILVs), which are secreted as exosomes (Fig. 1A) (Corrigan et al, 2014; Fan et al, 2020; Marie et al, 2023; Wells et al, 2023). Since these exosomes carry distinct cargos and are not generated in late endosomes, the previously identified origin of exosomes, they are termed Rab11-exosomes and have been shown to be generated by conserved subtype-specific mechanisms in humans and flies (Fan et al, 2020; Marie et al, 2023; Dixson et al, 2023).

Like mammalian DCGs, SC DCG biogenesis requires Arf1 and AP-1 (Wells et al, 2023; Gondré-Lewis et al, 2012). Recently, an Arf1-dependent switch to Rab11-positive recycling endosomal identity has been shown to take place as human secretory compartments mature, contrary to previously established models for regulated secretion (Stockhammer et al, 2024). The proteinaceous DCG in SCs is partially bordered by multiple small ILVs, which form chains that link it to the compartment's limiting membrane (Fig. 1A) (Fan et al, 2020). SC-specific knockdown of some *Endosomal Sorting Complexes Required for Transport* (*ESCRT*) genes, which regulate endosomal ILV formation, appears to disrupt DCG biogenesis, suggesting a function for specific ILV-generating mechanisms in this process (Marie et al, 2023). Interestingly, the glycolytic enzyme, glyceraldehyde 3-phosphate dehydrogenase (GAPDH), plays an evolutionarily conserved

'moonlighting' role in ILV and exosome clustering by tetramerizing on the outside of vesicles (Dar et al, 2021). This clustering activity is also required to assemble a single large DCG in SCs (Dar et al, 2021).

Here, we use genetic approaches to show that DCG protein aggregation is controlled by the homologues of two human proteins involved in amyloidogenic diseases. First, midline fasciclin (MFAS), the *Drosophila* orthologue of transforming growth factor-β-induced (TGFBI), a protein mutated in many forms of dominant amyloid corneal dystrophy (Han et al, 2016), is essential for all DCG protein aggregation. Second, APPL regulates the membrane- and ILV-dependent priming of MFAS aggregation, and when proteolytically cleaved, permits aggregated MFAS to coalesce and mature into a single large DCG. Furthermore, expressing pathological human Aβ-peptide mutants disrupts this maturation process, promoting lysosomal targeting of secretory compartments, but also inhibiting compartment motility, mirroring early trafficking defects observed in AD. Our data therefore suggest a link between APP, membranes, Rab11-exosomes and DCG biogenesis with relevance to AD pathology.

## Results

### *Drosophila* MFAS drives DCG assembly in SCs

We screened publicly available fly gene trap lines, in which green fluorescent protein (GFP) fusion proteins are expressed from endogenous gene loci, focusing on genes that are highly expressed in the AG (Appendix Table S1). Of these, *mfas* (Hu et al, 1998), which encodes the orthologue of the secreted fibrillar protein TGFBI (Nielsen et al, 2020; Corona and Blobe, 2021), was highly expressed specifically in SCs (Fig. 1B), consistent with recent single-cell RNAseq analysis (Immarigeon et al, 2021; Li et al, 2022). GFP-MFAS was concentrated in SC DCGs. Neither SC nor DCG morphology, which can be visualised by differential interference contrast microscopy (DIC), was affected by GFP-MFAS expression (Figs. 1B,C and EV1A), even when co-expressed with transgenes that permit adult SC-specific expression of other genes (Fig. EV1B–D) via the GAL4/UAS system under temperature-sensitive GAL80^ts-inducible control (Fan et al, 2020).

DCG biogenesis was followed in real-time using the *mfas* gene trap line. We observed two aggregation processes. First, multiple dispersed mini-cores initially assembled from diffuse, unevenly distributed clouds of GFP-MFAS that contacted the limiting membrane of immature compartments (Fig. 1E; Movie EV1). These mini-cores were mobile and fused with each other to make a large DCG within 30 min. Alternatively, about half of the DCGs formed within minutes by a single non-homogeneous aggregation event from a large diffuse GFP-MFAS cloud (Fig. EV1E; Movie EV2).

Knockdown of *mfas* with two independent RNAis specifically in adult SCs produced large compartments that lacked the DCG protein condensates normally detected by DIC (Figs. 1C,D,F and EV1A). In wild-type SCs, the limiting membrane of all DCG compartments is marked by Rab11 (Fan et al, 2020), visualised using a *YFP-Rab11* protein fusion expressed from the endogenous *Rab11* locus (Fig. 1D) (Dunst et al, 2015). *mfas* knockdown did not affect the number of Rab11-positive compartments when compared to control cells expressing an RNAi targeting the xanthine

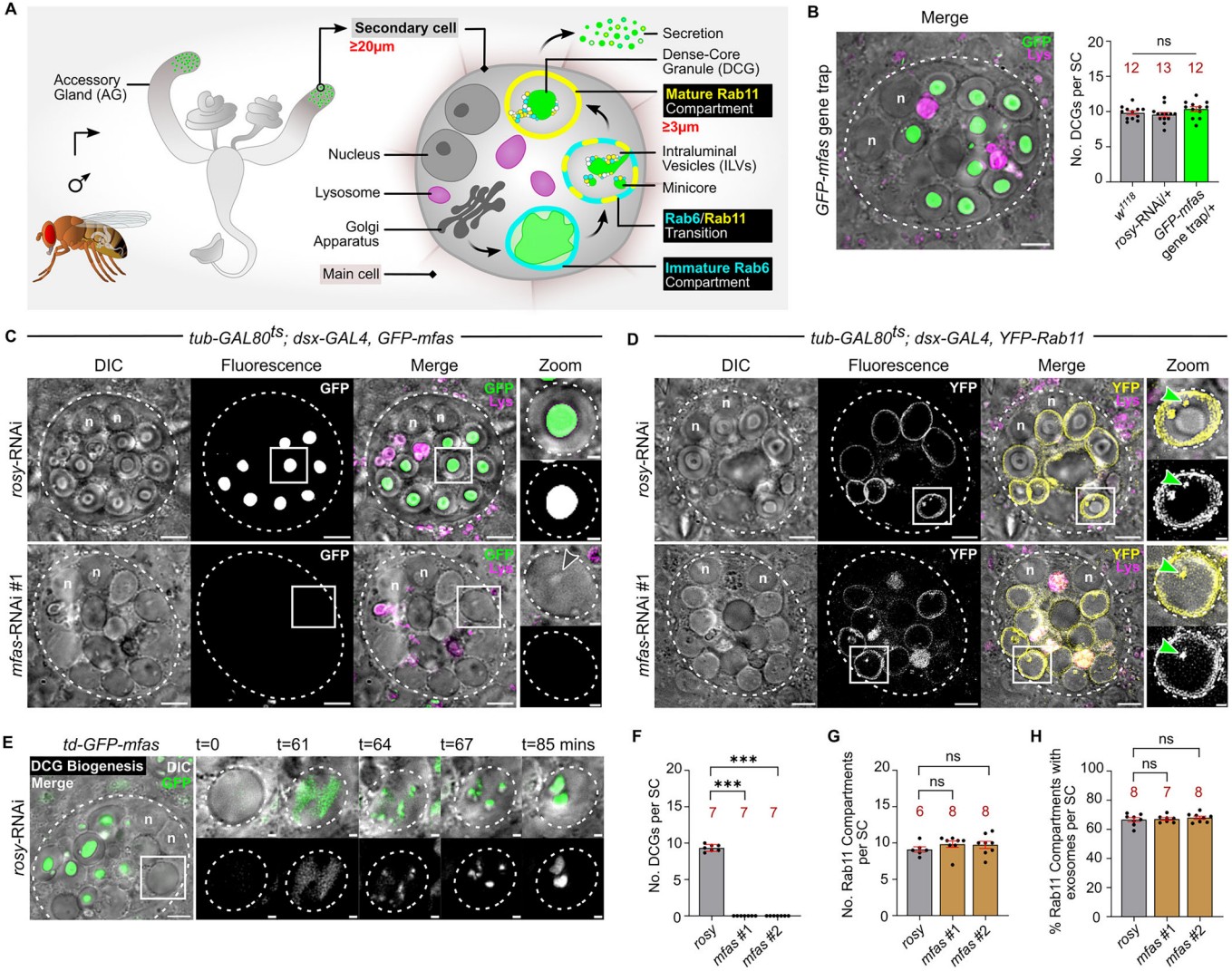

**Figure 1. *Drosophila* MFAS selectively drives DCG assembly in SCs.**

(A) Schematic showing male accessory gland, secondary cells (SCs) at the distal tip, and key SC compartments. (B) Ex vivo, wide-field fluorescence micrograph merged with differential interference contrast (DIC) image of SC from a 6-day-old male expressing *GFP-mfas* gene trap, showing fusion protein concentrated inside DCGs. *GFP-mfas* does not affect the number of DCGs per SC compared to controls. (C) SCs expressing *GFP-mfas* gene trap and SC-specific *rosy*-RNAi or *mfas*-RNAi. Following *mfas* knockdown, large compartments lack DCGs, which are normally readily identified by DIC (see Zoom panels for compartments outlined with white boxes), and only sporadic puncta are observed (grey arrowhead). (D) SCs expressing *YFP-Rab11* and SC-specific *rosy*-RNAi or *mfas*-RNAi. Most compartments contain Rab11-positive ILVs (green arrowhead in Zoom), which often co-localise with puncta detected by DIC. (E) Stills from a time-lapse movie of DCG biogenesis in SC expressing *GFP-mfas* gene trap, focusing on the compartment marked by the white box (DIC and GFP in top row; GFP in bottom row). *td-GFP-mfas* = *tub-GAL80^{ts}/+; dsx-GAL4, GFP-mfas/+*. (F–H) Bar charts showing DCG biogenesis (F), numbers of Rab11-positive large compartments (G) and the proportion of these compartments containing Rab11-positive ILVs (H) in *mfas* knockdown SCs (two independent RNAis) and control. In all images, approximate cell boundary and compartment boundaries are marked with a dashed white line; n nuclei of binucleate cells; LysoTracker Red (magenta) marks acidic compartments in (B–D). Scale bars = 5 and 1 μm in Zoom. For bar charts, data are mean ± SEM, analysed using the Kruskal–Wallis test; n number of animals (red, above bar), ***P < 0.001, ns not significant. In (F), P = 0.0005 for both comparisons. See also Fig. EV1; Movies EV1 and EV2. Source data are available online for this figure.

dehydrogenase gene, *rosy*, which does not affect DCG biogenesis (Figs. 1D,G and EV1B) (Marie et al, 2023). Furthermore, the number of compartments labelled with Rab6, which marks immature secretory compartments that lack DCGs and about half of the mature Rab11-positive DCG compartments (Wells et al, 2023), was also unaffected (Fig. EV1C,F).

We have previously shown that ILVs are present inside DCG compartments, using both EM and super-resolution analysis of fluorescently labelled transmembrane exosome markers, and that a

fraction of these ILVs is marked by Rab11, Rab6 or both (Fan et al, 2020; Wells et al, 2023). ILV formation is dependent on the Rab6 to Rab11 transition event required for DCG formation and maturation (Wells et al, 2023). The proportion of labelled compartments containing Rab11- and Rab6-positive puncta in their lumen was unaltered by *mfas* knockdown (Figs. 1D,H and EV1B,C,G). Suppressing *mfas* expression, therefore, does not detectably affect the Rab6 to Rab11 transition or exosome biogenesis during DCG compartment maturation.

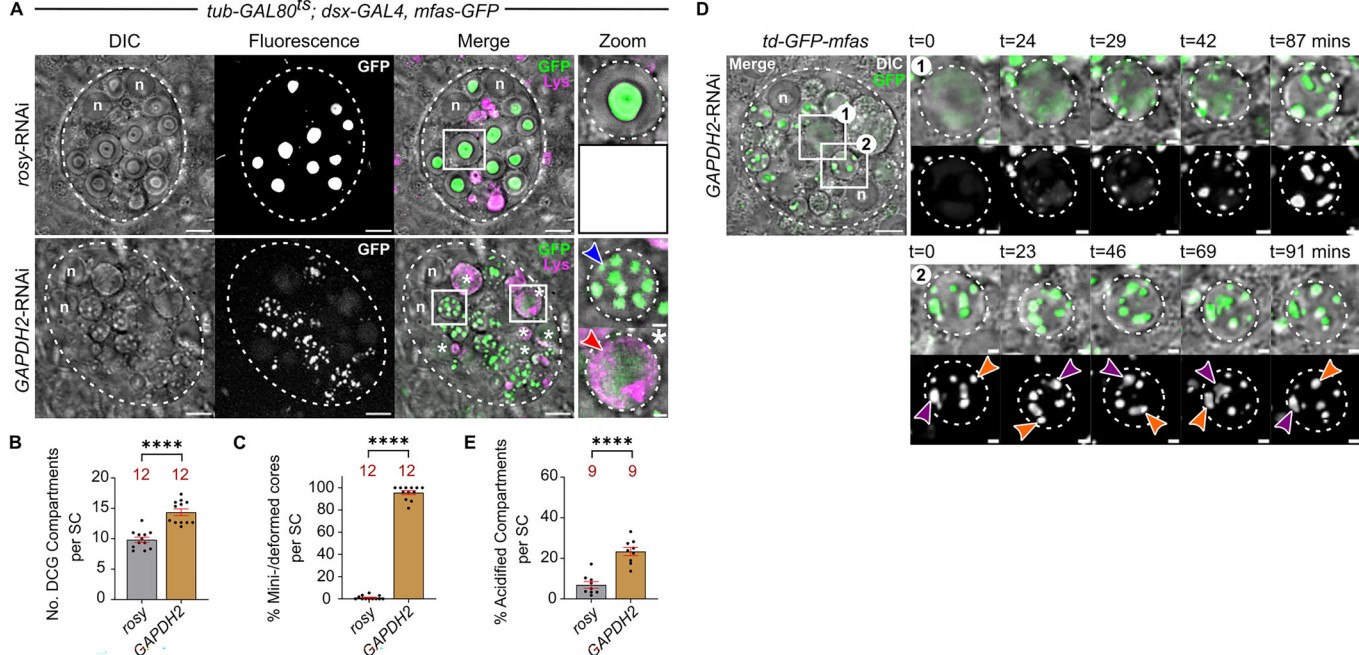

**Figure 2. GAPDH2 regulates mini-core fusion in SC DCG biogenesis.**

(A) SCs expressing *GFP-mfas* gene trap and SC-specific *rosy*-RNAi or *GAPDH2*-RNAi. Following *GAPDH2* knockdown, large secretory compartments contain multiple GFP-MFAS-positive mini-cores (blue arrowhead), visible by DIC (Zoom panels). White asterisks and white boxes marked with asterisks indicate compartments with DCG acidification phenotype (lower Zoom panel; red arrowhead marks acidic domain; not present in the control SC). (B, C) Bar charts plotting the number of DCG compartments (B) and proportion of compartments with multiple mini-cores (C). (D) Stills from a time-lapse movie of DCG biogenesis in SC expressing *GFP-mfas* gene trap and *GAPDH2*-RNAi, focusing on an immature DCG compartment (top rows [Box 1]) and a mature compartment (bottom rows [Box 2]). Note mini-cores remain motile within compartments that also rotate within the cell (two mini-cores marked by purple and orange arrowheads [Box 2]). (E) Bar chart plotting the proportion of large SC compartments with the DCG acidification phenotype. In all images, n nuclei; LysoTracker Red (magenta) marks acidic compartments in (A). Scale bars = 5 and 1 μm in Zoom. For bar charts, data are mean ± SEM, analysed using the Mann–Whitney test; *n* animal number above bar, ****P < 0.0001, ns not significant. See also Movie EV3 and Appendix Table S2. Source data are available online for this figure.

Although no central DCG was visible in the secretory compartments of *mfas* knockdown SCs using DIC, we checked whether proteins might still be able to condense in these compartments in a way that was not detectable by DIC, using another DCG marker, lipid-anchored GFP-GPI (Redhai et al, 2016). GFP-GPI normally labels the membranes of large Rab11-compartments and their ILVs, but much of it appears to be cleaved and incorporated into the DCG (Redhai et al, 2016) (Fig. EV1H). In *mfas* knockdown SCs, as in wild-type cells, the limiting membrane of the large secretory compartments was labelled; in addition, some small fluorescent internal structures, also visible by DIC, were observed, which we postulated were primarily membranous (Fig. EV1H). Indeed, closer examination of non-labelled *mfas* knockdown SCs and SCs expressing YFP-Rab11 or CFP-Rab6 revealed the presence of similar puncta in large non-acidic compartments, many of which contained these Rabs (e.g. marked puncta in Figs. 1C,D and EV1B), consistent with these puncta representing clustered ILVs. Therefore, although we cannot completely eliminate that there might be a minimal level of ILV-associated protein aggregation in the absence of *mfas*, this gene normally appears to play a highly specific and critical role in the rapid aggregation of proteins into mini-cores and large DCGs in the regulated secretory pathway of SCs.

## GAPDH is required for mini-core fusion in *Drosophila* SCs

The mini-core phase observed during DCG biogenesis (Fig. 1E) resembled the phenotype previously observed after knockdown of the *Drosophila* glyceraldehyde 3-phosphate dehydrogenase isoform, *GAPDH2*, in SCs, where compartments still undergo the Rab6 to Rab11 transition, but they contain multiple mini-cores, visible by DIC (Dar et al, 2021). This phenotype is attributed to a conserved role for GAPDH in humans and flies in clustering Rab11-exosomes and other secreted extracellular vesicles (EVs) (Dar et al, 2021). Following *GAPDH2* knockdown, the normal chains of clustered ILVs extending from compartment periphery to the central DCG (Fig. 1A) (Fan et al, 2020; Wells et al, 2023) are absent, and most of the resulting mini-cores are located near the compartment boundary and frequently surrounded by peripherally located ILVs (Dar et al, 2021).

Knockdown of *GAPDH2* in SCs expressing the *mfas* gene trap slightly increased the number of DCG compartments compared to controls, but consistent with our previous findings, almost all of these compartments contained multiple mini-cores, many in close proximity to the compartment's limiting membrane (Fig. 2A–C). Time-lapse imaging revealed that these mini-cores and the DCG compartments that contained them were mobile, similar to normal

cells. However, when the mini-cores collided, fusion usually did not take place (Fig. 2D; Movie EV3), perhaps because this process requires ILV adhesion.

*GAPDH2* knockdown SCs also contained several large compartments with diffuse or no GFP-MFAS and no central DCG visible with DIC microscopy, which were contacted by one or more peripheral crescent-shaped acidic structures, stained by the acid-sensitive dye LysoTracker Red (Fig. 2A). This 'DCG acidification phenotype', which might reflect a quality control step in DCG compartment biogenesis (Szenci et al, 2023), only appeared occasionally in control cells (Fig. 2E), and was generated by the fusion of DCG compartments with smaller lysosomal structures that rapidly dispersed the DCG (Fig. EV1G; Movie EV2). The phenotype was also seen in *mfas* knockdown cells (Fig. EV1I). We conclude that this 'DCG acidification phenotype' may occur more frequently in cells where DCG biogenesis is defective.

## GAPDH and other AD-associated glycolytic enzyme biomarkers co-isolate with human Rab1-exosomes

To confirm that GAPDH's association with Rab11-exosomes is conserved in human cells, we compared small EV (sEV) preparations from human HCT116 colorectal cancer cells under glutamine-depleted versus glutamine-replete conditions. Glutamine depletion reduces activity of the nutrient-sensing kinase complex, mechanistic target of rapamycin complex 1 (mTORC1), promoting a switch from release of primarily late endosomal exosomes, marked by CD63, to more Rab11-exosomes containing Rab11a (Fan et al, 2020). This treatment increased EV-associated GAPDH, as well as Rab11a, while, as expected, levels of CD63, a late endosomal exosome marker, were reduced (Fan et al, 2020) (Appendix Fig. S1).

In humans, both exosomes and GAPDH associate with β-amyloid plaques and may play a role in plaque assembly (Yuyama and Igarashi, 2017; Itakura et al, 2015; Lazarev et al, 2021). Levels of two other glycolytic enzymes, fructose-bisphosphate aldolase A (ALDOA) and pyruvate kinase (PKM), have recently been shown to be elevated in cerebrospinal fluid (CSF) of AD patients, forming part of a six-protein AD signature (Li et al, 2023). This study also highlighted 'extracellular exosome' as the most highly represented GO term for AD-enriched proteins. Interestingly, ALDOA and PKM were among the 48 proteins increased in our previous comparative proteomics analysis of Rab11a-exosome-enriched sEVs from HCT116 cells (Marie et al, 2023).

We tested whether these two glycolytic enzymes are elevated in Rab11a-exosome-enriched sEV preparations generated from another cell type, which undergoes a Rab11a-exosome switch in glutamine-depleted conditions, human HeLa cervical cancer cells (Fan et al, 2020). This cell line forms Rab11-positive recycling endosomes during secretory compartment maturation via an Arf1-dependent mechanism (Stockhammer et al, 2024), mirroring *Drosophila* SC biology (Wells et al, 2023). A comparative proteomics analysis of sEVs isolated by size-exclusion chromatography, either under glutamine-replete or glutamine-depleted conditions, identified 1156 proteins, of which 61 were significantly increased in Rab11a-exosome-enriched sEVs (Appendix Table S2). Only seven of these proteins were also enriched in HCT116 Rab11a-exosome-enriched preparations, including ALDOA and PKM (Appendix Table S2).

We conclude that specific glycolytic enzymes associated with the AD secretome are also elevated in human sEV preparations enriched in Rab11a-exosomes, including GAPDH, which is involved in Rab11-exosome-regulated protein aggregation in SCs. Given these observations, the previous links between GAPDH, exosomes and β-amyloid plaques (Dar et al, 2021; Yuyama and Igarashi, 2017; Itakura et al, 2015; Lazarev et al, 2021) and the proposed role of defective endosomal trafficking in initiating AD pathology (Cataldo et al, 2000), we tested the functions of transmembrane APP and its cleavage products (Fig. 3A) in SC DCG biogenesis and endosomal trafficking.

## *Drosophila* APPL regulates the formation of large DCGs in SCs

We knocked down the *Drosophila APP* homologue, *Appl*, in SCs using two independent RNAis. In both knockdowns, the total number of large DCG compartments remained unaltered, but most contained several mini-cores, and often an abnormally small and misshapen central DCG (Figs. 3B–D and EV2A). As with *GAPDH2* knockdown, most mini-cores were in close proximity to the compartmental limiting membrane. Furthermore, there was an increased proportion of large compartments with the DCG acidification phenotype (Figs. 3B,E and EV2A).

Analysis of *Appl* knockdown cells in YFP-Rab11 and CFP-Rab6 backgrounds revealed that the Rab identity of the defective mini-core compartments and the proportion of these compartments containing Rab-labelled ILVs, the precursors of Rab11-exosomes, remained unchanged compared to controls (Fig. EV2B–G). However, as with *GAPDH2* knockdown (Dar et al, 2021), and unlike controls, many Rab-labelled ILVs were clustered peripherally near the mini-cores (Fig. EV2B,C). The compartments with a DCG acidification phenotype were not marked by either Rab11 or Rab6 (Fig. EV2B,C), consistent with these compartments being targeted for lysosomal degradation.

We also analysed SCs from males hemizygous for an *Appl* null allele (Luo et al, 1992), *Appl^d* (Fig. EV2A,H–K). *Appl* null SCs contained similar numbers of DCG compartments to controls (Fig. EV2H). Like normal SCs, these compartments were all Rab11-positive and some were Rab6-positive (Appendix Fig. S2A,B). In contrast to *Appl* knockdown cells, mini-cores were uncommon (Fig. EV2A). However, DCGs in mutant cells were often misshapen and more frequently contacted the limiting membrane of the compartment (Fig. EV2A,I,K). Most notably, when compared to control and *Appl* knockdown cells, *Appl* null cells contained many more acidified DCG compartments (Fig. EV2A,J). Therefore, *Appl* is essential for the normal maturation of DCG compartments and appears to play an important role in the dissociation of DCG aggregates from the limiting membrane of these compartments.

Human APP partially rescues the behavioural defects caused by *Appl* loss-of-function in flies (Luo et al, 1992). It is proteolytically cleaved by secretases in the *Drosophila* nervous system, although processing at the β-secretase site is inefficient (Greeve et al, 2004; Ramaker et al, 2016). We found that overexpression of an APP-YFP fusion protein in SCs had no significant effect on DCG number, integrity and shape when compared to controls, as determined by DIC (Fig. 3A–E). Although YFP and GFP emission spectra overlap, GFP-MFAS is expressed at such high levels that APP-YFP was undetectable under the imaging conditions used

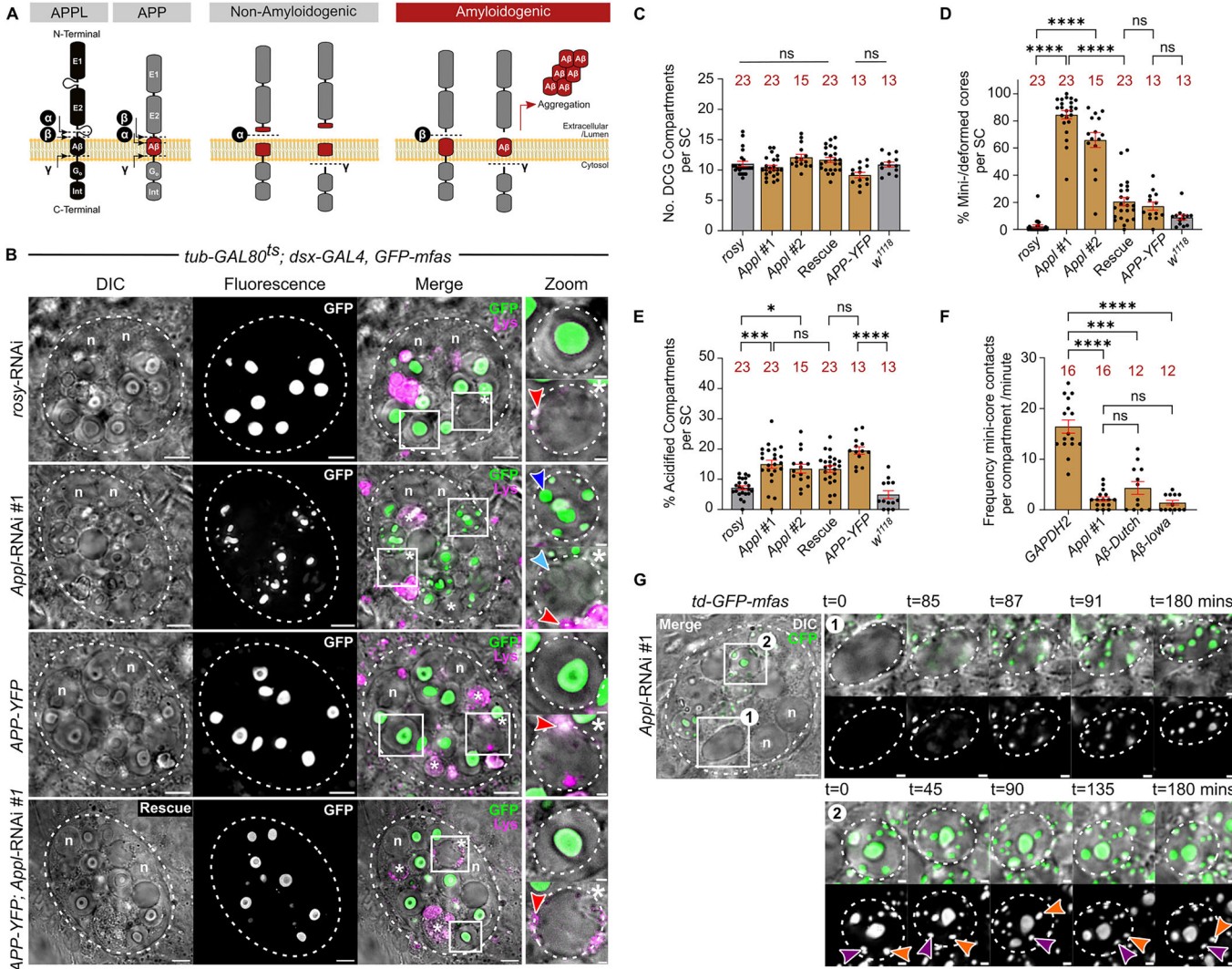

**Figure 3. *Drosophila* APPL regulates the formation of large DCGs in SCs.**

(A) Schematic showing structural similarities between human APP and *Drosophila* APPL proteins, the location of α-, β- and γ-secretase cleavage sites, and the APP cleavage products. (B) SCs expressing *GFP-mfas* gene trap and SC-specific *rosy*-RNAi, *Appl*-RNAi, APP-YFP, and *Appl*-RNAi and APP-YFP combined. Note multiple mini-cores (blue arrowhead) in *Appl* knockdown, also visible by DIC (white boxes and upper Zoom panels). White asterisks and white boxes marked with asterisks indicate DCG acidification phenotype (lower Zoom panel; red arrowheads mark acidic microdomains). Partially intact DCG marked with light blue arrowhead. (C–E) Bar charts showing effects of *Appl* knockdown and APP expression on the number of DCG compartments (C), mini-core formation (D) and DCG acidification phenotype (E). (F) Bar chart showing frequency of mini-core overlap over 30 min in time-lapse videos for single mini-cores selected from multiple different compartments, following SC-specific *GAPDH2* and *Appl* knockdown, and expression of human AD-associated Aβ-42-Dutch and Aβ-42 Iowa mutants. (G) Stills from a time-lapse movie of DCG biogenesis in SC from a 6-day-old male expressing *GFP-mfas* gene trap and *Appl*-RNAi, focusing on one compartment forming mini-cores (Box 1) and another mature compartment (Box 2). Note compartments and mini-cores are relatively immobile (two mini-cores marked by purple and orange arrowheads [Box 2]). In all images, n nuclei; LysoTracker Red (magenta) marks acidic compartments in (B). Scale bars = 5 and 1 μm in Zoom. For bar charts, data are mean ± SEM analysed using Kruskal–Wallis test; *n* = animal number above bar, *P < 0.05, ***P < 0.001, ****P < 0.0001, ns not significant. In (E), *rosy* vs *Appl* #1, P = 0.0003; *rosy* vs *Appl* #2, P = 0.038. In (F), *GAPDH2* vs Aβ-Dutch, P = 0.0008. See also Fig. EV2; Appendix Figs. S2 and S3A; Movie EV4. Source data are available online for this figure.

(Appendix Fig. S3A), allowing us to conclude that GFP-MFAS condensation in DCGs was also normal in these cells. However, APP-YFP did induce the DCG acidification phenotype (Fig. 3E), perhaps because overexpressed APP is not properly processed at the DCG compartment limiting membrane in these cells, and this promotes lysosomal targeting. Importantly, overexpressing other membrane-associated exosome markers or secreted proteins in SCs normally does not induce this phenotype (Fan et al, 2020) (Fig. EV1H).

Overexpressing APP-YFP in *Appl* knockdown cells almost completely rescued the mini-core phenotype, although about 10% of compartments displayed a novel phenotype in which the DCG's centre lacked fluorescent GFP-MFAS (Figs. 3B,D and EV2L), suggesting that a small proportion of DCGs did not assemble entirely normally. APP-YFP expression also could not suppress the DCG compartment acidification associated with *Appl* knockdown, presumably because it induces this phenotype itself (Fig. 3B,E). Overall, these findings suggest that human APP and fly APPL share

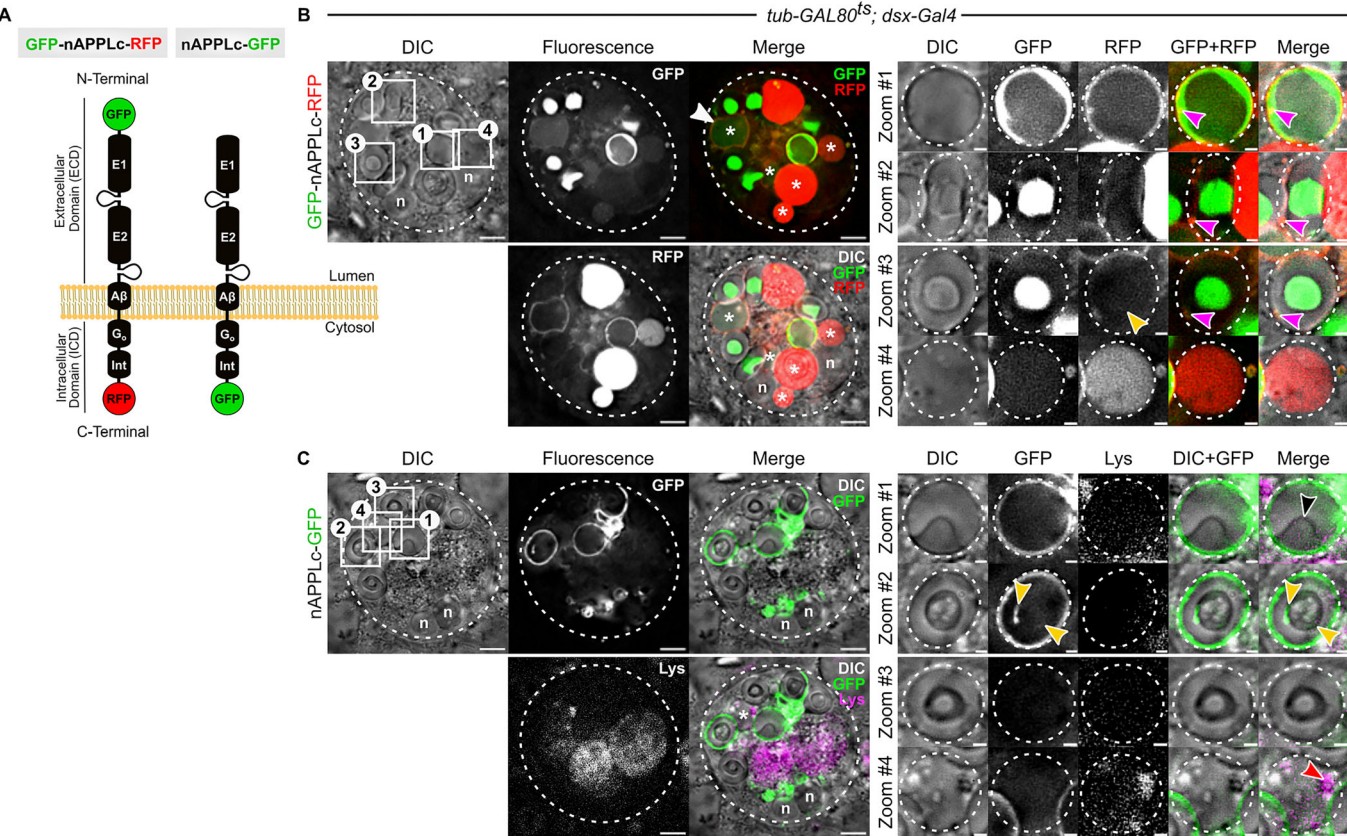

**Figure 4. Cleavage of *Drosophila* APPL accompanies normal DCG formation.**

(**A**) Schematic showing two fluorescently tagged APPL constructs employed. (**B**, **C**) SCs from 6-day-old males expressing either double-tagged APPL (dt-APPL; **B**) or APPL-EGFP (**C**). For both cells, magnified images in Zoom highlight a DCG precursor compartment (Zoom #1); a mature DCG compartment (in (**B**), DCG contains APPL's ECD (green) and limiting membrane carries APPL's ICD (red)) (Zoom #2); a more mature DCG compartment with no peripheral ICD (Zoom #3) and a compartment that appears to have the DCG acidification phenotype (Zoom #4). White asterisk marks other compartments with the DCG acidification phenotype. ILVs inside (yellow arrowheads) and at the periphery (magenta arrowheads) of DCG compartments are marked. In (**B**), the white arrowhead marks a very early acidified DCG compartment, and in (**C**), the black arrowhead (Zoom #4, Merge) marks a DCG that is starting to form inside the DCG precursor compartment, and the red arrowhead marks an acidic microdomain. In all images, n nuclei; LysoTracker Red (magenta) marks acidic compartments in (**C**). Scale bars = 5 and 1 μm in Zoom. See also Appendix Fig. S3B; Movie EV4. Source data are available online for this figure.

the functional activity required for normal DCG protein aggregation and subsequent separation from the DCG compartment's limiting membrane.

We investigated whether the mini-core phenotype caused by *Appl* knockdown was linked to the failure of motile mini-cores to collide and fuse, as we had observed for *GAPDH2* knockdown. Time-lapse movies of GFP-MFAS in *Appl* knockdown SCs revealed that, unlike *GAPDH2* knockdown, the multiple mini-cores formed by GFP-MFAS aggregation in each compartment remained peripheral and rarely moved relative to each other. Furthermore, the defective compartments were more static within the cytoplasm (Fig. 3F,G; Movie EV4), suggesting that these compartments are more stably attached to the surrounding cytoskeleton.

## Cleavage of *Drosophila* APPL accompanies normal DCG formation

Based on these findings, we hypothesised that transmembrane APPL interacts with aggregating proteins in DCG compartments

and is then proteolytically cleaved as part of the DCG maturation process. To investigate this further, an APPL construct tagged at both its extracellular N-terminal and intracellular C-terminal ends was overexpressed under inducible *dsx-GAL4/UAS* control in SCs (dt-APPL; Fig. 4A) (Ramaker et al, 2016). Although the compartmental organisation of SCs was disrupted following six days of expression, so that lysosomes were expanded, compartments representing different stages of DCG biogenesis could still be identified. Cells contained one or two large, relatively central compartments that did not have a central DCG and lacked the granular morphology of lysosomes observed with DIC, a phenotype characteristic of DCG precursor compartments (Wells et al, 2023). In these compartments, the enhanced GFP- (EGFP-) labelled extracellular domain (ECD) was located on and just inside the compartment's limiting membrane, which was marked by the monomeric RFP- (mRFP-) labelled intracellular domain (ICD; Fig. 4B, Zoom #1), suggesting that some of the APPL protein remained uncleaved. The GFP-labelled ECD inside the compartment, but at the periphery, appeared to co-localise with aggregated

material detected by DIC, suggesting that the protein aggregation process, which is normally Rab11-dependent, had been activated. In mature DCG compartments, the ECD was concentrated in the DCG, and frequently low levels of the ICD, which does not detach from APPL's transmembrane domain following cleavage by α- or β-secretase, were associated with parts or all of the limiting membrane (Fig. 4B, Zoom #2 and #3). Furthermore, sporadic ICD/ECD co-labelling was observed in small intra-compartmental vesicles and at the DCG periphery, where ILVs are known to cluster (Fan et al, 2020). Since the most mature DCG compartments lack the GFP-labelled ECD at their limiting membrane, we assessed their Rab11-identity using YFP-Rab11, and found that like normal cells, these compartments were Rab11-positive and therefore appeared to have matured normally (Appendix Fig. S3B).

In some cells, a compartment labelled peripherally by the ICD contained diffuse GFP inside it, consistent with early stages of DCG compartment acidification (white arrowhead; Fig. 4B). There were also compartments with high levels of internalised mRFP-labelled ICD, which by DIC, had the typical 'granular' morphology of lysosomes (Fig. 4B, Zoom #4).

In SCs, RFP- and mCherry-labelled proteins are frequently preferentially targeted to lysosomes compared to their GFP-equivalents (Fan et al, 2020), so we also assessed the localisation of an APPL-EGFP C-terminal fusion in SCs (Fig. 4A) (Penserga et al, 2019). Like the mRFP-tagged ICD, the EGFP-labelled ICD was observed at the limiting membrane of all DCG precursor compartments and some DCG compartments (Fig. 4C, Zoom #1 and #2), though other, potentially more mature, DCG compartments did not carry detectable levels of this marker (Fig. 4C, Zoom #3). EGFP-labelled APPL ICD was also more frequently and clearly detected in ILVs, many at the periphery of DCGs (Fig. 4C, Zoom #2). Several compartments displayed the DCG acidification phenotype, though EGFP fluorescence was not detectable, presumably because it is quenched in acidic compartments (Fig. 4C, Zoom #4).

These findings are consistent with a model in which full-length APPL traffics into Rab6-labelled DCG precursor compartments, most likely through the Rab11-positive recycling endosomal pathway. However, we cannot exclude the possibility that some arrives through the Golgi-Rab6 pathway when overexpressed. It is then cleaved as these compartments start to generate mini-cores and DCGs. The APPL ECD produced, and any additional ECD subsequently trafficked into DCG compartments, becomes associated with DCG protein aggregates, which separate from the limiting membrane. Some APPL ICD remains on ILV and DCG compartment membranes, but at least when APPL is overexpressed, most of the ICD is subsequently trafficked to lysosomes.

### *Drosophila* APPL and APPL cleavage regulate DCG protein aggregation and dissociation of these aggregates from membranes

To investigate DCG biogenesis further, we knocked down transcripts encoding the fly α- (Kuzbanian [Kuz]), β- (Beta-site APP-cleaving enzyme [Bace]) and γ- (Nicastrin [Nct]) secretases, key enzymes that cleave APP, separating the ECD from the ICD (Figs. 5A and EV3A–E). None of the knockdowns altered DCG compartment number (Fig. EV3B). However, DCG morphology

was affected by knockdown of *β-secretase*, with many DCGs lacking GFP-MFAS in their centre (Fig. EV3A,C,E). Knockdown of each secretase increased the number of acidified DCG compartments (Fig. EV3D), presumably due to the DCG compartment maturation process being disrupted. Overall, this suggests that the appropriate balance of α-, β- and γ-secretase activities is critical for DCG compartment maturation in SCs. However, because these proteases have targets other than APPL, aberrant APPL cleavage is not the only possible explanation for these phenotypes.

To assess the role of APPL cleavage, we overexpressed a form of APPL that lacks the α- and β-cleavage sites (APPL-Δsd; Fig. 5A,B) (Torroja et al, 1999). As observed for human APP-YFP and fluorescently tagged forms of APPL (Figs. 3 and 4), when compared to controls, overexpression of a wild type form of APPL (APPL-WT) produced some deformed DCGs and induced the DCG acidification phenotype (Figs. 5B–E and EV4A). DCG compartment number was slightly reduced, perhaps because of this lysosomal targeting (Fig. 5C). However, all DCG compartments without the acidification phenotype were YFP-Rab11-positive, and about half were Rab6-labelled, suggesting they had matured via the normal Rab6 to Rab11 transition mechanism (Wells et al, 2023) (Appendix Fig. S4).

Overexpression of the non-cleavable APPL-Δsd mutant reduced DCG compartment number further, generated many more acidified DCG compartments and expanded SC lysosomal area, when compared to APPL-WT overexpression (Figs. 5 and EV4A). In fact, it was possible to deduce that only about 85% of SCs retained their basic secretory morphology within the AG of these males after six days, since defective, dying or dead GFP-MFAS-labelled SCs that lacked normal secretory compartments, as determined by DIC, persisted and could be scored among the ~40 SCs in each AG lobe (Appendix Fig. S5). Therefore, failure to cleave APPL increases the targeting of DCG compartments for lysosomal degradation and reduces SC viability. Nevertheless, although DCG formation and compartment maturation were affected by APPL-Δsd expression, a large central core often still formed, suggesting that DCG aggregates can dissociate from non-cleaved APPL.

In human APP, the ECD's E1 domain (Fig. 5A) regulates APP dimerisation, a process modulated by the divalent cations, $Zn^{2+}$ and $Cu^{2+}$ (August et al, 2019), both of which can affect DCG biogenesis (Germanos et al, 2021; Jayawardena et al, 2019). We overexpressed non-cleavable APPL that lacked much of the E1 domain (APPL-ΔsdE1; Fig. 5A) (Torroja et al, 1999) to test the latter's function. Only about 40% of SCs retained large secretory compartments following 6 days of expression of this construct (Appendix Fig. S5). In these cells, most DCG compartments were defective and the majority of the non-acidified compartments contained a peripheral network of aggregated GFP-MFAS (Figs. 5 and EV4A,B). Therefore, APPL's E1 domain appears to control the association between the APPL ECD and protein aggregates during DCG compartment maturation, so that in its absence, more stable membrane-associated aggregates form. Although the proportion of acidified DCG compartments observed in SCs was unaffected when compared to APPL-WT-expressing cells, the lysosomal area was significantly increased (Fig. 5B,E,F), suggesting DCG compartments are preferentially targeted for degradation when APPL-ΔsdE1 is overexpressed.

To confirm that the Rab6 to Rab11 transition associated with DCG compartment maturation was still occurring in cells

expressing non-cleavable forms of APPL, we again assessed the Rab6 and Rab11-identity of these compartments, and found that non-acidified mature aggregate-containing compartments were all Rab11-positive, and about half were also Rab6-positive (Appendix

Fig. S4). With APPL-ΔsdE1 overexpression, Rab11 and Rab6 formed an intra-compartmental peripheral network, suggesting that ILVs are associated with the peripheral protein aggregates induced in this genotype.

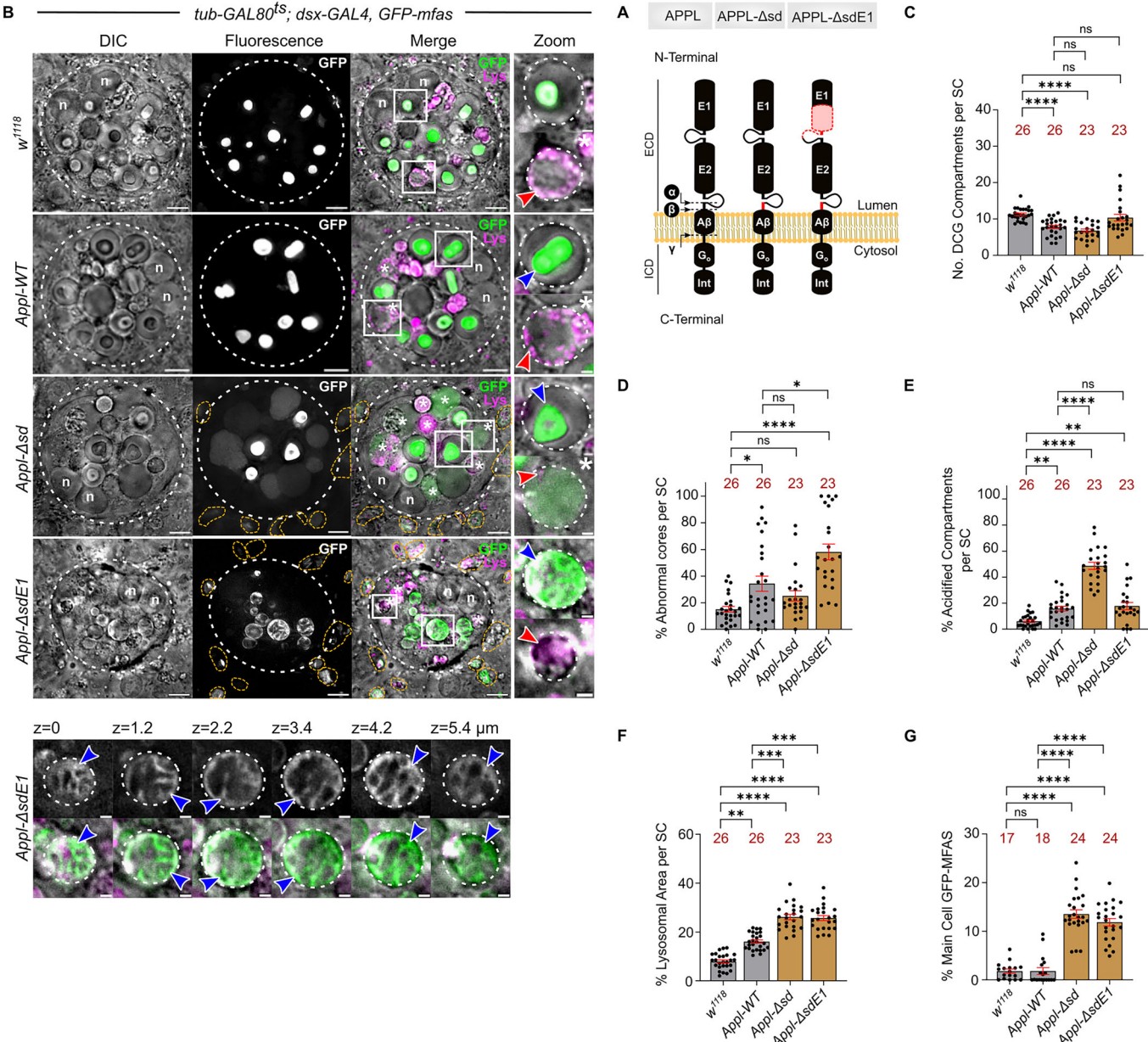

**Figure 5. *Drosophila* APPL regulates DCG protein aggregation, and its cleavage is required for normal DCG formation.**

(A) Schematic showing APPL constructs lacking α- and β-secretase cleavage sites (deletion in red near transmembrane domain). APPL-ΔsdE1 construct lacks a large part of the E1 domain. (B) SCs expressing *GFP-mfas* gene trap and either no other transgene, or SC-specific wild-type APPL (APPL-WT), APPL-Δsd or APPL-ΔsdE1. Note abnormal DCG phenotypes compared to *w[1118]* control (compartments outlined with white boxes, upper Zoom; blue arrowheads mark abnormal DCGs). For APPL-ΔsdE1, the lower panels show individual z-plane images of the highlighted compartment to illustrate the peripheral network with limited centrally located GFP-MFAS aggregation. White asterisks and white boxes marked with asterisks indicate DCG acidification phenotype (lower Zoom; red arrowheads mark acidic microdomains). Enlarged acidic main cell compartments containing GFP are outlined (orange dashed lines). (C–F) Bar charts showing DCG compartment number (C), proportion of abnormal DCGs (D), proportion of large compartments with DCG acidification phenotype (E), SC lysosomal area (F), and accumulation of GFP-MFAS in main cells expressed as percentage of total main cell area that contains GFP ((G); see also Fig. EV4). In all images, n nuclei; LysoTracker Red (magenta) marks acidic compartments. Scale bars = 5 and 1 μm in Zoom. For bar charts, data are mean ± SEM, analysed using the Kruskal–Wallis test; n = animal number above bar, *P < 0.05, **P < 0.01, ***P < 0.001, ****P < 0.0001, ns not significant. In (D), *w[1118]* vs *Appl-WT*, P = 0.034; *Appl-WT* vs *Appl-ΔsdE1*, P = 0.015. In (E), *w[1118]* vs *Appl-WT*, P = 0.009; *w[1118]* vs *Appl-ΔsdE1*, P = 0.009. In (F), *w[1118]* vs *Appl-WT*, P = 0.0056; *Appl-WT* vs *Appl-Δsd*, P = 0.0001; *Appl-WT* vs *Appl-ΔsdE1*, P = 0.0003. See also Figs. EV3 and EV4; Appendix Figs. S4 and S5. Source data are available online for this figure.

We investigated whether overexpression of APPL constructs in general might induce major defects in SC compartmental organisation over time, by assessing the presence of morphologically abnormal SCs at 3, 6 and 12 days after eclosion (Appendix Fig. S5). For APPL-WT-expressing SCs, almost all cells retained their secretory morphology throughout the time course, as also observed in control cells. However, overexpression of non-cleavable APPL induced severe morphological defects in essentially all SCs after 12 days of expression. For the APPL-ΔsdE1 mutant, the majority of SCs displayed this phenotype even after 3 days. Nevertheless, some cells, which presumably expressed the lowest levels of APPL-ΔsdE1 throughout adulthood, displayed the characteristic aggregate network phenotype even in 12-day-old males (Appendix Fig. S5). We conclude that while overexpressing APPL-WT in SCs has little, if any, effect on the basic secretory morphology of SCs, despite inducing compartment acidification, non-cleavable forms of APPL reduce SC viability over time, with the formation of defective DCGs appearing to precede such changes.

Overexpression of the two non-cleavable APPL proteins in SCs also produced an additional phenotype. The other, much more numerous, epithelial cell type in the accessory gland, main cells (Fig. 1A), contained high levels of internalised GFP, which appeared to be located within unusually enlarged acidic compartments (Figs. 5B,G and EV4B,C). This was not simply explained by the uptake of more cell debris produced by 'unhealthy' SCs, because the phenotype was observed with APPL-Δsd, which has modest effects on cell morphology at 6 days, and it was not restricted to the main cells located adjacent to morphologically abnormal SCs. GFP-MFAS, therefore, appears to continue to be secreted in the presence of APPL-Δsd proteins, but is either preferentially endocytosed from the accessory gland lumen by non-expressing cells and/or is not degraded normally when internalised by these cells.

Taken together, our data suggest that the absence or overexpression of APPL, particularly non-cleavable mutant forms, leads to defects in DCG biogenesis. In all of these cases, more compartments become targeted for lysosomal degradation, but may not traffic efficiently to lysosomes. When APPL is not cleaved and membranes remain associated with protein aggregates, this can also affect the uptake and/or breakdown of secreted MFAS by other cells.

## Amyloidogenic Aβ-42 peptides disrupt DCG biogenesis in SCs

We hypothesised that APPL's functions in DCG compartment maturation and trafficking might be selectively misregulated in Alzheimer's Disease. To test this, we overexpressed in SCs a wild-type form of the pathological Aβ-42-peptide generated by β- and γ-secretase cleavage of APP (O'Brien and Wong, 2011), and two mutant peptides with elevated amyloidogenic activity associated with familial AD, the Dutch (Van Broeckhoven et al, 1990) and Iowa (Grabowski et al, 2001) mutants (Van Nostrand et al, 2002) (Fig. 6A,B). These were produced as fusions with an N-terminal ER signal sequence to direct them to the secretory system. In *Drosophila*, expression of these molecules in either the CNS or the eye induces neurodegenerative phenotypes in adults, with wild-type Aβ-42 producing the mildest phenotypes (Chouhan et al, 2016; Metsla et al, 2022).

Overexpression of these Aβ-42 molecules in SCs disrupted aspects of DCG formation (Fig. 6A,C–E). While wild type Aβ-42 did not affect the number of DCG compartments and intact DCGs per cell, the number of DCG compartments increased in SCs expressing the Dutch and Iowa Aβ-42 mutants and more than 30% of these contained mini-cores, often in addition to a central abnormally shaped small DCG (Fig. 6A,C,D). An increased number of acidified DCG compartments was also observed, even with wild-type Aβ-42 overexpression (Fig. 6A,E). Time-lapse movies of DCG compartment maturation in SCs expressing either the Dutch or Iowa Aβ-42 mutant peptides revealed a very similar phenotype to *Appl* knockdown with static peripheral mini-cores and compartments (Fig. 6F,G; Movies EV5 and EV6). In both cases, new mini-cores could still condense in immature DCG compartments (top panels of Fig. 6F,G), but they lacked motility. Secretory compartments in SCs expressing different Aβ-42-peptides still retained the expected Rab11 and Rab6 identity (Appendix Fig. S6), suggesting a selective defect in DCG aggregation and compartment maturation that was unlikely to result from general defects in intracellular trafficking.

Since Aβ-42 oligomers can affect APP processing (Puzzo et al, 2017; Wang et al, 2017), we tested whether the cleavage of double-tagged dt-APPL might be affected by expression of wild type and mutant Aβ-42. Unlike controls, mature DCG compartments in Aβ-42-expressing backgrounds exhibited EGFP/mRFP co-labelling at their limiting membrane, suggesting that some APPL remained uncleaved (Appendix Fig. S7). Peripheral, crescent-shaped, EGFP-ECD-labelled protein aggregates were also observed more frequently in DCG compartments, when Aβ-42-Dutch was expressed. This is consistent with compartments failing to mature normally and retaining limiting membrane-associated aggregates, albeit not in the mini-core conformation observed in the absence of overexpressed APPL. Mutant Aβ-42 expression induced the formation of co-labelled puncta inside these compartments, which presumably represent one or probably clustered ILVs carrying uncleaved APPL (Appendix Fig. S7). Aβ-42-peptide expression, therefore, appears to either indirectly or directly disrupt APPL processing.

We also checked whether prolonged Aβ-42 expression might induce major morphological defects in SCs by analysing phenotypes over a 12-day time course (Appendix Fig. S8). Expression of wild type and mutant Aβ-42 only induced SC DCG compartment phenotypes after 6 days. At this stage, most SCs retained normal cellular morphology. It was only after 12 days that most SCs expressing the two mutant peptides displayed aberrant morphology (Appendix Fig. S8), consistent with it being a consequence of earlier, more specific defects in these cells.

To determine whether expression of Aβ-42 in SCs affected secretion of GFP-MFAS or its extracellular aggregation, we assessed the accumulation of GFP in the accessory gland lumen over six days post-eclosion. In wild-type 6-day-old males, relatively homogeneously distributed GFP-MFAS was observed (Appendix Fig. S9A), consistent with our previous observations that DCGs are rapidly dispersed following secretion (Miserey-Lenkei et al, 2010). Wild type and mutant Aβ-42 expression did not induce obvious extracellular mis-aggregation, though the total levels of secreted GFP-MFAS were increased (Appendix Fig. S9A,B; for Aβ-42-Dutch, this apparent increase did not reach significance). Furthermore, endocytosed GFP-MFAS abnormally accumulated in acidic compartments of main cells in these glands (Fig. EV5),

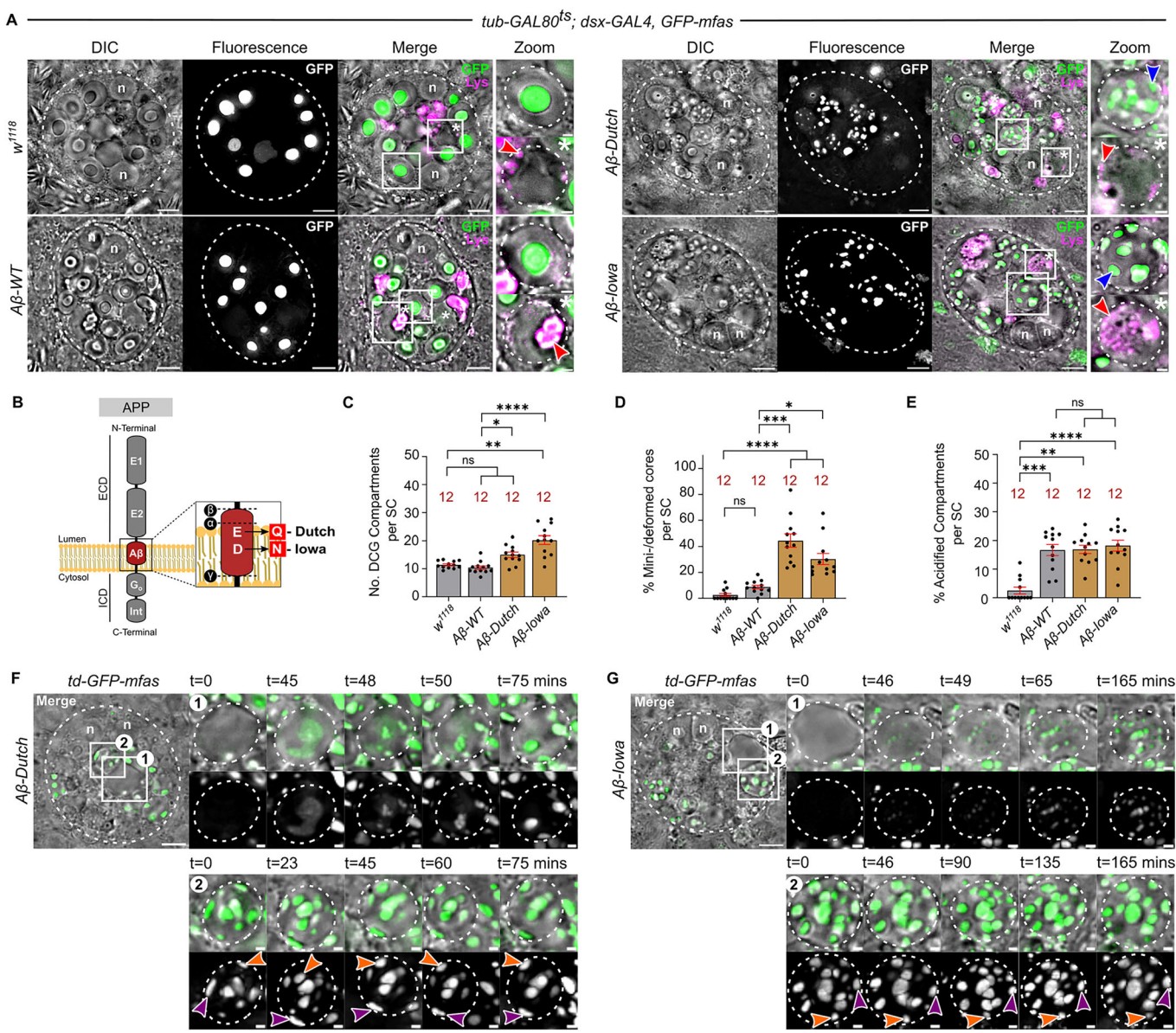

**Figure 6. Expression of pathological mutant Aβ-42 peptides in SCs disrupts DCG biogenesis and increases lysosomal targeting.**

(A) SCs expressing *GFP-mfas* gene trap and either no other transgene, or wild type Aβ-42-peptide, or either the Iowa or Dutch mutant Aβ-42 peptides. White asterisks and white boxes marked with asterisks indicate DCG compartments with acidification phenotype (lower Zoom panel; red arrowheads mark acidic microdomains). Blue arrowheads mark mini-cores. (B) Schematic showing different Aβ-42 peptides tested. (C–E) Bar charts showing DCG compartment number (C), proportion of compartments with a mini-core phenotype (D), and proportion of large compartments with DCG acidification phenotype (E). (F, G) Stills from time-lapse movies of DCG biogenesis in SCs expressing *GFP-mfas* gene trap with Dutch (F) or Iowa (G) Aβ-42 mutant, focusing on one compartment as it forms mini-cores (Merge, Box 1) and another mature compartment (Box 2). Note compartments and mini-cores are relatively immobile (two mini-cores marked by purple and orange arrowheads [Box 2]). In all images, n nuclei; LysoTracker Red (magenta) marks acidic compartments in (A). Scale bars = 5 and 1 µm in Zoom. For bar charts, data are mean ± SEM, analysed using the Kruskal–Wallis test; *n* = animal number above bar, *P < 0.05, ***P < 0.001, ****P < 0.0001, ns not significant. In (C), $w^{1118}$ vs Aβ-Iowa, P = 0.0013; Aβ-WT vs Aβ-Dutch, P = 0.012. In (D), Aβ-WT vs Aβ-Dutch, P = 0.0002; Aβ-WT vs Aβ-Iowa, P = 0.018. In (E), $w^{1118}$ vs Aβ-WT, P = 0.0008; $w^{1118}$ vs Aβ-Dutch, P = 0.0011. See also Fig. EV5; Appendix Figs. S6–S9; Movies EV5 and EV6. Source data are available online for this figure.

though at lower levels than when non-cleavable forms of APPL were expressed in SCs (Figs. 5G and EV4C).

Finally, we assessed the effect on Rab11-exosome secretion by co-expressing in SCs both Aβ-42 peptides and a GFP-labelled form of the FGF receptor, Breathless-GFP (Btl-GFP), a marker of Rab11-exosomes (Fan et al, 2020; Marie et al, 2023; Dar et al, 2021), and

counting the number of GFP-labelled puncta in the AG lumen (Appendix Fig. S9C). The number of secreted puncta were unaffected in all three cases when compared to control glands (Appendix Fig. S9D).

Overall, we conclude that in the fly model, mutant Aβ-peptides modulate intracellular DCG protein aggregation events and disrupt

the endolysosomal system, before any detectable extracellular aggregates are observed. Furthermore, the endocytosis of secreted DCG proteins is altered, modulating the endolysosomal organisation of the cells involved.

# Discussion

APP cleavage products are thought to be central players in amyloidogenic protein aggregation events in AD. However, Aβ's roles in the early stages of this disease remain unclear, though changes in secretion and endolysosomal trafficking, which can influence APP cleavage, are observed (Kimura and Yanagisawa, 2018), while loss of APP modulates secretory processes (Nalivaeva and Turner, 2013). Visualising APP- and Aβ-modulated events that occur inside secretory compartments, which are typically at the limit of light microscope resolution, is challenging.

By studying APP function ex vivo in a *Drosophila* cell type with highly enlarged secretory and endosomal compartments, we demonstrate that the fly APP homologue, APPL, specifically regulates protein aggregation events that are essential for the formation of a large DCG. Normal DCG biogenesis requires transmembrane APPL, but subsequent proteolytic release of the APPL extracellular domain is also critical. If this latter event fails, or APPL expression is suppressed, or pathological Aβ-42 peptides are expressed in SCs, DCG biogenesis and compartment maturation are disrupted. This leads to defective compartment trafficking and targeting for lysosomal degradation, mirroring early AD-associated neuronal phenotypes (Kimura and Yanagisawa, 2018).

## TGFBI homologue, MFAS, drives protein aggregation in SC DCGs

Using real-time imaging of the *GFP-mfas* gene trap, we visualised the protein aggregation events that lead to the formation of the very large (~3-µm diameter) SC DCG. Often, a cloud of GFP-MFAS protein rapidly aggregates into a central core via a non-homogeneous intermediate. However, in other cases, many mini-cores initially form at the periphery of the compartment, then become mobile and fuse together. *GAPDH2* knockdown, which suppresses ILV clustering (Dar et al, 2021), appears to inhibit fusion, perhaps because GAPDH2 promotes ILV adhesion required for mini-core fusion.

Knockdown experiments revealed that MFAS, an extracellular matrix protein originally reported to be membrane-associated (Hu et al, 1998), is essential for both mini-core and large DCG biogenesis (Fig. 1). The human MFAS orthoologue, TGFBI, contains a C-terminal integrin-binding RGD domain, but otherwise is structurally very similar (Nielsen et al, 2020). *TGFBI* mutations lead to autosomal dominant corneal dystrophy by forming extracellular amyloid-like deposits (Han et al, 2016). Interestingly, human APP and TGFBI are co-expressed in corneal cells (Choi et al, 2019) and associate together in amyloid-like deposits in calcific aortic valve disease (Heuschkel et al, 2020).

Our work suggests that MFAS naturally forms large aggregates under specific microenvironmental conditions. These conditions are met during the Rab11 transition in maturing SC DCG compartments (Wells et al, 2023), a process that appears to be conserved in human cells (Stockhammer et al, 2024). They likely involve changes in pH and/or ion concentrations, which have previously been implicated in DCG biogenesis (Gondré-Lewis et al, 2012; Yoo and Albanesi, 1990). However, GFP-MFAS levels increase during the hour before DCG assembly (see Movies EV1 and EV2), which is presumably also critical.

TGFBI is the most abundant protein in supermeres, non-vesicular human multimolecular extracellular signalling nanoparticles, which include a diverse range of proteins and RNAs (Zhang et al, 2021). These particles also contain the cleaved extracellular domain of APP, extracellular proteases and GPI-anchored proteins, all of which are present in SC DCGs (Fig. 4C) (Rylett et al, 2007; Redhai et al, 2016; Corrigan et al, 2014). This suggests a link between SC DCGs, Rab11-exosomes and supermeres, which now requires further investigation.

## APP regulates protein aggregation events, and its cleavage is required for normal DCG assembly and secretory trafficking

Rab11-exosomes are a novel exosome subtype made in Rab11-labelled recycling endosomal compartments and involved in physiological and pathological intercellular signalling (Fan et al, 2020; Marie et al, 2023). Here we show that in SCs, newly-formed Rab11-exosomes have an additional role together with endosomal membranes, co-ordinating the normal protein aggregation events required for large DCG biogenesis. This process requires APPL (Fig. 7), whose function can be partly substituted by human APP. APPL is the only APP homologue in flies; it is strongly expressed in the nervous system, but is also transcribed in other tissues (Leader et al, 2018).

We found that overexpressing wild-type forms of APPL and APP induces increased lysosomal targeting of DCG compartments (Figs. 3E, 4 and 5E) without affecting general cell morphology and viability (quantified for APPL in Appendix Fig. S5). Early DCG compartment maturation, most notably the Rab6 to Rab11 transition and large DCG biogenesis, appears to occur relatively normally (Appendix Fig. S4). Analysis of a double-tagged APPL protein revealed that intact APPL localises to the limiting membrane of DCG precursor compartments, but is cleaved during ILV and DCG biogenesis. APPL's ECD associates with the DCG, and some membrane-linked ICD remains on ILVs and the compartment's limiting membrane, while most is targeted to lysosomes (Fig. 4B,C). If APPL cannot be cleaved at its α- and β-secretase sites, aggregated protein does not fully dissociate from the limiting membrane of DCG compartments, more compartments are targeted for lysosomal degradation (Fig. 5), and SCs start to develop major morphological defects (Appendix Fig. S5).

Knocking down individual secretases also increases the level of DCG compartment lysosomal targeting (Fig. EV3). In these cases, most DCGs detach from the compartment's limiting membrane, presumably because APPL's ECD can still be cleaved off by alternative proteases. *β-secretase* knockdown produces the most pronounced effects on DCG biogenesis, which is of particular interest, given the reported co-localisation of mammalian BACE1 and APP in Rab11-positive endosomes of neuronal dendritic spines and presynaptic boutons (Das et al, 2016). The primary DCG defect in *β-secretase* knockdown SCs is the absence of GFP-MFAS from the DCG's centre, a phenotype also occasionally observed in *Appl* knockdown cells rescued by APP-YFP (Fig. EV2L). One possible

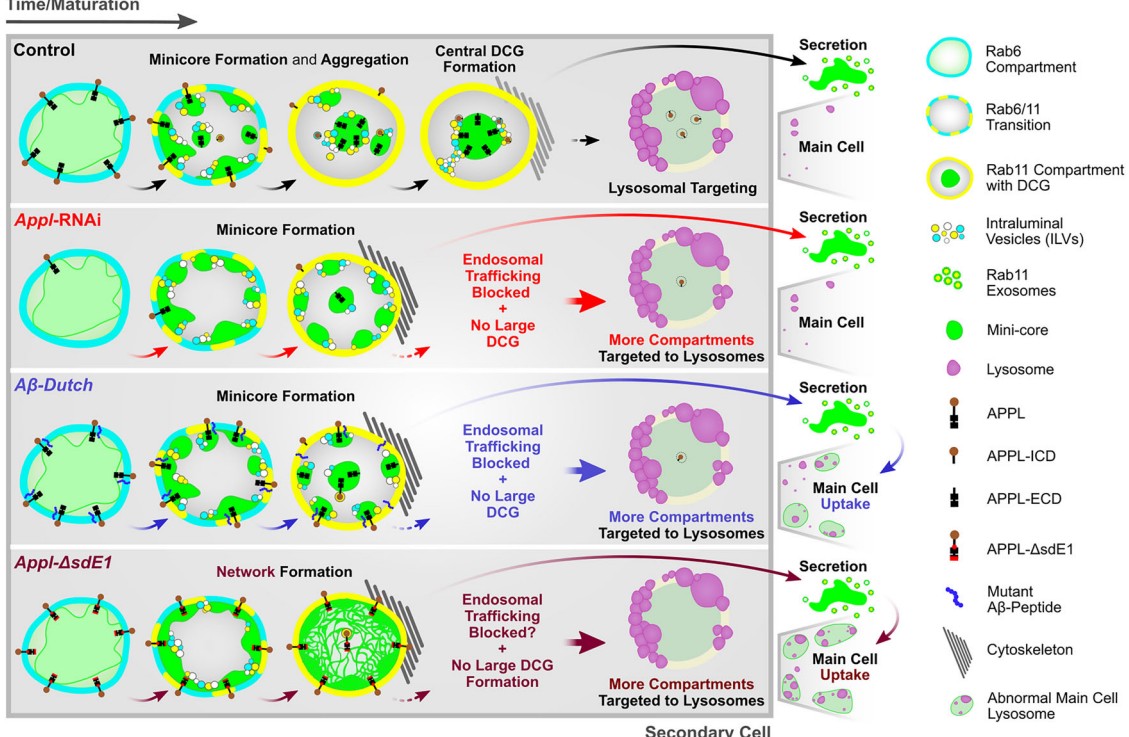

**Figure 7. Schematic model of APPL-regulated DCG formation in *Drosophila* secondary cells.**

Schematic shows DCG assembly process in control conditions (top row) with large central DCG (green) forming from peripheral mini-cores that interact with both the compartment's limiting membrane and ILVs. Formation of protein aggregates detectable by DIC requires a Rab6 to Rab11 transition (Wells et al, 2023), and assembly of a large central DCG involves clustering of ILVs regulated by GAPDH (Dar et al, 2021). *Appl* knockdown or pathological (Dutch) mutant Aβ-42 expression suppresses coalescence of mini-cores, increases lysosomal targeting, but also suppresses secretory compartment motility, presumably by stabilising cytoskeletal interactions. GFP-MFAS is more readily endocytosed by main cells, when non-cleavable forms of APPL or Aβ-peptides are expressed, leading to lysosomal enlargement in these cells. The APPL-ΔsdE1 mutant also promotes limiting membrane-associated MFAS aggregation, producing a peripheral network.

explanation is that dynamic central clustering of peripheral GFP-MFAS-positive aggregates during maturation is abnormal, perhaps because membrane:aggregate dissociation is incomplete or delayed.

As with overexpressing non-cleavable APPL, when APPL is reduced or lost, aggregated MFAS more frequently associates with the limiting membrane of DCG compartments, either as mini-cores in knockdown cells or as a single large core in the *Appl* null mutant (Figs. 3 and EV2). The phenotypic difference between knockdown and mutant may either reflect incomplete knockdown or the accumulation of genetic modifiers in the homozygous mutant line over time. Whatever the explanation, the mini-core knockdown phenotype is rescued by human APP, suggesting that APPL's protein aggregation activity is evolutionarily conserved. We conclude that in the absence of APPL, MFAS aggregation can still take place, but aggregates do not fully dissociate from membranes, a step normally regulated by APPL cleavage (Fig. 7).

Secretory compartment maturation in both *Appl* mutant and knockdown cells appears disrupted because these compartments are preferentially targeted for lysosomal degradation. Furthermore, in the knockdown, where mini-core movement can be followed in real-time, DCG compartments become visibly less motile within the cytoplasm, perhaps because links with the cytoskeleton are abnormally stabilised. A recent transcriptomics and proteomics analysis of *Appl* mutant flies has highlighted changes in proteostasis

and autophagy, which are also detected in humans and mice (Nithianandam et al, 2023). Our data provide a potential cellular explanation for activation of these degradative pathways.

What triggers targeting of DCG compartments for lysosomal degradation, either when APPL levels are altered, or when APPL fails to be cleaved? In these cases, DCG aggregates typically remain abnormally associated with the compartment's limiting membrane. Therefore, this defect may have the combined effect of disrupting compartment maturation and targeting compartments to degradative compartments, in some cases, also inhibiting endolysosomal trafficking.

APPL not only controls the maturation of DCG compartments and dissociation of membrane:aggregate complexes, it also regulates MFAS-mediated protein aggregation itself. The unique peripherally concentrated network of GFP-MFAS aggregation induced by non-cleavable APPL lacking a functional extracellular E1 domain (Fig. 5A,B) strongly suggests that the E1 domain modulates aggregate priming activity of membranes or membrane:aggregate stability. In human APP, the E1 domain is involved in divalent cation-regulated APP dimerisation (August et al, 2019), and also undergoes a conformational change at acidic versus neutral pH (Hoefgen et al, 2015). These microenvironmental changes are both involved in regulating DCG aggregation (Germanos et al, 2021; Jayawardena et al, 2019; Yoo and Albanesi,

1990; Wu et al, 2001). The E1 domain, therefore, provides a potential microenvironmental sensor for regulation of protein aggregate priming that could be critically relevant to aggregation-associated pathologies.

## Pathological Aβ-peptides disrupt DCG biogenesis and compartment motility

Expression in SCs of mutant Aβ-42 peptides that induce neurodegeneration in mammals and flies also disrupted DCG compartment maturation (Fig. 6A), phenocopying changes in SCs with reduced APPL function. Most notably, although compartments mature to Rab11-identity, they contain mini-cores at their periphery. More of these compartments become acidified and are also immobile, a phenotype observed several days before SCs start to exhibit major morphological defects. Defective neuronal endolysosomal trafficking is an early marker of AD (Peric and Annaert, 2015; Kimura and Yanagisawa, 2018), and our study suggests that this may be linked to dysfunction of a physiological APP-dependent process in secretion. Indeed, APPL does not appear to be cleaved and trafficked normally in the presence of Aβ-42 peptides (Appendix Fig. S7), which may be of particular significance, given that Aβ-oligomers have been proposed to exert their pathological effects by interfering with APP function (Puzzo et al, 2017; Wang et al, 2017).

We previously observed that DCG compartments occasionally fuse with large lysosomes in normal SCs (Corrigan et al, 2014). Here, we found that compartments also sporadically become acidified by small peripheral acidic structures (Fig. EV1G). This DCG acidification phenotype is observed much more frequently following *Appl* knockdown or expression of either non-cleavable APPL or Aβ-peptides, perhaps because trafficking of these compartments to large lysosomes is suppressed in these genetic backgrounds. In fact, most of these defective compartments still appear to secrete their DCGs and ILVs (Appendix Fig. S9), and their DCG contents can accumulate in other cells, disrupting their lysosomal degradation (Figs. EV4 and EV5). This suggests that changes in APP-dependent protein aggregation can induce lysosomal defects both in secretory cells and in other cells that endocytose these secretions.

## Relationship of APPL's subcellular roles in SCs to neuronal APP functions and AD pathology

Are the functions of APPL in SCs relevant to APP's roles in neurons? When APP or APPL is expressed in other *Drosophila* cell types, such as neurons, it is trafficked and processed in the regulated secretory pathway (Ramaker et al, 2016; Neuman et al, 2021). Neuronal loss of APPL function leads to learning and behavioural phenotypes, and defects in neuronal outgrowth and synaptogenesis (Luo et al, 1992; Mora et al, 2013; Torroja et al, 1999). Several studies support the idea that Aβ-peptide-induced tissue phenotypes and associated neurodegeneration in humans and flies represent APP loss-of-function phenotypes, potentially linked to altered secretory trafficking in neurons (Gouras et al, 2014; Kepp, 2016; Singh et al, 2017; Kim and Bezprozvanny, 2023).

Could the Rab6/Rab11-regulated DCG pathway characterised in SCs (Wells et al, 2023) and also potentially employed by human cells (Stockhammer et al, 2024) participate in early neuronal events

associated with APP function and AD? Subcellular localisation of the ECD and ICD of APP and APPL in SCs as well as fly and human neurons suggests that secretase cleavage can take place within secretory and endosomal compartments (Muresan et al, 2009; Yuyama and Igarashi, 2017). Furthermore, Rab6 is reported to play a role in regulating the proteolytic processing of APP (McConlogue et al, 1996; Ramaker et al, 2016), as is the adaptor protein complex AP-1 (Januário et al, 2022), which is essential for the trafficking that controls DCG biogenesis in SCs (Wells et al, 2023).

In addition, Rab11-directed trafficking is implicated in the cleavage of APP and formation of Aβ-peptides (Das et al, 2016; Udayar et al, 2013; Sultana and Novotny, 2022), processing events that appear to take place in maturing synaptic vesicles (Groemer et al, 2011; Del Prete et al, 2014). Munc-18-interacting proteins (Mints), adaptor proteins that bind to the endocytic sorting motif of APP, regulate the secretion of Aβ-peptides from mammalian neurons (Ho et al, 2008; Sullivan et al, 2014), while in flies, APPL/Mint interactions are implicated in synapse formation (Ashley et al, 2005). Both Mint1 and Munc-18 control DCG secretion in neuroendocrine cells (Schütz et al, 2005).

Finally, exosomes are associated with protein aggregation and APP-mediated functions in flies and humans. In *Drosophila*, larval neuromuscular junction establishment, which is regulated by APPL (Ashley et al, 2005), is also controlled by Rab11-dependent exosomes (Koles et al, 2012). Furthermore, exosomes of unknown endosomal origin (Edgar et al, 2015) have a role in Aβ-peptide secretion, propagation and plaque formation (Sardar Sinha et al, 2018; Kaur et al, 2021), and exosome-containing compartments have recently been implicated in early Aβ oligomerisation events (Eckman et al, 2023). Our study suggests Rab11-exosomes may be responsible, and is supported by the observation that Rab11-exosome markers are enriched in the CSF secretome from AD patients (Appendix Table S2) (Li et al, 2023). Indeed, despite the endolysosomal trafficking defect observed in SCs following Aβ-42 expression, secretion of DCG compartments appears to be increased.

The *Drosophila* SC model offers new opportunities to study specific events in pathological amyloidogenesis in vivo. For example, GAPDH has been suggested to act as a seed for Aβ amyloidogenesis (Gerszon and Rodacka, 2018), a proposal that can now be tested genetically in flies. Furthermore, it will be interesting to further study the intercellular propagation of aggregates induced by uncleaved APPL and Aβ-peptides to determine whether it mirrors amyloid spreading in AD. Finally, cross-seeding between amyloid proteins has been observed experimentally (Subedi et al, 2022) and proposed to explain the unusually high co-morbidities of amyloidogenic diseases (Spires-Jones et al, 2017). The observation that orthologues of two amyloidogenic proteins, APP and TGFBI, normally co-aggregate via reversible interaction with membranes might provide clues regarding the biological basis of these phenomena and explain previous observations made in vitro (Yam et al, 2012).

AD may share some cell and molecular defects with other neurodegenerative diseases (Tofaris and Buckley, 2018). In this regard, Parkinson's Disease is commonly associated with endolysosomal trafficking phenotypes, while CHMP2B and CHMP1B, whose homologues both play roles in Rab11-exosome and DCG biogenesis in fly SCs (Marie et al, 2023), are linked to

frontotemporal dementia and hereditary spastic paraplegia respectively (Goedert et al, 2012; Reid et al, 2005). Whether defects in protein aggregate dissociation from endosomal and exosomal membranes are a common feature of these diseases warrants further investigation.

# Methods

### Reagents and tools table

| Reagent/resource | Reference or source | Identifier or catalogue number |
|---|---|---|
| **Experimental models** | | |
| *Drosophila: w[1118]* | Partridge Lab: UCL | N/A |
| *Drosophila: w[1118];; GFP-mfas[MI11275-GFSTF.2]* | BDSC | # 63204 |
| *Drosophila: w[1118]; if/CyO; dsx-GAL4/TM6B* | Goodwin Lab, Oxford; Rideout et al, 2010 | N/A |
| *Drosophila: w[1118]; tub-GAL80[ts]; TM2/TM6B,Tb* | BDSC | # 7108 |
| *Drosophila: w[1118]; tub-GAL80[ts]/CyO; dsx-GAL4/TM6B* | Wilson Lab; Corrigan et al, 2014 | N/A |
| *Drosophila: y, w[1118]; CFP-Rab6* | Eaton Lab, Max Plank; Prince et al, 2019 | N/A |
| *Drosophila: w[1118]; CFP-Rab6, tub-GAL80[ts]/CyO; dsx-GAL4* | Wilson Lab; Wells et al, 2023 | N/A |
| *Drosophila: y, w[1118]; YFP-Rab11* | Eaton Lab, Max Plank, Dresden (and BDSC); Dunst et al, 2015 | # 62549 |
| *Drosophila: w[1118]; tub-GAL80[ts]/CyO; YFP-Rab11, dsx-GAL4* | Wilson Lab; Redhai et al, 2016 | N/A |
| *Drosophila: (td-GFP-mfas) w[1118]; tub-GAL80[ts]/CyO; dsx-GAL4, GFP-mfas/TM6B* | This study | N/A |
| *Drosophila: y, sc, v; UAS-rosy-RNAi* | BDSC | # 44106 |
| *Drosophila: (mfas #1) y, sc, v; UAS-mfas-RNAi* | BDSC | # 52905 |
| *Drosophila: (mfas #2) y, v; UAS-mfas-RNAi* | BDSC | # 58256 |
| *Drosophila: y, v; UAS-GAPDH2-RNAi* | BDSC | # 26302 |
| *Drosophila: (Appl #1) w[1118];; UAS-Appl-RNAi* | VDRC | # 42673 |
| *Drosophila: (Appl #2) y, v;; UAS-Appl-RNAi* | BDSC | # 28043 |
| *Drosophila: y, w[1118]; UAS-APP-YFP* | BDSC | # 32039 |
| *Drosophila: (Aβ-42 WT) w[1118]; UAS-Aβ[42]Wild-type – with N-terminal ER signal sequence* | BDSC | # 33769 |
| *Drosophila: (Aβ-42-Dutch) w[1118]; UAS-Aβ[42]E693Q-Dutch – with N-terminal ER signal sequence* | BDSC | # 33775 |
| *Drosophila: (Aβ-42 Iowa) y, w[1118]; UAS-Aβ[42]D694N-Iowa – with N-terminal ER signal sequence* | BDSC | # 33779 |
| *Drosophila: (α-secretase) y, sc, v; UAS-kuzbanian-RNAi* | BDSC | # 66958 |
| *Drosophila: (β-secretase) w[1118];; UAS-bace-RNAi* | VDRC | # 15541 |
| *Drosophila: (γ-secretase) y, v; UAS-nicastrin-RNAi* | BDSC | # 27498 |
| *Drosophila: Appl[d] w[1118]* | BDSC | # 43632 |
| *Drosophila: (Appl-WT) w[1118]; UAS-Appl.T* | BDSC | # 38403 |
| *Drosophila: (Appl-Δsd) w[1118]; UAS-Appl.sd* | BDSC | # 29863 |
| *Drosophila: (Appl-ΔsdE1) w[1118];; UAS-Appl.sdDeltaE1* | BDSC | # 29866 |
| *Drosophila: w[1118]; UAS-EGFP-Appl-mRFP /TM3,Sb* | Kretzschmar Lab, OHSU; Ramaker et al, 2016 | N/A |
| *Drosophila: w[1118]; UAS-Appl-EGFP* | Godenschwege Lab, FAU; Penserga et al, 2019 | N/A |
| *Drosophila: w[1118]; UAS-GFP-GPI (3.1)* | Eaton Lab, Max Plank, Dresden; Greco et al, 2001 | N/A |
| *Drosophila: w[1118]; tub-GAL80[ts]/CyO; dsx-GAL4, UAS-btl-GFP/TM6* | Wilson Lab; Fan et al, 2020 | N/A |
| **Recombinant DNA** | | |
| **Antibodies** | | |
| Mouse anti-Tubulin | Sigma | #T8328 |
| Mouse anti-CD81 | Santa Cruz | #23962 |
| Mouse anti-CD63 | BD Biosciences | #556019 |
| Rabbit anti-Syntenin-1 | Abcam | ab133267 |
| Rabbit anti-Tsg101 | Abcam | ab125011 |
| Mouse anti-hGAPDH | DSHB | 2G7 |
| Mouse anti-Rab11 | BD Biosciences | #610657 |
| anti-mouse IgG (H + L) HRP conjugate | Promega | #W4021 |
| anti-rabbit IgG (H + L) HRP conjugate | Promega | #W4011 |
| **Oligonucleotides and other sequence-based reagents** | | |
| **Chemicals, Enzymes and other reagents** | | |
| LysoTracker™ Red DND-99 | Thermo Fisher Scientific | # L7528 |
| Paraformaldehyde | Sigma-Aldrich | # 16005 |
| Phosphate Buffer Saline (PBS) | Thermo Fisher Scientific | # P4417 |
| Vectashield with DAPI | Vector Laboratories | # H-1200 |
| **Software** | | |
| Fiji | ImageJ | imagej.net |
| Prism 9.0 | GraphPad | graphpad.com |
| **Other** | | |

## Fly stocks and husbandry

All *Drosophila* strains used in this study are detailed in the key resource table. Where possible, RNAi lines were selected that have already been employed for gene knockdown in other studies, which are highlighted. The transgenic lines were acquired from Bloomington *Drosophila* Stock Centre (BDSC) and Vienna *Drosophila* Resource Centre (VDRC), unless otherwise stated: *dsx-GAL4* (provided by S. Goodwin, Oxford, UK) (Rideout et al, 2010), *CFP-Rab6* (Prince et al, 2019) and *YFP-Rab11* fusion genes at endogenous *Rab* locus (provided by S. Eaton, Max Plank, Germany) (Dunst et al, 2015), *GFP-mfas* gene trap (MI11275-GFSTF.2; BDSC 63204) (Nagarkar-Jaiswal et al, 2015), *UAS-rosy*-RNAi (TRiP.HMS02827; BDSC 44106) (Marie et al, 2023), *UAS-mfas*-RNAi #1 (TRiP.HMC03645; BDSC 52905) (Ni et al, 2011) and #2 (TRiP.HMJ22320; BDSC 58256) (Ni et al, 2011), *UAS-GAPDH2*-RNAi (TRiP.JF02072; BDSC #26302) (Spannl et al, 2021; Marie et al, 2023), *UAS-Appl*-RNAi #1 (GD3170; VDRC 42673) (Goguel et al, 2011) and #2 (TRiP.JF02878; BDSC 28043) (Singh and Mlodzik, 2012), *UAS-APP-YFP* (BDSC 32039) (Gunawardena et al, 2003), *UAS-Aβ-42 WT* (BDSC 33769) (Wu et al, 2017), *UAS-Aβ-42-Dutch* (BDSC 33775; Vitruvean), *UAS-Aβ-42 Iowa* (BDSC 33779; Vitruvean) (Chouhan et al, 2016) (all Aβ-42 constructs include a cleavable ER signal sequence to direct them to the secretory system), *UAS-α-secretase*-RNAi (TRiP.HMS05424; BDSC 66958; *kuz*) (Tian et al, 2023), *UAS-β-secretase*-RNAi (GD5366; VDRC 15541; *Bace*) (Bolkan et al, 2012), *UAS-γ-secretase*-RNAi (TRiP.JF02648; BDSC 27498; *Nct*) (Restrepo et al, 2022), *Appl^d* (BDSC 43632) (Luo et al, 1992), *UAS-Appl-WT* (BDSC 38403) (Torroja et al, 1999), *UAS-Appl-Δsd* (BDSC 29863) (Torroja et al, 1999), *UAS-Appl-ΔsdE1* (BDSC 29866) (Torroja et al, 1999), *UAS-GFP-GPI* (Redhai et al, 2016; Greco et al, 2001) (provided by S. Eaton, Dresden, Germany), *w^1118* (provided by L. Partridge, UCL, UK), *UAS-GFP-Appl-RFP* (provided by D. Kretzschmar, OHSU, USA) (Ramaker et al, 2016), *UAS-Appl-GFP* (provided by T. Godenschwege, FAU, USA) (Penserga et al, 2019), SC-specific temperature-sensitive driver lines were generated by combining *dsx-GAL4* with ubiquitously expressed repressor *tub-GAL80^ts* (BDSC 7108) to produce *tub-GAL80^ts; dsx-GAL4* (Fan et al, 2020). This driver line was additionally combined with endogenously fluorescent-tagged *GFP-mfas, CFP-Rab6* or *YFP-Rab11* to generate *CFP-Rab6, tub-GAL80^ts; dsx-GAL4* (Wells et al, 2023), *tub-GAL80^ts; YFP-Rab11, dsx-GAL4* (Fan et al, 2020), and *tub-GAL80^ts; dsx-GAL4, GFP-mfas* (this study).

Flies were maintained at 25 °C under a 12-hour light/dark cycle on standard cornmeal agar medium [12.5 g agar (F.Gutlind & Co. Ltd), 75 g cornmeal (B. T. P. Drewitt), 93 g glucose (Sigma-Aldrich, #G7021), 31.5 g inactivated yeast (Fermipan Red, Lallemand Baking), 8.6 g potassium sodium tartrate tetrahydrate (Sigma-Aldrich, #S2377), 0.7 g calcium chloride dihyrdrate (Sigma-Aldrich, #21907), and 2.5 g nipagin (Sigma-Aldrich, #H5501) dissolved in 12 ml ethanol, per litre]. They were transferred onto fresh food every 3–4 days. Female flies carrying the driver line *tub-GAL80^ts; dsx-GAL4* alone or in combination with *GFP-mfas, CFP-Rab6* or *YFP-Rab11* (and in some cases, UAS-regulated forms of Appl or Aβ-42) were crossed with male flies carrying UAS-transgenes to induce a temperature-controlled SC-specific expression of target genes. These crosses were maintained at 25 °C. Virgin male offspring were collected upon eclosion and typically transferred

to 29 °C for 6 days to activate post-developmental SC-specific transgene expression. Dissection and imaging were then performed.

## Preparation of accessory glands for imaging

Accessory glands were dissected and prepared as described in Fan et al, 2020 and Wells et al, 2023. For live-cell imaging, six-day-old adult male virgin flies were anaesthetised using $CO_2$. These flies were submerged in ice-cold 1X PBS (Thermo Fisher Scientific) during the micro-dissection procedure. The male reproductive tract was taken out of the body cavity by carefully pulling the last abdominal segment with micro-forceps. The testes, seminal vesicles, ejaculatory bulb, fat tissues and the gut were gently removed to avoid tissue folding and interference during accessory gland imaging. The glands were then incubated with 500 nM Lysotracker Red (Thermo Fisher Scientific) for 5 min on ice, followed by a wash with ice-cold 1x PBS. The ex vivo-prepared glands were stably mounted between two coverslips (rectangular: Coverslip No.1, 22 mm × 50 mm, Fisher, #1237-3128, and round: coverslip No.1, 13 mm, #49492, VWR) in a drop of 1x PBS, held together by a custom-built metal holder. Excess PBS was removed using a filter paper until the glands were slightly flattened and ready for imaging.

For confocal analysis, micro-dissections were performed in PBS containing 4% paraformaldehyde (Sigma-Aldrich). The glands were fixed for 20 min at room temperature and then washed in 1x PBS for 3 × 5 min prior to mounting onto SuperFrost microscope slides (VWR). After removing excess PBS using Whatman paper (Whatman), the glands were immersed in a drop of Vectashield with DAPI (Vector Laboratories) and stably positioned with a coverslip (22 mm × 22 mm, 0.13–0.17 mm, Fisher).

## Live-cell imaging, deconvolution and time-lapse movies

Based on Wells et al, 2023, live SCs were imaged at room temperature using a DeltaVision Elite wide-field fluorescence deconvolution microscope (GE Healthcare Life Sciences) at 100x (Olympus UPlanSApo NA 1.4; oil objective), using immersion oil with a refractive index of 1.514 (Cargille Labs) and a manual auxiliary magnification of 1.6x. An EMCCD Evolve-512 camera was used to capture images. Three SCs were imaged per accessory gland for ≥10 individual virgin males. The images acquired were typically z-stacks spanning a depth of 8–12 μm with a z-distance of 0.2 μm. The Resolve 3D-constrained iterative deconvolution algorithm within SoftWoRx 5.5 Software (GE Healthcare Life Sciences) was subsequently used to deconvolve z-stack images to improve the image quality prior to analysis.

For the DV Elite, SC morphology was visualised using Differential Interference Contrast (DIC), with the following laser settings: GFP (FITC), 475 nm excitation at 10% laser intensity; YFP, 513 nm excitation at 32% laser intensity; CFP, 440 nm excitation at 50% laser intensity; and RFP (mCherry), 550 nm excitation at 10% laser intensity.

For some experiments in the EV Figures and Appendix Figures, imaging was performed using the Leica Thunder inverted wide-field microscope (Leica) at 63x (Leica HC PL APO NA 1.4, oil objective) and 100x (Leica HCX PL FLUOTAR NA 1.3, oil objective) with a K8 sCMOS camera. The z-stack thickness and distance were consistent with those used on the DV Elite. Thunder

technology with small volume computational clearing (SVCC) was applied to enhance contrast and eliminate out-of-focus blur. SC morphology was visualised using FLURO-Bright Field imaging. The LED settings were as follows: GFP, 475 nm excitation at 10% laser intensity; YFP, 510 nm excitation at 30% laser intensity; and RFP, 550 nm excitation at 20% laser intensity.

For time-lapse imaging experiments, samples were prepared using the same method except that incubation with Lysotracker Red (Thermo Fisher Scientific) was typically excluded unless compartment acidification was being assessed. Time-lapse imaging was conducted using the DeltaVision Elite wide-field fluorescence deconvolution microscope (GE Healthcare Life Sciences) using identical methods to those described above, with the following exceptions. Firstly, auxiliary magnification was not utilised, in order to reduce the effects of cell/tissue drift over time, except for the control, *rosy-RNAi* cell in Movie EV1. Secondly, whilst z-stacks of entire cells were acquired for time-lapse imaging experiments, for all but Movie EV2, the z-distance between individual slices was increased from 0.2 μm to between 0.5 and 0.8 μm to limit photobleaching and phototoxicity. The gap between timepoints during these experiments was 60 s for all videos shown here except Movie EV2, where the time interval was 90 s; additional videos were acquired with gaps of up to 120 s between frames. Finally, deconvolution was used as described above, but the final results of deconvolution were saved as 32-bit floating-point images.

In order to best show all structures and fluorescence in compartments of interest, the images displayed in videos were made using z-projections that included the fluorescent signal from every z-slice within the bounds of the relevant compartment, typically four to eight image slices in the GFP-MFAS channel. For the DIC channel, a single representative z-slice was used generally taken from the centre-most point of the compartment.

## Fixed accessory glands imaging

Fixed samples were imaged on a Zeiss LSM980 with Airyscan 2 Super-resolution upright laser scanning confocal microscope equipped with 10x (Zeiss 0.45 NA; dry) and 40x (Zeiss 1.30 NA; oil; Zeiss immersion oil, refractive index 1.518) objectives. High-resolution images of the accessory gland lumen were acquired using the 40x objective with 0.7x zoom on the ZEN blue suite Software (Zeiss). For Btl-GFP puncta counts, the images were captured from the central-third region of the accessory gland. Z-stacks were acquired with a 2-μm interval, 18-μm thickness, with a total of ten slices.

## Analysis and parameters

### DCG phenotypes and number of mature compartments
Deconvolved images were analysed using Fiji/ImageJ. The number of intact DCGs marked by GFP-MFAS and DCG-containing compartments were scored using the StarDist 2D plugin, DSB 2018 model, which identifies each core as a separate entity. In the case of *mfas* knockdowns, the number of compartments were quantified manually using the differential interference contrast (DIC) channel.

Abnormal cores included both those having a GFP-negative (GFP-) centre and those with mini-/deformed cores, with the exception of cells overexpressing the *Appl*$^{sdΔE1}$ mutant, which exhibited an additional peripheral network-like compartment

phenotype. DCGs were scored as containing a GFP- centre if they contained a non-fluorescent centre with ≥1 μm diameter, while the mini-/deformed core phenotype was defined by the presence of multiple small cores of diameter ≥0.5 μm and/or a misshapen core, where the ratio of the lengths of the longest and shortest DCG axes was >1.4. Compartments with the peripheral network-like phenotype failed to form a round central DCG, and most of the aggregated GFP-MFAS was in close proximity to the limiting membrane. These latter abnormal cores/compartments were manually scored using the z-stack in both DIC and Merge channels. The percentage of abnormal and mini-/deformed cores was calculated relative to the total number of non-acidic DCG- (GFP-MFAS-) containing compartments per SC.

Intact or abnormally shaped DCGs that maintained contact with the limiting membrane through several planes in the z-stack were scored manually and were represented as a percentage relative to the total number of DCG compartments per SC.

To determine the frequency at which mini-cores either collide or overlap along the same z-axis in individual compartments, a single mini-core associated with the limiting membrane of each compartment was selected. Each instance of the mini-core's fluorescent signal moving to overlap with other mini-cores and/or being in contact with another mini-core at the start of the movie was considered a mini-core overlap. Images were acquired each minute over a duration of 30 min, and the total number of overlaps was recorded.

To quantify compartments marked by either the CFP-Rab6 or YFP-Rab11 fusion protein expressed from the endogenous *Rab* locus, fluorescently labelled compartments were manually examined using z-stacks for the CFP or YFP channels (Wells et al, 2023). The proportion of these compartments containing *Rab6/11*-positive exosomes was established by evaluating which compartments contained internal fluorescent puncta in these genetic backgrounds.

### Acidification of secretory compartments
Mature DCG compartments that are associated with acidic structures and potentially undergoing lysosomal clearance (the DCG acidification phenotype) were scored as acidified compartments. These compartments manifest different phenotypes depending on the stage of lysosomal clearance. Some have a single peripheral lysosomal structure with a slightly diffuse DCG that has maintained its shape, while others have completely diffuse GFP-MFAS and no obvious DCG structure by DIC. Some feature multiple acidic structures arranged in an arc around the compartment boundary, with or without diffuse GFP-MFAS. Others are >80% covered by acidified domains, but the compartments still retain their circular shape. The percentage of acidified compartments per SC was determined as a proportion of the total number of both non-acidic DCG-containing compartments and acidified compartments per SC.

To calculate the lysosomal area, a freehand tool on Fiji was employed to outline and measure the area of the SC. The threshold and analyze particles tools were utilised to determine the lysosomal area in the complete projection of the lysotracker (RFP) channel. The percentage of lysosomal area per SC was then calculated relative to the total area of the SC.

### GFP-MFAS-containing main cells
The Rectangle tool on Fiji was used to determine the area of the field of view, and the freehand tool was employed to outline and

measure the area of the SC. Subsequently, the threshold and analyse particles tools were applied to measure the area covered by GFP-MFAS in the main cells using the complete projection of the GFP channel. The percentage of the main cell area containing GFP-MFAS was calculated within the field of view, excluding the SC area.

### Counting morphologically abnormal SCs

Using the FLUORO-Bright-Field channel on the Thunder microscope, secondary cells with abnormal morphology were identified at the distal tip of the AG lobe. The plasma membrane of these cells was difficult to detect, and they lacked large intact compartments with DCGs or aggregated protein within them. The percentage of total SCs with normal cellular morphology was calculated for each animal.

### GFP-MFAS and Btl-GFP secretion into the accessory gland lumen

The central region of fixed accessory glands was imaged to examine the luminal phenotype. Using the rectangle tool, two squares of equal size (150 × 150 pixels) were marked in the middle of each arm, resulting in a total of four squares per animal. The mean grey value of these four areas was calculated using the same settings for all images, and presented as a mean GFP-MFAS intensity per gland.

For the Btl-GFP exosome secretion assay, a maximum intensity projection was generated using the GFP channel (see also Corrigan et al, 2014 and Marie et al, 2023). A rolling pin value of 50 was applied to remove a smooth, even background from the images. The rectangle tool was used to create three squares of equal size (100 × 100 pixels) to mark the areas containing puncta within the AG lumen. A total of six squares per animal were analyzed and the puncta within the lumen were quantified and presented.

## Image processing and preparation for figures

Live-cell images for this study were prepared using deconvoluted stacks collected from the DV Elite microscope. In all figures, the DIC channel was a single-slice image chosen for its optimal representation of SC morphology. Max-projections of slices ranging from 3 to 10 μm were employed for all fluorescent channels, specifically highlighting DCG compartments and Rab6-/Rab11-compartments with ILVs. The fluorescence intensity that best captured the phenotype was selected and utilised for image processing across all the live-cell data. Consistent projection of slices and settings was applied to generate images for each genotype shown in the figures. All images were cropped to identical dimensions, focusing primarily on SC details. To enhance contrast, sharpen tool in ImageJ was uniformly applied to all images, ensuring unbiased and accurate data representation.

For fixed-gland images, a single slice that most accurately represented the phenotype was acquired using the LSM980 confocal microscope. To ensure consistency in fluorescent intensity quantification, the GFP gain settings were kept constant across all genotypes.

## Human sEV isolation

Human HCT116 were purchased from ATCC (CCL-247) to ensure authenticity and tested regularly for mycoplasma. HCT116 Rab11a-exosome analysis was undertaken using the techniques described in Fan et al, 2020 and Marie et al, 2023. Approximately $8–9 \times 10^6$ HCT116 cells were seeded per 15 cm cell culture plate, using 10–15 plates per condition, and allowed to settle for 16–18 h in complete medium before the 24-h EV collection. This involved culture for 24 h in serum-free DMEM/F12 medium without L-glutamine (#21331046; Life Technologies), which was supplemented with 1% ITS (Insulin-Transferrin-Selenium; #41400045 Life Technologies), and either 2.00 or 0.15 mM glutamine (Life Technologies). Cells grew to ~90% confluence by the end of the collection period. The culture medium was centrifuged at $500 \times g$ for 10 min at 4 °C and $2000 \times g$ for 10 min at 4 °C to remove cells, debris and large vesicles. The supernatant was filtered to remove remaining large EVs using 0.22-μm filters (Millex).

The filtrate was concentrated to a volume of ~30 ml using a tangential flow filtration (TFF) unit with a 100 kDa membrane (Vivaflow 50 R, Sartorius) coupled to a 230 V pump (Masterflex). The sEV suspension was then further concentrated using 100 kDa Amicon filters by sequential centrifugation at $4000 \times g$ for 10 min at 4 °C to give a final volume of 1 ml. This was injected into a size-exclusion column (column size 24 cm × 1 cm) containing Sepharose 4B (84-nm pore size) using an AKTA start system (GE Healthcare Life Science) and eluted with PBS, collecting 30 × 1 ml fractions. Fractions corresponding to the initial 'EV peak' (typically fractions two to five) were pooled in 100 kDa Amicon tubes to a final volume of ~100 μl.

## Nanoparticle tracker analysis (NanoSight®) of sEVs

Using the NS500 NanoSight®, between three and five 30-s videos were captured per EV sample at a known dilution (normalised to protein mass of secreting cells). Particle concentrations were measured within the linear range of the NS500 (~$2–10 \times 10^8$ particles per ml). Particle movement was analysed by NTA software 2.3 (NanoSight Ltd.) to determine particle size distribution and concentration.

## Western analysis

Both cell lysates and EV preparations were lysed in RIPA or 1X sample buffer. Protein preparations were dissolved in either reducing (with 5% β-mercaptoethanol) or non-reducing (for CD63 and CD81 detection) sample buffer and heated to 90–100 °C for 10 min before loading. A pre-stained protein ladder (Bio-Rad) was also used. Proteins were separated by electrophoresis using 10% mini-PROTEAN precast gels (Bio-Rad). For gels of sEV proteins, lanes were loaded with EV lysates extracted from the same protein mass of secreting cells. This ensured that changes in band intensity on the blots with glutamine depletion reflected a net change in secretion of the marker on a per-cell basis (see Fan et al, 2020).

Proteins were wet-transferred to polyvinylidene difluoride (PVDF) membranes at 100 V for 1 h using a Mini Trans-Blot Cell (Bio-Rad). Membranes were blocked with either 5% milk (CD63 detection) or 5% BSA in TBS buffer with Tween-20 (TBST) for 30 min and probed overnight at 4 °C with primary antibody diluted in blocking buffer. The membranes were washed for 3 × 10 min with TBST, then probed with the appropriate secondary antibodies for 1 h at 22 °C and then washed for 3 × 10 min. Signals were detected using the enhanced chemiluminescent detection reagent

(Clarity, Bio-Rad) and a Touch Imaging System (Bio-Rad). Relative band intensities were quantified by ImageJ.

Antibody suppliers, catalogue numbers and concentrations used were: mouse anti-Tubulin (Sigma #T8328, 1:4000), mouse anti-CD81 (Santa Cruz #23962, 1:500), mouse anti-CD63 (BD Biosciences # 556019, 1:500), rabbit anti-Syntenin-1 antibody (Abcam ab133267, 1:500), rabbit anti-Tsg101 (Abcam ab125011, 1:500), anti-GAPDH (DSHB hGAPDH-2G7, 1:250), mouse anti-Rab11 (BD Biosciences #610657, 1:500), anti-mouse IgG (H + L) HRP conjugate (Promega #W4021, 1:10000), anti-rabbit IgG (H + L) HRP conjugate (Promega #W4011, 1:10000).

### Comparative proteomics analysis of HeLa cell sEV preparations

Human HeLa cells were purchased from ATCC (CCL-2) to ensure authenticity and tested regularly for mycoplasma. Four paired samples of conditioned medium (serum-free basal medium [DMEM/F12] supplemented with 1% ITS [insulin-transferrin-selenium; #41400045 Life Technologies]) were collected over a 24-h period from glutamine-depleted and glutamine-replete HeLa cells, using the same culture conditions as for HCT116 cells, except that the cells were seeded at $4 \times 10^6$ cells per plate. sEVs were isolated by size-exclusion chromatography, as described above.

Each sEV sample was then lysed in 100 μL RIPA buffer for 30 min on ice, followed by 10 min centrifugation at $17,000 \times g$ at 4 °C. The clear supernatants were transferred to fresh tubes. Proteins were reduced with DTT (final concentration 5 mM) and alkylated with Iodoacetamide (final concentration 20 mM) for 30 min each and then precipitated with methanol/chloroform. They were then mixed with 600 μl methanol, 150 μl chloroform and 450 μl ddH$_2$O and centrifuged again for 3 min at $17,000 \times g$. The upper phase was carefully removed, and a further 450 μl of methanol was added. After another centrifugation step for 5 min, supernatants were removed and discarded. The protein pellets were resuspended in 50 μl 100 mM TEAB and digested with 200 ng trypsin (Promega sequencing grade) overnight at 37 °C.

Sample amounts were normalised based on protein concentration measurements for TMT labelling. Approximately 5.6 μg of peptides in 50 μl 100 mM TEAB were labelled with 0.06 mg of a TMT10plex label for 1 h, then quenched with 5 μl 400 mM Tris-HCl for 15 min. All eight samples were combined, desalted on SOLA HRP SPE cartridges (Thermo Scientific) and dried down in a vacuum centrifuge. Samples were resuspended in 2% acetonitrile with 0.1% formic acid.

Samples were analysed on the Dionex Ultimate 3000/Orbitrap Fusion Lumos platform. Peptides were separated on a 50 cm, 75 μm ID EasySpray column (ES803; Thermo Fisher) on a 60-minute gradient of 2 to 35% acetonitrile (containing 0.1% formic acid and 5% DMSO) at a flow rate of 250 nl/min. Data were acquired using the MultiNotch MS3 method, as described previously (McAlister et al, 2014).

Mass spectrometry raw data were analysed in Proteome Discoverer 2.1. Proteins were identified with Sequest HT against the UPR Homo sapiens database (retrieved February 2017). Mass tolerances were set to 10 ppm for precursor and 0.5 Da fragment mass tolerance. TMT10plex (N-term, K), Oxidation (M) and Deamidation (N, Q) were set as dynamic modifications, alkylation (C) as a static modification. The mass spectrometry proteomics data have been deposited in the ProteomeXchange Consortium via the PRIDE (Vizcaíno et al, 2016) partner repository with the dataset identifier PXD053641.

### Quantification and statistical analysis

For comparing multiple experimental genotypes with the control, we applied the non-parametric Kruskal–Wallis test followed by Dunn's multiple comparisons post hoc test. When comparing two groups, a non-parametric Mann–Whitney test was used. These statistical analyses were performed on GraphPad Prism. All graphs displayed in the figures show the mean value for each genotype and include error bars representing the standard error of mean (SEM). Each 'n' in the analysis corresponds to the number of AGs (animals) analysed for each genotype (≥7), with the value for each gland calculated as the mean for three SCs. Each experiment was repeated at least three times. It was not straightforward to undertake blind analysis, because several of the phenotypes were immediately apparent, even by low-resolution microscopy.

For Western blots, relative signal intensities were analysed using the Kruskal–Wallis test.

## Data availability

The imaging and data analysis datasets produced in this study are provided in Source Data except for the movies, which are available at https://www.ebi.ac.uk/biostudies/bioimages/studies/S-BIAD1697.

The mass spectrometry proteomics data have been deposited in the ProteomeXchange Consortium via the PRIDE (Vizcaíno et al, 2016) partner repository at https://www.proteomexchange.org/ with the dataset identifier PXD053641.

The source data of this paper are collected in the following database record: biostudies:S-SCDT-10_1038-S44318-025-00497-y.

## Peer review information

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

## Acknowledgements

We are grateful to all the staff at the Micron Bioimaging Facility for their support, to Amy Cording for her technical assistance, and to Raphael Heilig and Benedikt Kessler for their support with the proteomics analysis. We thank Suzanne Eaton, Stephen Goodwin, Elodie Prince, Francois Karch, Linda Partridge, Tanja Godenschwege and Doris Kretzschmar, as well as the Bloomington and Vienna *Drosophila* Stock Centres for *Drosophila* stocks. We acknowledge the support of the BBSRC (BB/L007096/1, BB/N016300/1, BB/R004862/1, BB/W00707X/1 and BB/W015455/1), Cancer Research UK (C19591/A19076 and C602/A18974) and the National Science and Technology Council, Taiwan (111-2320-B-008-001-MY2 and 113-2320-B-008-003) to S-JF. PJS was the recipient of a grant from the Balliol Interdisciplinary Institute (BII), a subsidiary of Balliol College, Oxford. AW was supported by a

Krebs Memorial Scholarship from the Biochemical Society, and RF by Wellcome (097813/11/Z) and the John Fell Fund (133/075). For the purpose of Open Access, the author has applied a CC BY public copyright licence to any Author Accepted Manuscript (AAM) version arising from this submission.

## Author contributions

**Preman, J Singh**: Resources; Data curation; Formal analysis; Validation; Investigation; Methodology; Writing—review and editing. **Bhavna Verma**: Resources; Data curation; Formal analysis; Validation; Investigation; Visualisation; Methodology; Writing—review and editing. **Adam Wells**: Resources; Data curation; Formal analysis; Validation; Investigation; Visualisation; Methodology; Writing—review and editing. **Cláudia C Mendes**: Resources; Investigation; Methodology; Writing—review and editing. **Dali Dunn**: Investigation; Methodology; Writing—review and editing. **Ying-Ni Chen**: Data curation; Investigation; Visualisation; Methodology; Writing—review and editing. **Jade Oh**: Investigation; Methodology; Writing—review and editing. **Lewis Blincowe**: Resources; Investigation; Methodology; Writing—review and editing. **S Mark Wainwright**: Resources; Investigation; Methodology; Writing—review and editing. **Roman Fischer**: Conceptualisation; Data curation; Formal analysis; Supervision; Funding acquisition; Validation; Visualisation; Methodology; Project administration; Writing—review and editing. **Shih-Jung Fan**: Conceptualisation; Formal analysis; Supervision; Funding acquisition; Validation; Methodology; Project administration; Writing—review and editing. **Adrian L Harris**: Conceptualisation; Supervision; Funding acquisition; Methodology; Project administration; Writing—review and editing. **Deborah, C Deborah C I Goberdhan**: Conceptualisation; Supervision; Funding acquisition; Methodology; Writing—original draft; Project administration; Writing—review and editing. **Clive Wilson**: Conceptualisation; Supervision; Funding acquisition; Visualisation; Methodology; Writing—original draft; Project administration; Writing—review and editing.

Source data underlying the figure panels in this paper may have individual authorship assigned. Where available, figure panel/source data authorship is listed in the following database record: biostudies:S-SCDT-10_1038-S44318-025-00497-y.

## Disclosure and competing interests statement

The authors declare no competing interests.

# Expanded View Figures

**Figure EV1.   *Drosophila* MFAS selectively drives DCG assembly in SCs, related to Fig. 1.**

(A–C) Ex vivo, wide-field fluorescence micrographs and DIC images of SCs from 6-day-old males expressing SC-specific *rosy*-RNAi or either of two independent *mfas*-RNAis with the *GFP-mfas* gene trap (A), *YFP-Rab11* (B) or *CFP-Rab6* (C). Rab-positive ILVs (green arrowheads) in compartments are marked in Zoom panels. In *mfas* knockdown cells, sporadic intra-compartmental puncta detected by DIC, which are often Rab-positive, are marked with grey arrowheads. (D) Bar chart of DCG compartment number shows that expressing *rosy*-RNAi in adult SCs using the *tub-GAL80^{ts}; dsx-GAL4, GFP-mfas* SC-specific GAL4 driver line has no effect relative to controls. (E) Stills from a time-lapse movie of DCG biogenesis and DCG acidification in SC from a 6-day-old male expressing *GFP-mfas* gene trap. For the compartment marked by the white box (top row of zoomed images), a single DCG forms rapidly from a GFP-MFAS cloud. For compartment marked by white box and asterisk (bottom row), small LysoTracker Red-positive compartments (red arrowheads) contact and spread around the periphery of the DCG compartment, then start to acidify the lumen around the DCG (66 min) before rapid dispersion of the core (67.5 min; GFP and by DIC). Purple arrowhead marks DCG; it can persist for 20 min following the start of the acidification process. *td-GFP-mfas = tub-GAL80^{ts}/+; dsx-GAL4, GFP-mfas/+*. (F, G) Knockdown of *mfas* with two independent RNAis has no effect on the number of Rab6-positive large compartments (F) or the proportion of these compartments containing Rab6-positive ILVs (G). (H) Knockdown of *mfas* in SCs expressing GFP-GPI, a DCG and a membrane marker, produces compartments with GFP at the limiting membrane, like controls and with internal concentrations of GFP in puncta that are also detected by DIC (grey arrowhead). (I) Bar charts showing proportion of large compartments with DCG acidification phenotype (E) in GFP-MFAS-expressing knockdown SCs. In all images, approximate cell boundaries and compartment boundaries are marked with a dashed white line; *n* = nuclei of binucleate cells; LysoTracker Red (magenta) marks acidic compartments. Scale bars = 5 and 1 μm in Zoom. For bar charts, data are mean ± SEM, analysed using the Kruskal–Wallis test, followed by Dunn's multiple comparisons post hoc test; *n* = animal number above bar, ***$P < 0.001$, ns not significant. In (I), *rosy* vs *mfas* #1, $P = 0.0010$; *rosy* vs *mfas* #2, $P = 0.0009$.

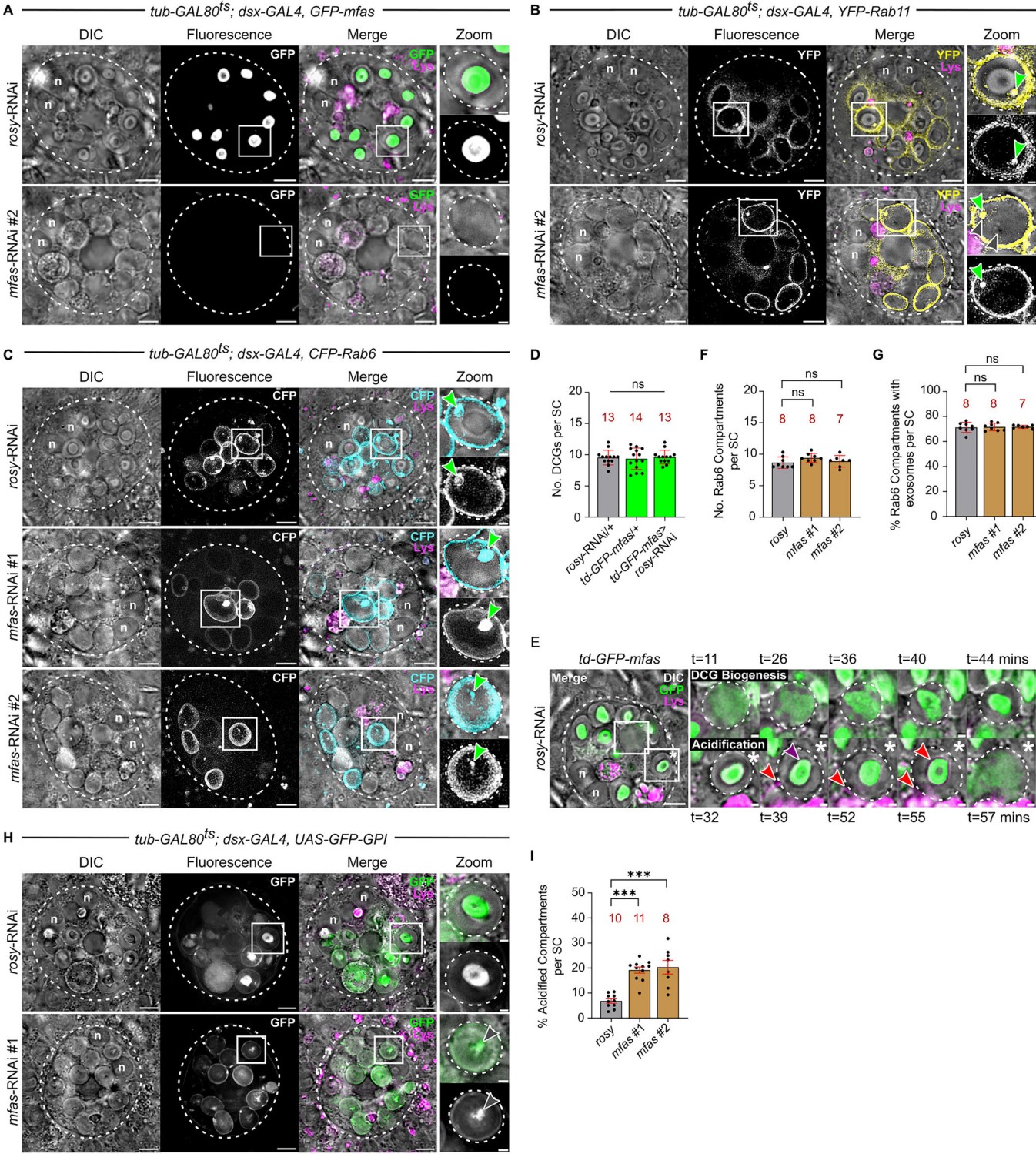

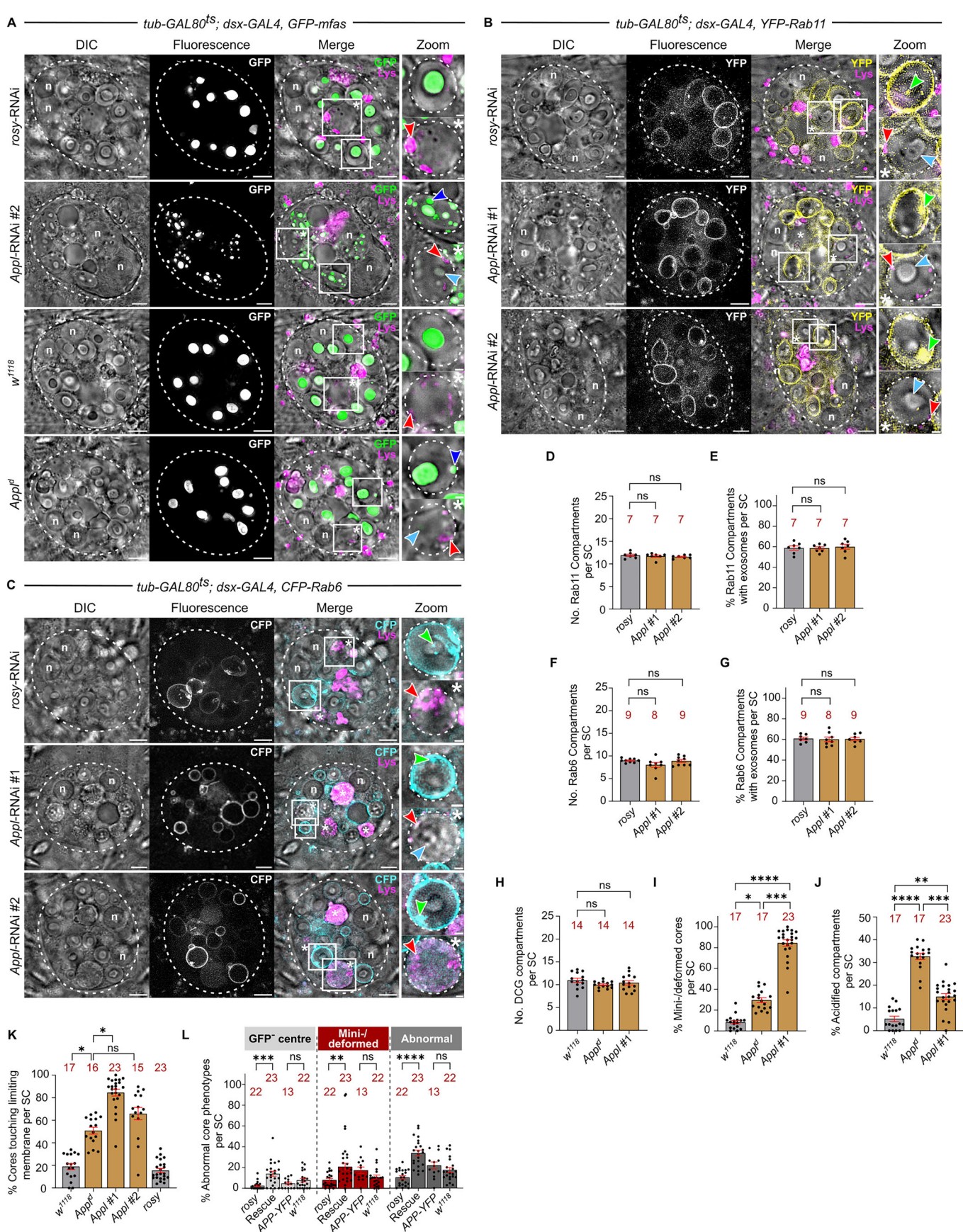

◀

**Figure EV2. *Drosophila* APPL regulates formation of large DCGs in SCs, related to Fig. 3.**

(A) SCs expressing *GFP-mfas* gene trap and SC-specific *rosy*-RNAi or *Appl*-RNAi #2 (top two rows), or without additional transgenes, but in a control *w^1118* or *Appl^d* mutant background (bottom two rows). Note that following knockdown of *Appl*, large secretory compartments contain multiple mini-cores (blue arrowheads), labelled with GFP-MFAS and visible by DIC (white box and upper Zoom panel). *Appl^d* mutant SCs contain distorted DCGs that frequently contact the compartment's limiting membrane and sporadic mini-cores (white box, top Zoom panel). The DCG acidification phenotype is also more commonly observed in *Appl* knockdown and *Appl* mutant backgrounds (white asterisks and white boxes marked by asterisks, shown in lower Zoom panels; red arrowheads mark acidic microdomains). DCGs that have not yet been dissipated are marked with light blue arrowheads. (B, C) SCs expressing SC-specific *rosy*-RNAi or either of two independent *Appl*-RNAis with *YFP-Rab11* (B) or *CFP-Rab6* (C). Zoom panels show mini-core phenotype (white box, top panel) with intra-compartmental Rab puncta (green arrowheads), and DCG acidification phenotype with no Rab association (white box marked with an asterisk, bottom panel). (D–G) Bar charts showing numbers of Rab11-positive (D) or Rab6-positive (F) large compartments, and proportion of these compartments containing Rab11-positive (E) or Rab6-positive (G) ILVs in SCs with knockdown of *Appl* using two independent RNAis versus controls. (H–K) Bar charts comparing the effects of the *Appl^d* null mutant with SC-specific knockdown of *Appl* and controls. *Appl^d* typically does not affect DCG compartment number (H) or induce mini-core formation, but DCGs are often deformed (I) and touch the compartment's limiting membrane (K). The DCG acidification phenotype is particularly prominent in *Appl^d* SCs (J). (L) Bar chart showing DCG phenotypes induced by APP-YFP expression with and without *Appl* knockdown versus control SCs. Note that in *Appl* knockdown cells rescued by APP-YFP, about 10% of DCGs are mis-assembled, lacking GFP-MFAS at their centre (GFP-centre), perhaps because high-efficiency APP-YFP cleavage is required to prime the aggregation of proteins at the centre of all compartments in this genetic background. In all images, n nuclei; LysoTracker Red (magenta) marks acidic compartments. Scale bars = 5 and 1 μm in Zoom. For bar charts, data are mean ± SEM, analysed using the Kruskal–Wallis test; n = animal number above bar, *$P < 0.05$, **$P < 0.01$, ***$P < 0.001$, ****$P < 0.0001$, ns not significant. In (I), *w^1118* vs *Appl^d*, $P = 0.014$; *Appl^d* vs *Appl* #1, $P = 0.0004$. In (J), *w^1118* vs *Appl* #1, $P = 0.0070$; *Appl^d* vs *Appl* #1, $P = 0.0002$. In (K), *w^1118* vs *Appl^d*, $P = 0.032$; *Appl^d* vs *Appl* #1, $P = 0.024$. In (L), *rosy* vs Rescue (GFP-centre), $P = 0.0001$; *rosy* vs Rescue (Mini-/deformed), $P = 0.0058$.

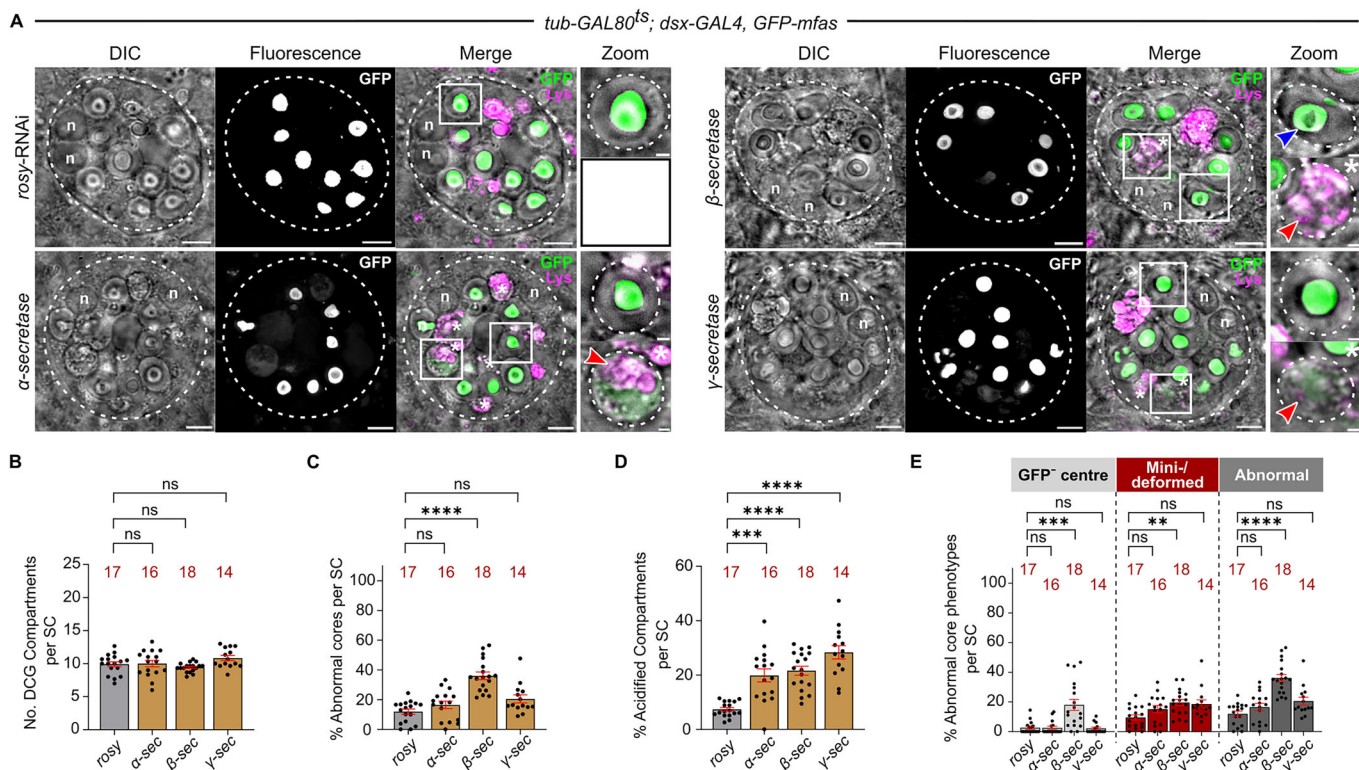

Figure EV3. Secretases involved in *Drosophila* APPL processing regulate DCG maturation, related to Fig. 5.

(A) SCs expressing *GFP-mfas* gene trap and SC-specific *rosy*-RNAi or RNAi targeting *α-, β-* and *γ-secretases*. Note β-secretase knockdown affects DCG morphology, so that GFP-MFAS is frequently absent from a large central region within the DCG, while other secretase knockdowns rarely affect DCG morphology (shown for compartments outlined with white boxes in upper Zoom). White asterisks and white boxes marked with asterisks in the Merge channel indicate DCG compartments with acidification phenotype (lower Zoom; red arrowheads mark acidic microdomains). (B–E) Bar charts showing effects of knockdowns on the number of DCG compartments (B), and proportion of abnormal DCGs (C). A greater proportion of large compartments display the DCG acidification phenotype following secretase knockdown than in controls (D). The different DCG phenotypes observed following knockdown of *α-, β-* and *γ-secretases* are categorised in (E). Note that *β-secretase* knockdown induces the formation of DCGs that lack GFP-MFAS at their centre. In all images, n nuclei; LysoTracker Red (magenta) marks acidic compartments. Scale bars = 5 and 1 μm in Zoom. For bar charts, data are mean ± SEM, analysed using the Kruskal–Wallis test; n = animal number above bar, **P < 0.01, ***P < 0.001, ****P < 0.0001, ns not significant. In (D), *rosy* vs *α-sec*, P = 0.0009. In (E), *rosy* vs *β-sec* (GFP-centre), P = 0.0008; *rosy* vs *β-sec* (Mini-/deformed), P = 0.0048.

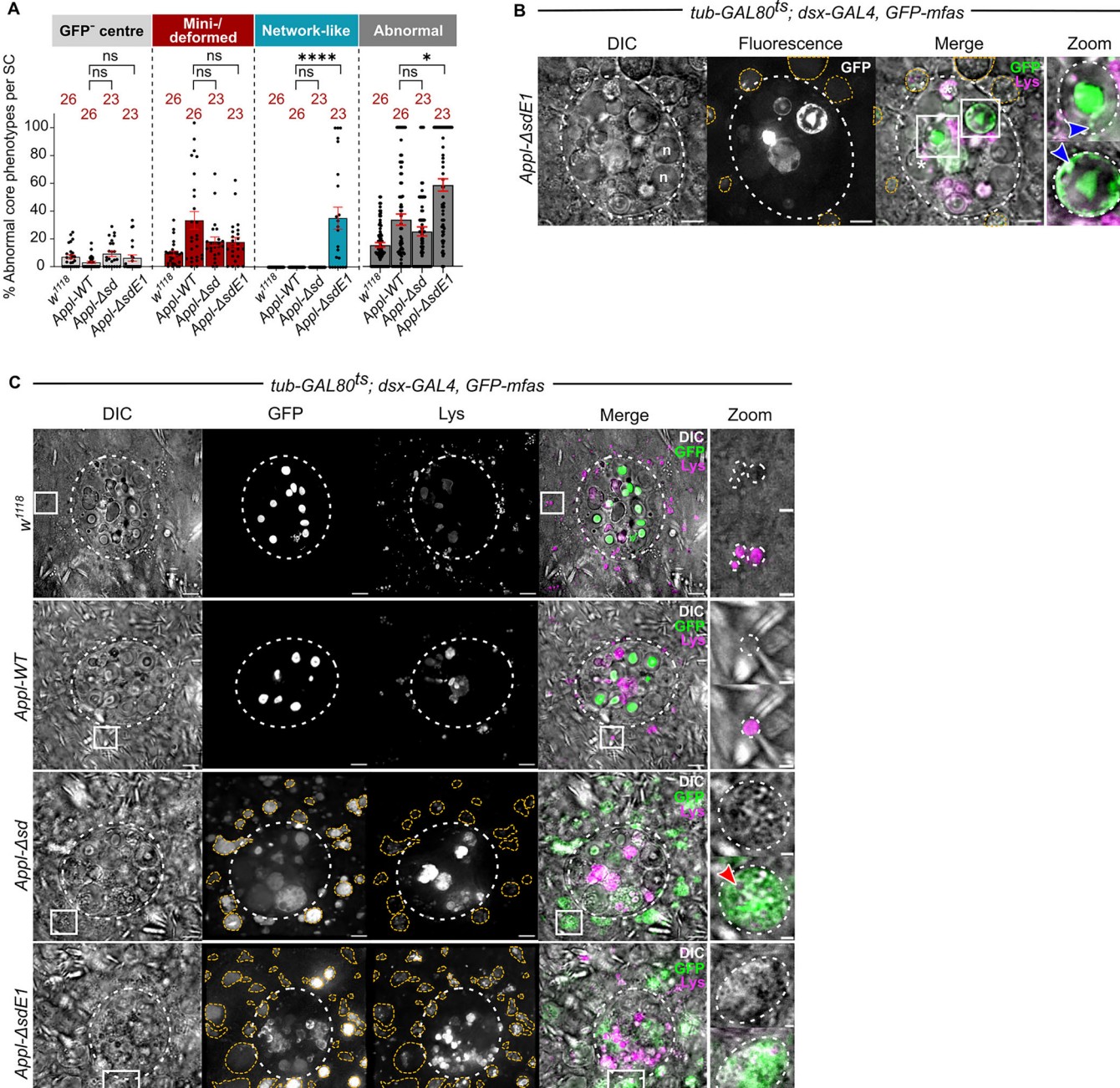

**Figure EV4. *Drosophila* APPL and its cleavage regulate normal DCG formation, and the uptake of GFP-MFAS by other cells, related to Fig. 5.**

(A) Bar chart showing that overexpression of APPL-WT, APPL-Δsd and APPL-ΔsdE1 produces abnormal DCGs, with APPL-ΔsdE1 generating a unique network phenotype. (B) SCs expressing *GFP-mfas* gene trap and APPL-ΔsdE1 (Fig. 4B). Note two abnormal DCG compartments that have a central abnormally shaped DCG, but also contain peripheral GFP-MFAS aggregates (blue arrowheads in compartments outlined with white boxes shown in Zoom panels). (C) SCs and surrounding main cells expressing *GFP-mfas* gene trap alone or with wild-type APPL (APPL-WT), APPL-Δsd or APPL-ΔsdE1 in SCs. Note for mutant APPL expression, abnormal accumulation of GFP-MFAS in main cell compartments that typically exhibit limited LysoTracker Red staining (compartments outlined by orange dashed lines); one example is outlined by a white box and shown in Zoom panels (DIC alone and DIC/Merge; red arrowheads mark acidic microdomains). In all images, n = nuclei; LysoTracker Red (magenta) marks acidic compartments in (B, C). Scale bars = 5 and 1 μm in Zoom. For bar charts, data are mean ± SEM, analysed using the Kruskal–Wallis test; n = animal number above bar, *P < 0.05, ****P < 0.0001, ns not significant. In (A), *Appl-WT* vs *Appl-ΔsdE1* (Abnormal), P = 0.015.

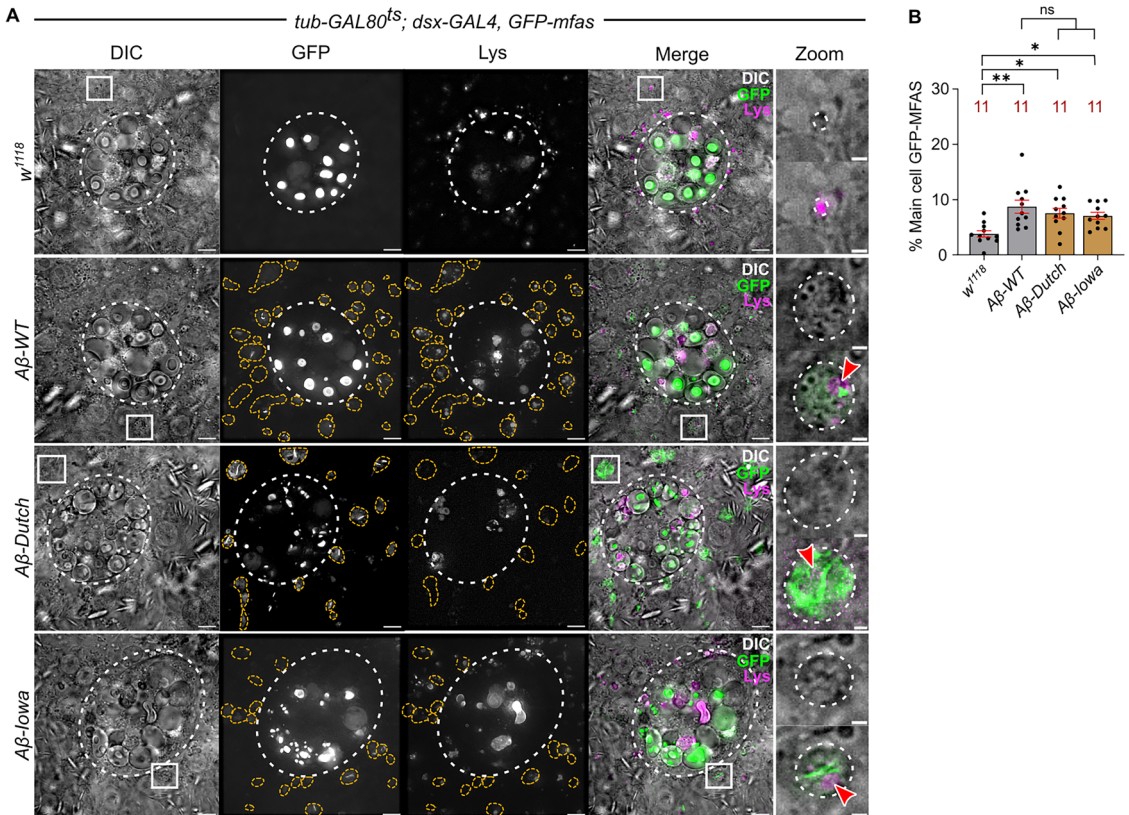

**Figure EV5. Aβ-42-peptide expression in SCs promotes uptake of GFP-MFAS by other accessory gland cells, related to Fig. 6.**

(A) SCs and surrounding main cells expressing *GFP-mfas* gene trap alone or with wild-type Aβ-42-peptide, or either the Iowa or Dutch mutant Aβ-42 peptides. Note for Aβ-42-peptide expression, abnormal accumulation of GFP-MFAS in main cell compartments that typically exhibit limited LysoTracker Red staining (compartments outlined by orange dashed lines); one example is outlined by a white box and shown in Zoom (DIC alone and DIC/Merge; red arrowheads mark acidic microdomains). (B) Bar chart showing the accumulation of GFP-MFAS in main cells of 6-day-old males overexpressing SC-specific wild type Aβ-42-peptide, or either the Iowa or Dutch mutant Aβ-42 peptides or *w1118* controls, expressed as percentage of total main cell area that contains GFP. In all images, n nuclei; LysoTracker Red (magenta) marks acidic compartments in (A). Scale bars = 5 and 1 μm in Zoom. For the bar chart, data are mean ± SEM, analysed using the Kruskal–Wallis test; *n* = animal number above bar, *P < 0.05, **P < 0.01, ns = not significant. In (B), *w1118* vs *Aβ-WT*, P = 0.0029; *w1118* vs *Aβ-Dutch*, P = 0.011; *w1118* vs *Aβ-Iowa*, P = 0.048.

