## [Peer Review File · The EMBO Journal]

Amyloid- β disrupts APP-regulated protein aggregation and dissociation from recycling endosomal membranes

Preman Singh, Bhavna Verma, Adam Wells, Claudia Mendes, Dali Dunn, Ying-Ni Chen, Jade Oh, Lewis Blincowe, Mark Wainwright, Roman Fischer, Shih-Jung Fan, Adrian Harris, Deborah Goberdhan, and Clive Wilson

Corresponding author(s): Clive Wilson (clive.wilson@dpag.ox.ac.uk)

Review Timeline:

Submission Date:	7th Aug 24
Editorial Decision:	9th Aug 24
Appeal Received:	21st Aug 24
Editorial Decision:	8th Nov 24
Revision Received:	12th Mar 25
Editorial Decision:	27th Apr 25
Revision Received:	2nd May 25
Accepted:	30th May 25

Editor: Ioannis Papaioannou

Transaction Report:

Dear Prof. Wilson,

Thank you for submitting your manuscript (EMBOJ-2024-118705) to The EMBO Journal for our consideration. I have now read it carefully and discussed it with the other members of our editorial team. In addition, I have consulted an external expert in this field with familiarity with our journal and its scope. I regret to say that our conclusion was that we cannot offer publication in The EMBO Journal.

We acknowledge that in this study you provide evidence indicating that biogenesis of dense-core granules (DCGs) in the large secretory compartments of *Drosophila* male accessory gland secondary cells (SCs) is regulated by the *Drosophila* homolog of TGFBI and APPL. Your study suggests that proteolytic cleavage of APPL permits aggregates to coalesce and mature into a large central DCG. Furthermore, your results suggest that expression of pathological human Abeta peptides disrupts this maturation process and inhibits compartment motility. We recognize that these findings will be of interest to those working in this field.

While discussing your manuscript in our team, however, a critical concern that was raised was whether proteolytic processing of APPL by any of the three secretases is definitively shown in the study. Furthermore, and considering the particularly broad readership of The EMBO Journal, we were not sure if the advance provided by the study warrants further consideration here. We therefore consulted an external advisor regarding the potential suitability of the manuscript for The EMBO Journal. Our advisor was not particularly supportive, highlighting also the fact that in their view additional data would indeed be required to definitively prove the proteolytic processing of APPL by any of the three secretases. One additional note they made -that nevertheless might be a minor matter of presentation- is that in the schematic Abeta appears to have been expressed without a signal peptide, which -if correct- would raise questions regarding its presence within the secretory pathway.

Taking into account our editorial concerns as well as the critical feedback of our external advisor, I regret to say that we think this study might be better-suited for a journal with a more specialized readership than ours, and we have therefore decided not to proceed with in-depth peer review.

I am sorry to have to disappoint you on this occasion, but I would like to thank you for your interest in our journal and the opportunity to consider your manuscript. I wish you every success in publishing these results in a more suitable journal.

Yours sincerely,

** As a service to authors, EMBO Press provides authors with the possibility to transfer a manuscript that one journal cannot offer to publish to another EMBO publication or the open access journal Life Science Alliance launched in partnership between EMBO Press, Rockefeller University Press and Cold Spring Harbor Laboratory Press. The full manuscript and if applicable, reviewers' reports, are automatically sent to the receiving journal to allow for fast handling and a prompt decision on your manuscript. For more details of this service, and to transfer your manuscript please click on Link Not Available. **

Department of Physiology, Anatomy and Genetics

Sherrington Building
University of Oxford
Parks Road
Oxford OX1 3PT

Direct Line: (01865) 282662
Laboratory: (01865) 282661
Switchboard: (01865) 272169

Fax: (01865) 272420

Professor Clive Wilson

*Professor of Cell and Developmental Genetics
Tutorial Fellow in Medicine - St. Hugh's College*

e-mail: clive.wilson@dpag.ox.ac.uk

20 August 2024

Ioannis Papaioannou, PhD
Editor, The EMBO Journal

Dear Dr Papaioannou

Re: EMBOJ-2024-118705

Thank you for the significant time and effort that you have put into considering our manuscript, involving both your editorial team and an external advisor. Having discussed your email, we are concerned about the rationale for rejecting our manuscript and are therefore writing to appeal this decision.

From your email, our understanding is that your editorial team recognised the potential interest of our findings that APPL plays a cleavage-dependent role in DCG protein aggregation, a process that is disrupted by human A β -peptide expression. However, for this to be of sufficient interest to your readership, you thought it was essential to demonstrate that one or more of the three AD-relevant secretases mediated the cleavage. We agree that we have not demonstrated this directly, but strongly believe that this should not be a rationale to reject the manuscript for the following reasons:

1. *Cleavage by α -, β - or γ -secretase is not a prerequisite in our APPL-dependent aggregation/dissociation model.*

The key features of our model are that APPL's ECD normally functions in the control of membrane-dependent DCG protein aggregation and that it is then released by cleavage from membranes and secreted. It is important that the ECD is cleaved close to APPL's transmembrane domain and we have provided evidence for this (see below). α - and/or β -secretase cleavage is likely to be involved, but this is not an essential part of the model.

2. *A short sequence in APPL containing the α - and β -secretase cleavage sites is required for membrane/aggregate separation.*

We have demonstrated that the proteolytic release of a GFP-tagged APPL-ECD coincides with formation of the central DCG (**Fig. 4C**), using a transgenic that has previously been employed to monitor the processing of APPL by α - and β -secretase (ref. [47]). We have also shown that APPL- Δ sd mutant proteins, which lack the short sequence where the α - and β -secretases cut, do not support normal DCG biogenesis (**Fig. 5**). These mutants have been widely used to provide evidence for α - and/or β -secretase's role in APPL function, eg. refs. [19, 20], and Rieche F, et al. (2018) Curr Biol. 28:817-823.e3 ; Bolkan BJ, et al. (2012) J Neurosci. 32:16181-92 ; Wentzell JS, et al. (2012) Neurobiol Dis. 46:78-87. The ECD in these mutants is non-cleavable and not secreted at detectable levels in neurons; in fact, this defect is not neuron-specific, because while overexpressed full-length APPL is cleaved even in non-neuronal S2 cells, which do not normally produce APPL, APPL- Δ sd's ECD is not released (refs. [19, 48]). Of particular note in our experiments, we find that APPL- Δ sd with an E1 domain deletion promotes protein aggregation in a network at the periphery of compartments (**Fig. 5A**). Taken together, these data strongly support APPL's role in protein aggregation, but also the critical requirement for it to be cleaved within the sequence cut by α - and

β -secretases to form a central DCG.

3. *DCG formation is disrupted by secretase knockdown.*

We observe defects in DCG biogenesis, particularly following knockdown of β -secretase (**Figs. S4 and S5A**). However, as we explain in the manuscript, such experiments have limited value because these secretases have other targets, which may also be involved in DCG biogenesis. Our central approach, phenotypic analysis of cells expressing secretion-defective mutants lacking α - and β -secretase cleavage sites (**Point 2**) and of cells with secretase knockdown (or mutation), has been used by several other groups to assess the roles of APPL and secretase cleavage in the fly brain, eg. refs [19, 20], Rieche F, et al. (2018) Curr Biol. 28:817-823.e3 ; Bolkan BJ, et al. (2012) J Neurosci. 32:16181-92.

4. *We cannot use western blotting to demonstrate APPL cleavage in our model.*

In *Drosophila* brains, altered cleavage of APPL involving the α - and β -secretases has been distinguished in some studies using western analysis, either after overexpressing specific secretases (ref. [17]) or in heterozygous secretase mutant backgrounds (Rieche F, et al. (2018) Curr Biol. 28:817-823.e3). At least five or more heads are required for this analysis per sample. In contrast to the fly brain with about 100,000 neurons expressing APPL, our accessory gland model has only 80 secondary cells per male, which express much lower levels of APPL. Even if we overexpressed APPL in secondary cells, which will likely artificially induce changes in the balance of different protease cleavage events, it would not be realistic to dissect the number of glands required to detect an APPL signal. Our *β -secretase* knockdown experiment suggests this enzyme is expressed by secondary cells and is therefore likely to cut APPL, but we cannot exclude that another protease, which cuts in the short sequence deleted in the APPL- Δ sdE1 mutant, is involved in APPL cleavage during DCG compartment maturation.

5. *The A β -peptides expressed in secondary cells have an ER signal sequence.*

Regarding the additional point from the external advisor, our schematics show mature proteins and peptides after the signal sequence has been removed, which we can highlight in a revised version of the text. These A β -peptide transgenics were generated and previously employed by others, eg. refs [55,60]. They include a signal sequence to permit secretion; such constructs have been used extensively to induce neurodegenerative phenotypes in flies.

6. *New evidence reveals that the A β -Dutch mutant blocks APPL cleavage.*

We think one of the key issues that might concern reviewers and your readership is whether the mutant A β -peptides actually affect the normal regulation of APPL, not just the process it controls. Over recent months, we have been generating new fly lines carrying multiple transgenes to study the A β phenotypes in more detail. We have now found that in cells expressing A β -Dutch, cleavage of the double-tagged APPL protein is suppressed in DCG compartments and its aggregation-promoting properties are altered, explaining the observed mini-core phenotype and supporting our proposal that the APPL cleavage event is disease-relevant. We are currently repeating these experiments for a final time to confirm this finding.

In summary, we have for the first time characterised a cell type in which the sub-compartmental dynamics of regulated secretion and APPL processing can be followed as DCGs mature. This has revealed previously undescribed cellular functions for APPL, which can be substituted for by human APP, and are disrupted by mutant A β -peptide expression, leading to early AD-related cellular phenotypes. We believe that our findings provide a new way to conceptualise the previously reported properties of APP and the defects observed in AD within a unified sub-compartmental model. Our work suggests that a secretase, probably β -secretase, drives the critical APPL cleavage event, but we cannot eliminate the possibility that another protease is also involved. In a field where most researchers focus on neurons and the brain, we have turned to a different and much less abundant cell type. While this system has its limitations, most notably our inability to precisely characterise APPL's biochemical processing in these cells, it has provided insights into dynamic APPL-mediated mechanisms that have not been possible to identify and study in any other cells.

In our submitted manuscript, we had decided not to consider the role of secretases in detail in the Discussion or to highlight this aspect in the Abstract. In our cover letter, however, we inadvertently mentioned the role of 'secretases' in our model, rather than 'the secretase cleavage region', and we apologise for any confusion that this caused. Based on your comments, we think it would be helpful to clarify the points above by a brief addition to the Discussion, as well as include the new data showing the effect of A β -Dutch on APPL cleavage.

In light of the arguments above, we ask that you reconsider your decision and allow a slightly revised version of the manuscript to be reviewed for potential publication in the *EMBO Journal*. If it would be helpful to clarify any of the points above, I would be happy to discuss them with you.

Thank you for considering our request. We look forward to hearing from you.

With best wishes

Professor Clive Wilson

Dear Clive,

Thank you again for the submission of your manuscript (EMBOJ-2024-118705R-Q) to The EMBO Journal for our consideration. As I have already informed you, your manuscript has been seen by four experts in the field, and we have received the full set of their detailed reports (included below), which I have already shared with you for your information. I would like to thank you for your thorough and informative point-by-point response to the referees' comments as well as your provisional revision plan, which were very helpful for us to reach a fair and balanced decision on the manuscript.

Although the referees generally agree that the findings are potentially interesting and significant, their input on whether the conclusions are fully supported by the available data is mixed, and while referee #1 is very supportive of the manuscript, a number of major technical and other concerns were raised by referees #2-#4. These three referees point out that some of the findings seem to have been overinterpreted, and not all referees agree on their relevance.

We acknowledge that in your detailed point-by-point response to the referees' reports, you provide further explanations to some of their comments, and you describe a provisional major revision plan including substantial new experimental work, re-analysis of the already available data, and re-writing of some sections of the manuscript. Overall, you express your willingness to address the referees' concerns and prepare a strengthened version of your manuscript that will be putting your work and its conclusions in a broader context. In light of the referees' reports and your thorough responses to them, as well as your provisional revision plan, which we find likely sufficient to address the referees' concerns, I would like to invite you to submit a revised version of your manuscript along the lines you describe in your revision plan. I would like to kindly ask you to carefully consider all possible interpretations of the results, taking the comments of the referees on board, and discussing them in detail and in a balanced manner in your revised manuscript avoiding any interpretation that is not fully supported by the available data. Importantly, the revised manuscript should be clarifying the relevance and significance of the findings in the context of the previously (and recently) published literature, ensuring that the work will be accessible by our broad readership and not only by field specialists.

Please also submit a detailed point-by-point response to the referees' comments along with your revised manuscript, describing all additions and changes to the manuscript. I should add that it is The EMBO Journal policy to allow only a single round of major experimental revision. Considering the mixed input we received from the referees, I would like to explain that the outcome of the re-review process cannot be guaranteed and depends on the completeness of your responses in the revised version, and the level of support we will receive from the referees upon re-review.

We generally allow three months as standard revision time (February 7th, 2025), which according to your revision plan would be realistic for the planned experiments, but we may be able to grant an extension to allow enough time for the revision should the need arise. Should you foresee a problem in meeting the three-month deadline, please let us know. As a matter of policy, competing manuscripts published during this period will not negatively impact our assessment of the conceptual advance presented by your study. However, we request that you contact us as soon as possible upon publication of any related work, to discuss how to proceed.

Please let me know if you have any questions or comments that you would like to discuss further with me.

Thank you again for the opportunity to consider your work for publication in The EMBO Journal. I am looking forward to your revision.

Best wishes,

Ioannis

Instructions for preparing your revised manuscript

1. When you are ready to submit the revision, please upload:

- A Word file of the manuscript text (including legends of main Figures, EV Figures and Tables). Please make sure that changes

are highlighted (or "tracked") to be clearly visible.

- Individual production-quality figure files (one file per figure). When assembling your figures, please refer to our figure preparation guidelines in order to ensure proper formatting and readability in print as well as on screen:

If the data shown in a figure are obtained from n {less than or equal to} 2, please use scatter plots showing the individual data points.

- i. the name of the statistical test used to generate error bars and P values
- ii. the number (n) of independent experiments (please specify technical or biological replicates) underlying each data point (discussion of statistical methodology can be reported in the Materials and Methods section, but figure legends should contain a basic description of n , P , and the test applied)
- iii. the nature of the bars and error bars (s.d., s.e.m.).

- A point-by-point response to the referees' comments, with a detailed description of the changes made (as a word file). All referees' concerns must be fully addressed and their suggestions taken on board. When preparing your letter of response to the referees' comments, please bear in mind that this will form part of the Review Process File and will therefore be available online to the community. Please note that you have the possibility to opt out of the transparent process at any stage prior to publication by letting the editorial office know (contact@embojournal.org); if you do opt out, the Review Process File link will point to the following statement: "No Peer Review File is available with this article, as the authors have chosen not to make the review process public in this case.". For more details on our Transparent Editorial Process, please visit our website:

<https://www.embopress.org/page/journal/14602075/authorguide#transparentprocess>

- Expanded View (EV) files (replacing Supplementary Information) that are collapsible/expandable online. A maximum of 5 EV Figures can be typeset. EV Figures should be cited as "Figure EV1, Figure EV2" etc. in the text, and their respective legends should be included in the manuscript file after the legends of regular figures. See detailed instructions regarding Expanded View files here:

- For the figures that you do NOT wish to display as Expanded View figures, they should be bundled together with their legends in a single PDF file called "Appendix", which should start with a short Table of Contents (including page numbers). Appendix figures should be referred to in the main text as: "Appendix Figure S1, Appendix Figure S2" etc. Please see detailed instructions here: <https://www.embopress.org/page/journal/14602075/authorguide#expandedview>

- A complete author checklist, which you can download from our author guidelines (<https://www.embopress.org/page/journal/14602075/authorguide>). Please note that the checklist will also be part of the Review Process File.

2. Please note that no statistics should be calculated and shown in Figures if $n=2$. Please also note that each p value should be reported as an exact value.

3. Before submitting your revision, primary datasets (and computer code, where appropriate) produced in this study need to be deposited in appropriate public databases (see <https://www.embopress.org/page/journal/14602075/authorguide#dataavailability>). In particular, we kindly ask you to deposit the mass spectrometry datasets produced in your study. Their accession numbers, databases, and the specific URLs (links) should be listed in a formal "Data availability" section (placed after Methods), following the example:

"The datasets (and computer code, if applicable) produced in this study are available in the following databases:

RNA-Seq data: Gene Expression Omnibus GSE46843 (<https://www.ncbi.nlm.nih.gov/geo/query/acc.cgi?acc=GSE46843>)

Chip-Seq data: Gene Expression Omnibus GSE46748 (<https://www.ncbi.nlm.nih.gov/geo/query/acc.cgi?acc=GSE46748>)".

*** The Data Availability Section is restricted to new primary data that are part of this study. In case you have no data that require deposition in a public database, please state so instead of referring to the database: "Our study includes no data deposited in public repositories." under the heading "Data availability". ***

*** All links should resolve to a page where the data can be accessed. ***

*** Please remember to provide in the Data availability section of your revised manuscript reviewer passwords if the datasets are not yet public. ***

*** Please use detailed data citations for already available datasets that were re-analyzed in your study - for more information on the format, see point #9 below. ***

4. Please check that the title and the abstract of the manuscript are brief, yet explicit, even to non-specialists. The length of the title should not exceed 100 characters, and the abstract should be a single paragraph not exceeding 175 words.
5. All materials and methods need to be described in the manuscript using our "Structured Methods" format, which is now required for all research articles. According to this format, the Methods section includes a single "Reagents and Tools Table" - listing key reagents, experimental models, software and relevant equipment including their sources and relevant identifiers- followed by a "Methods and Protocols" section describing the methods. Please download and fill our Reagents and Tools Table template (.docx), which you can find in our author guide: <https://www.embopress.org/page/journal/14602075/authorguide#structuredmethods>. When submitting your revised manuscript, please do not include the Reagents and Tools Table in the Methods section of the manuscript but upload it as a separate file choosing the file type "Reagent Table".
6. Please also note our reference format: <https://www.embopress.org/page/journal/14602075/authorguide#referencesformat>.
7. At EMBO Press we ask authors to provide source data for the main manuscript figures. Our source data coordinator will contact you to discuss which figure panels we would need source data for and will also provide you with helpful tips on how to upload and organize the files.
8. Please remember: digital image enhancement is acceptable practice, as long as it accurately represents the original data and conforms to community standards. If a figure has been subjected to significant electronic manipulation, this must be noted in the figure legend or in the "Materials and Methods" section. The editors reserve the right to request original versions of figures and the original images that were used to assemble the figure.
9. Our journal encourages inclusion of data citations in the reference list to directly cite datasets that were obtained from public databases. Data citations in the article text are distinct from normal bibliographical citations and should directly link to the database records from which the data can be accessed. In the main text, data citations are formatted as follows: "Data ref: Smith et al, 2001" or "Data ref: NCBI Sequence Read Archive PRJNA342805, 2017". In the Reference list, data citations must be labeled with "[DATASET]". A data reference must provide the database name, accession number/identifiers, and a resolvable link to the landing page from which the data can be accessed at the end of the reference. Further instructions are available at: <https://www.embopress.org/page/journal/14602075/authorguide#referencesformat>.
10. We request authors to consider both actual and perceived competing interests. Please review our policy (<https://www.embopress.org/page/journal/14602075/authorguide#conflictsofinterest>) and update your competing interests statement if necessary. Please name this section 'Disclosure and competing interests statement' and place it after the Acknowledgements section.
11. Please note that all corresponding authors are required to provide an ORCID ID upon submission of a revised manuscript (<https://orcid.org/>). Please find instructions on how to link your ORCID ID to your account in our manuscript tracking system in our Author guidelines (<https://www.embopress.org/page/journal/14602075/authorguide#authorshipguidelines>).
12. We use CRediT to specify the contributions of each author in the journal submission system. CRediT replaces the author contribution section, which should be removed from the manuscript. Please use the free text box to provide more detailed descriptions. See also guide to authors: <https://www.embopress.org/page/journal/14602075/authorguide#authorshipguidelines>.
13. Further information is available in our Guide For Authors: <https://www.embopress.org/page/journal/14602075/authorguide>
14. We would also welcome the submission of cover suggestions or motifs to be used by our Graphics Illustrator in designing a cover.
15. Please use the link below to submit your revision:
<https://emboj.msubmit.net/cgi-bin/main.plex>

Referee #1:

Singh and Verma et al. have presented data studying the process of dense-core granule biosynthesis and secretion using a

Drosophila prostate-like secondary cell model system. They identify a GFP fly line (mfas) that has a GFP fusion to the gene encoding the protein TGFBI. The protein product is localised to DCGs within secondary cells. Depleting the expression of mfas via RNAi caused a defect in DCG formation. They use the MFAS-GFP line to image DCG biogenesis and observe an early phase of the fusion of smaller granules they term mini-cores. This early phase phenocopied a previously observed phenotype on a mutant in the gene encoding the protein GAPDH2. This leads the authors to deduce a correlative link between DCG, secretory proteins, and Alzheimer's disease-associated machinery and biomarkers. The authors then test the APP fly homologue APPL, which revealed defective DCG biogenesis, which could be at least partially complemented using human APP. Using a dual-tagged APPL, they could demonstrate that DCG biogenesis is correlated with APPL cleavage events. Knock-down using RNAi of APPL secretases also resulted in a DCG phenotype, as did overexpression of APPL cleavage mutants. Overexpression of pathogenic ABeta peptides also resulted in defective DCG stalled at the mini-core stage. Overall, the manuscript is well written, and the experiments well performed. Conclusions are reasonable and balanced and there are a number of important and interesting findings. I have a number of minor comments below, but on the whole, I am very supportive of this manuscript, which I think makes a valuable contribution to the field, and if the observations hold true in dense-core vesicle biogenesis in human neurons- will fundamentally change our understanding of APP cell biology.

Minor comments:

The resolution of the figures is extremely poor, some text was hard to read.

On line 129, the authors state: "Knockdown of mfas with two independent RNAis specifically in adult SCs produced large secretory compartments with no DCGs (Figures 1C, 1E and S1A)". I agree with the interpretation but i think a little more accuracy is needed here as it is so important for the manuscript. I suggest something like "produced large secretory compartments with no observable DCGs in the brightfield indicating a lack of protein condensation" or something the authors are comfortable with.

The manuscript is heavily dependent on microscopy phenotypes. I have no major issue with this, however in this case the quantification becomes absolutely essential and effectively is the data. However, the manuscript is somewhat lacking here. For quantification and statistical analysis, each "n" should equal an independent experimental repeat (as that is the variability that is important). However, I think each cell may resemble the other. Some of the n values are listed as more than 60, and it would strengthen the manuscript if the number of independent experimental repeats is clarified and possibly increased.

It was unclear to me what exactly was overexpressed in figure 6 (the abeta peptides and mutants). In the text, it says, "wild-type form of the pathological A β -42 peptide". However, the peptide is extracellular- so I presume it was signal peptide-Abeta peptide (or mutants). I would very much appreciate it if this was clarified. If these Abeta peptides are being expressed cytosolically, then i would suggest this is a misleading reagent and I would recommend this data be removed from the manuscript.

Referee #2:

This manuscript focuses on the role of MFAS and APP in the DCG maturation processes in health and disease. More specifically, the authors demonstrated that APP is essential for protein aggregation events and hence for DCG biogenesis and trafficking. When APP is mutated or its cleavage is perturbed, or its expression is suppressed, DCG maturation is inhibited resulting in deformed compartments, trafficking defects and lysosomal degradation.

This work elucidated the physiological role of APP that could enhance our understanding on early stage AD. Additionally, the maturation of DCGs is addressed, a process poorly studied so far. Although the findings are considered of general significance, the conclusions of the paper are not entirely justified by the presented data.

Main remarks

1. Drosophila TGFBI drives DCG assembly in SCs

It is not clear why MFAS was investigated as a potential DCG protein. Other known DCG proteins should be used to show DCG morphology. To further show the selectivity of MFAS in DCG formation, it would be helpful to tag a DCG marker and measure colocalization with MFAS.

What are these structures that are MFAS-negative and Rab11-positive in the mfas-RNAi condition.

It would be interesting to quantify acidified compartments in the mfas-RNAi conditions.

2. GAPDH is required for mini-core fusion in Drosophila SCs

In line 165, the authors claim that, 'In SCs, DCGs are partially coated with chains of clustered ILVs that extend to the compartment's limiting membrane'. ILVs are small vesicles formed during the maturation of early endosomes into late endosomes/MVBs that give rise to exosomes upon fusion with the PM. To call them ILVs an EM analysis is required. In Hassan Dar et al. 2021, the authors characterized these RAB11-positive compartments as part of the endosomal pathway. Here, it is not clear whether these compartments are DCGs or MVBs or even an intermediate. I believe that this should be further investigated with additional markers and EM.

In line 171, the authors mention that 'the mini-cores appeared to collide less frequently and when they did, fusion usually did not take place'. It would be helpful to quantify the frequency of contacts (movie S3).

3. GAPDH co-isolates with human Rab11-exosomes and other AD-associated glycolytic enzyme biomarkers

The whole set of experiments in this paragraph was performed in order to show that GAPDH positive exosomes are also positive in AD-associated markers in HELA cells as previously demonstrated in HCT116 colorectal cancer cells, and further justify the investigation of APP role in DCG biogenesis. In my view, this is not relevant to this study so it should either be removed or significantly reduced as part of the next section (Drosophila APPL regulates formation of large DCGs in SCs). Line 184-185: This is not the definition of exosomes. It is not the consensus that all exosomes are Rab11- positive. Exosome biogenesis could be ESCRT- dependent or independent. It is not as simple as presented.

Additionally, since there is no evidence that these vesicles are of endosomal origin, please refer to them as Rab11- small EVs.

4. Drosophila APPL regulates formation of large DCGs in SCs

Again here, authors use wrongly the term ILVs and exosomes in RAB11- and RAB6- positive compartments, without demonstrating their endosomal origin. Additionally, I have a concern regarding the size of these so-called exosomes and whether they can be characterized as such with these settings in the microscope. What is the minimum size it could be measured with these settings? Only EM analysis would show the structure corresponding to ILVs/exosomes (size, membrane structure, etc).

Please explain in detail the differences between Appl null and knockdown cells regarding DCG biogenesis.

Please explain the presence of this halo formation in DCGs in APP-YFP and APP-YFP, Appl-RNAi cells.

In Figure S3C, Rab6-positive structures seem to be increased in APP RNAi cells, but the quantification is not in agreement with the images.

5. Cleavage of Drosophila APPL accompanies normal DCG formation &

6. Drosophila APPL and APPL cleavage regulate DCG protein aggregation and dissociation of these aggregates from membranes

To show a correlation between cleavage and maturation, RAB11- and RAB6- tagged cells should be used.

Fig. 4b should be moved in Fig. 5, or Fig. 4&5 could be merged.

Fig. S6: Mean intensity would be more informative.

General comments

a. Throughout the paper it is not clear whether the Rab11- positive compartments are DCGs or MVBs or an intermediate. Maybe the isolation of these compartments and a subsequent proteomic analysis could elucidate the composition of these compartments.

b. LysoTracker can target mild to strong acidic membranous structures such as lysosomes, endosomes, phagosomes and autophagosomes. To show DCG degradation additional markers are required and maybe a biochemical approach would be more appropriate. Autophagy should be considered as an alternative degradation pathway of deformed DCGs.

c. How secretion, mentioned in the discussion section, is addressed?

d. Please improve the quality/resolution of the images

e. The Discussion should be more focused and reduced

Referee #3:

GENERAL SUMMARY: In this manuscript by Singh et al, the authors use secondary cells within the Drosophila accessory gland to explore the function of APP in dense core granule (DCG) and exosome biogenesis. They identify MFAS as a novel dense core granule cargo and test the impact of expression of APP/A β mutants on maturation of these MFAS-containing dense core granules. They suggest that APP plays a role in driving aggregation of DCG cargo, since expression of uncleavable forms of APP or Alzheimer's disease-associated variants of A β interfere with DCG coalescence. The authors posit that these findings mirror intracellular trafficking defects that may underlie early pathogenic changes in Alzheimer's disease. Taken together, the authors describe a new experimental/model system to study the cell biological functions of APP and the role of disease-associated variants in intracellular trafficking.

OVERALL OPINION: The biology described is interesting, but the relevance of the findings to Alzheimer's disease is uncertain. This uncertainty stems largely from the unique nature of the secretory organelles in secondary cells, which produce both a dense core granule and exosome-like intraluminal vesicles within the same maturing secretory vesicle. Moreover, while the experiments are largely well-controlled, the interpretation of the data is sometimes questionable, and some conclusions are overstated. Furthermore, the manuscript lacks sufficient broader context to guide general readers, instead relying heavily on familiarity with the corresponding author's previous work. Given these concerns and the broad readership of EMBO Journal, this manuscript may be more suitable for a specialist journal.

MAJOR CONCERNS

1) Fig. 1C, E: The authors state that knockdown of *mfas* results in large secretory compartments without DCG formation. Wouldn't this be expected if the *mfas*-RNAi reagents are working well? Are the authors implying that formation of all DCGs, including those containing other cargoes are affected, or only the MFAS cargo? Imaging another dense core granule cargo, such as another unrelated hit from the initial screen, with *mfas*-RNAi would help to clarify interpretation of the results. This is critical, since the authors equate defects in *mfas* distribution to defects in dense core granule biogenesis throughout the manuscript. If they cannot definitively show that all dense core granule biogenesis is affected, the presented findings are only relevant to *mfas*-containing dense core granules.

2) In the Fig. 7 summary model, the authors state that initial "clumping" of DCG cargo into minicores depends on the Rab6 to Rab11 transition that occurs during maturation of these secretory organelles. There is no evidence presented in this manuscript to support this statement. The authors then posit that APPL is required for coalescence of minicores into a central large DCG. Given that APPL localizes to the peripheral membrane of these secretory organelles (shown in Fig. 4), it doesn't make sense for this protein to have a specific role in minicore fusion during DCG formation. How do the authors explain the presence of unfused minicores upon loss or mutation of APPL? Additionally, the "network formation" phenotype described for APPL Δ s Δ E1 seems overstated, since the images shown in Fig. 5A show more of a diffuse "cloud" of DCG cargo vs. a peripheral network.

3) Interpretation of expression of the APP and A β mutants presented in Figs. 5 and 6 is complicated. The authors directly state that expression of these mutants induces an ectopic "granule acidification" phenotype that is rarely observed in controls. Therefore, how can the authors be sure that any effects on *mfas* dense core granule formation are specific to expression of the mutant proteins, vs. making the cells generally "sick," thus inducing trafficking defects.

4) The two GAPDH Results sections (lines 154-218) come out of nowhere, since GAPDH was not discussed in the Abstract or Introduction. The logic of examining the role of GAPDH in DCG biogenesis should be clearly laid out and introduced. Given that GAPDH is not included in the summary model presented in Fig. 7, it is not clear whether this data even belongs in this manuscript.

5) Fig. 3B-E: The authors use both GFP-*mfas* and APP-YFP in these experiments. Given that it is not possible to separate GFP and YFP without spectral deconvolution, and the authors show that cleaved APPL localizes to DCGs in Fig. 4, it's not clear that any conclusions can be drawn about the role of human APP in GFP-*mfas* DCG formation.

6) As stated in the overall summary, this reviewer is concerned that understanding of the findings of this manuscript rely heavily on familiarity with the corresponding author's previous work. At the very least, the Introduction needs to do a better job of contextualizing the secondary cell system relative to more familiar secretory cell types (e.g., pancreatic beta-cells), with which more readers are likely to be familiar.

7) In the gene trap screen described in Figure 1, was *mfas* the only "hit?" A supplementary table showing all lines screened and a brief summary of the results would be helpful to understand the scope of the screen. Other hits from the screen could also be used to strengthen the conclusions of other experiments (see point 1 above).

MINOR CONCERNS

8) Lines 287-289/Fig. 4C: Is Zoom #4 labeled properly? This image does not appear to contain "peripheral" RFP/ICD.

9) The placement of Fig. 4B is misleading, considering that the data using these constructs is largely discussed in Fig. 5, not Fig. 4D next to which 4B is placed.

10) The title is misleading and inappropriate. There is no evidence in the manuscript to support the conclusion that A β inhibits the normal function of APP. Additionally, the authors are not examining true recycling endosomes; instead, they use the secretory organelles in secondary cells which are essentially a unique hybrid of secretory granules and recycling endosomes.

11) The placement of bars and asterisks to denote statistical significance is confusing in several panels throughout the manuscript (Figs. 3F, 5B-E, S5E, 6C-E).

ADDITIONAL SUGGESTIONS

12) Lines 128-131: Are the authors equating dominant mutations in human TGFBI with RNAi knockdown of fly *mfas*? Without knowing the nature of the human mutations, it is not clear that such a parallel can be drawn.

13) Movies: It is very difficult to correlate the movies with the still images shown in the figures. Adding an arrow in the movies that points to the specific DCG analyzed in the figures would be helpful.

14) Line 194/Fig. S2: Why are levels of CD63 expected to be reduced? Is this based on a previous study? If so, that study should be referenced.

Referee #4:

- General Summary

The present work addresses the molecular mechanisms underlying the biogenesis of large dense core granules (DCG) and the contribution of amyloid precursor protein (APP) to this process.

This mechanism was addressed through multiple orthogonal in vivo, molecular, and cellular methods, providing convincing results of the contribution of APPL (the drosophila ortholog of human APP) to DCG biogenesis.

The contribution of APPL to DCG biogenesis and maturation provides a potential link to the pathophysiology of Alzheimer's disease (AD), whose genetic basement relies on APP metabolism dysregulation and A peptide production and aggregation to form the so-called amyloid deposits, one of the two neuropathological hallmarks of AD.

As APP metabolism occurs within the secretory pathway, the endolysosomal, autophagy, and proteasome-associated mechanisms, dysregulation can happen in any of these compartments. DCG represents an unexplored cell compartment in which the APP function is unknown, and the consequences of APP mutations are encountered in rare genetic forms of AD. The present work, therefore, provides the first evidence of the contribution of APPL/APP and the potential consequence of APP mutations to DCG biogenesis and maturation.

- Opinion about the principal significance of the study, its questions and findings

The present work provides unequivocal experimental evidence of the contribution of APPL and human APP to large TGFBI/MFAS DCG biogenesis and maturation. The DCG is a vesicular cell compartment in which peptide hormones form protein aggregates into non-soluble dense core granules, which are then released and dissipated upon secretion.

As previously established by co-authors, DCG originates from the trans-golgi network. Rab6 vesicles interact with Rab11 recycling endosomes to form immature DCG and mature DCG containing intraluminal vesicles.

Co-authors first showed that Transforming Growth Factor- β -induced (TGFBI)/MFAS (in Drosophila) is necessary and essential for DCG maturation and protein aggregation, making TGFBI a marker of DCG maturation and protein aggregation.

After that, a fluorescently labeled TGFBI reporter was used to monitor DCG biogenesis and maturation and the contribution of APPL, APP, and mutant APP constructs to DCG.

The results convincingly provide evidence of the contribution of APPL and APP to DCG biogenesis and maturation, priming intraluminal formation, and TGFBI aggregation to form DCG. Loss of APPL or APP dysregulation, therefore, impairs DCG formation.

Several results are potentially overinterpreted and may not support the conclusions of authors that either necessitate additional experiments or a more detailed argumentation.

- Specific major concerns essential to be addressed to support the conclusions

1) Figure 1A provides the scheme of DCG formation, described in the introduction chapter.

Silencing of TGFBI according to Figure 1 and Figure S1 neither impaired YFP-Rab11 nor CFP-Rab6 vesicular formation in flies expressing fluorescently labeled Rab6 or Rab11 separately.

However, transition Rab6/Rab11 vesicles are an indicator of DCG biogenesis, enabling the conclusion that DCG biogenesis is not altered by TGFBI silencing.

Could the authors provide evidence of dually labeled Rab6 and Rab11 vesicles to demonstrate that all steps of DCG biogenesis are preserved in the TGFBI silencing paradigm?

Per se, the presented results enable us to conclude that Rab6 and Rab11 vesicles are observed, but it cannot be concluded that Rab11 vesicles are DCGs lacking TGFBI aggregates. Alternatively, can the author provide additional results to demonstrate that the fusion of Rab6 to Rab11 or recruitment of Rab11 to Rab6 vesicles is preserved in TGFBI silenced cells?

As TGFBI forms amyloid protein aggregate, are these aggregates stained by Thioflavin S or T similarly to other secreted hormone aggregates? Labeling of matured DCG using a different indicator of maturation would strengthen the conclusion of the lack of DCG in the late maturation step.

Page 7, line 151. The authors conclude that TGFBI drives rapid DCG protein aggregation in SCs during compartment maturation.

The limited number of intraluminal Rab11 vesicles supports the conclusion that TGFBI silencing represses DCG maturation before mini-core formation. The authors should further argue how their results support their conclusion.

GAPDH is a glycolytic enzyme necessary to produce ATP in the glycolysis cycle. Glycolysis provides energy to key cellular mechanisms such as microtubule-dependent vesicular transport, sperm flagella motion... Lack of GAPDH represses mini-core fusion into large TGFBI mature dense core intraluminal vesicles. According to the authors, energy provided by glycolysis may be

necessary for mini-core fusion, as supported by several glycolytic enzymes in Rab11 isolated exosomes. The link between GAPDH, exosomes, and amyloid deposits provides the rationale to question the potential role of APP in DCG biogenesis and DCG-derived exosome secretion.

Several cellular vesicular compartments produce extracellular vesicles together merged as exosomes. For instance, exosomes can be produced from multivesicular bodies or herein by DCG vesicles.

It is still determined why the authors performed the proteomic analysis of Rab11 exosomes in an autophagy-related paradigm. Rab11 facilitates the crosstalk between autophagy and the endosomal pathway (Szatmári et al., 2014), and Rab6 promotes insulin receptor and cathepsin trafficking to regulate autophagy induction (Ayala et al., 2018), as referenced in the authors' paper published in 2023.

Their results show that APP and APLP2 are reduced in Rab11 purified exosomes (Table 1) while Rab11 exosome secretion increases. How do authors interpret these results if APPL and APP contribute to DCG maturation? Does the reduced expression of APP promote mini-core granules secretion as exosomes? Authors should determine whether DCG Rab11-derived exosomes remain secreted or not. Herein, the results are puzzling and questioning the rationale of the present work. The rationale between GAPDH, autophagy, Rab11, DCG, APP, and Alzheimer's is not straightforward.

I would rather suggest that authors suppress results with GAPDH and autophagy directly, which could be addressed in different works.

The rationale for the potential contribution of APP to DCG is supported by the presence of APP and APLP2 in Rab11 exosomes, likely produced through a DCG-dependent pathway. Then, is APPL found in DCG? Does APP contribute to DCG biogenesis and maturation? And does APP human mutation impair DCG?

Page 10: Silencing of APPL using siRNA leads to a "mini-core" phenotype, like GAPDH2 knockdown. APP has been described as driving down the regulation of Oxphos and enhancing glycolysis through the non-amyloidogenic metabolism pathway (Lopez Sanchez et al., 2017).

Would the loss of APPL repress the glycolytic cycle and thus mimic the GAPDH2 knockdown?

The APPLd mutant Figure S3D shows a phenotype different from that of APPL silencing, and precisely, the "mini-core" DCG phenotype is not observed. How can authors explain this discrepancy? DCG is observed in APPLd drosophila as observed in Figure S3A. Rab6 and Rab11 DCG are not shown with the APPLd. For a complete overview of two complementary models of APP loss of function, Rab6 and Rab11 vesicles should be shown APPLd drosophila.

Previous studies have shown that APP, carboxy-terminal fragments, including AICD (amyloid intracellular domain) and Abeta peptides, are found in exosomes. According to the present study exosomes arising from DCG contain principally the N-terminal domain of APP and potentially Abeta peptides.

Page 12 and Page 45 are missing methodological details. The authors showed that APPL silencing induces a "mini-core" phenotype. The first rescue experiments in Figure 3S should be shown.

In Figure 4, APPL fluorescent constructs are under a GAL4-UAS expression system, which supposes overexpression of APPL fluorescent constructs together with the endogenous expression of APPL. Control experiments should include the overexpression of APPL UAS-GAL4 in Rab11, Rab6, and MFAS drosophila to demonstrate that overexpression of non-fluorescent APP overexpression does not impair or modify the DCG biogenesis and maturation. Otherwise, experiments with APP fluorescent constructs should be performed in the APPL silenced paradigm first to show that the "mini-core" phenotype is rescued and determine the N-terminal and C-terminal contribution of APP to DCG biogenesis and maturation. Moreover, deciding whether APP is addressed to DCG via Rab6 vesicle or Rab11 appears essential. Authors should, therefore, consider showing the APP double fluorescent construct and the APP construct with either a fused fluorescent protein at the N- or C-terminus of the APP. Moreover, acidification could be related to the overexpression of APPL constructs.

According to the literature, Kuzbanian/ADAM10 drosophila homologous localizes to the cell plasma membrane principally. BACE1 has been described as localizing to axonal Rab11 endosomes (Burggia-Prevot et al., 2014). In the present study, cleavage of APPL (Page 14) appears essential for DCG since lack of APPL N-terminal cleavage leads to an acidification of DCG as acidified vesicles are neither Rab11 nor Rab6. Drosophila expressing the APPL defective for either alpha or beta-cleavage should be shown. These experiments are essential to support the conclusion that overexpression of these APPL cleavage defective constructs leads to a preferential routing of DCG to lysosome compartments and that Rab6 and Rab11 fusion are early DCG biogenesis steps that are not impaired. The overexpression of the APPL constructs remains a question. The role of APP cleavage and DCG biogenesis and maturation should be addressed by secretase silencing, such as Kuzbanian and dbase.

Previous studies have shown that exosomes contain APP, carboxy-terminal fragments, including AICD (amyloid intracellular domain) and Abeta peptides. According to the present study, exosomes arising from DCG contain principally the N-terminal domain of APP and potentially Abeta peptides. The APP C-terminal fragments should not be observed since, in Figure 4, mature RFP is not visualized in mature DCG but localized to acidic compartments. This would, therefore, make a difference between

multivesicular body-derived exosomes and those derived from DCG. Moreover, the localization of GFP in "mini-core" and "dense-core" granules suppose that core structures are not delineated by a lipid membrane and are opposed to exosomes produced from multivesicular bodies. In multivesicular bodies, exosomes are produced following membrane invagination by the ESCRT pathway and leading to an inverted orientation of APP. The C-terminus is inside the vesicle, while the N-terminus of APP is inside. In Figure 4, the mini-core and dense-core contain the N-terminus of APP but not the C-terminus (no RFP labeling).

- Minor concerns that should be addressed

Page 7, line 144 from 152: DCG biogenesis and maturation observed by video microscopy should be placed after line 127, page 6.

If the referee understood well, mini-core intraluminal vesicles fuse to form the dense core intraluminal matured vesicles. They are then supposedly an intermediate step, occurring after Rab6 / Rab 11 vesicle fusion and DCG matured vesicles. Could the authors clearly state, based on their observations, the kinetic of DCG biogenesis and maturation, including the mini-core vesicles?

Figure 4B: Representation of APPL, APPLdeltaSD, and APPLdeltasdE1 constructs should be moved to Figure 5. nAPPLc-GFP construct should be represented. Is the nAPPLc-GFP in Figure 4D like the APPL-GFP in Figure 4A?

Could the author homogenize the nomenclature used? Use TGFBI or MFAS everywhere, please.

- Any additional non-essential suggestions for improving the study (which will be at the author's/editor's discretion)

The present work interests the scientific committee of cell biology and Alzheimer's disease. Only a limited number of scientists are experts in cell vesicle trafficking and DCG biogenesis, maturation, and secretion. Could authors make a slight effort to make their manuscript more accessible to a general audience?

We are grateful to the editor and all the reviewers for their insightful and helpful comments. We have responded to them all below. We also appreciate the opportunity to revise our manuscript and believe that the revised version is now clearer and provides more data to support our conclusions. Furthermore, during the revision process, new parallels with the secretory biology of secondary cells have been reported for human cells, supporting the argument that our findings are likely to be relevant to secretory mechanisms in humans.

Editor's comments

As you will see, the input varies from very supportive and with only minor comments (ref. 1) to more reserved and with major concerns (refs. 2 and 4) to more critical with the comment that the work seems to lack the broader context that would make it interesting for the broad readership of The EMBO Journal (ref. 3). Importantly, a concern that is shared between three of the referees is that the findings have been overinterpreted without sufficient data to conclusively prove them.

Major concerns and overinterpretation: We have addressed these issues below, either by providing additional experimental data, reanalysing data, or by clarifying our findings and in some cases, placing them more explicitly in the context of previously published studies on secondary cells (SCs) and on human secretory cells. The Introduction now includes additional text concerning our current understanding of SC biology that complements the original introductory schematic in Figure 1A. We believe this will provide the background required to more easily interpret the data in the manuscript.

Broad context (see Ref #3's 'overall opinion' below): A recent paper (Stockhammer et al., 2024, 'ARF1 compartments direct cargo flow via maturation into recycling endosomes', Nat Cell Biol, doi: 10.1038/s41556-024-01518-4) has revealed the similarities between Arf1-dependent regulated secretory compartment formation in human cells versus *Drosophila* secondary cells, making our analysis likely to be of direct relevance to other secretory cell types in invertebrates and vertebrates. We appreciate that this new paper marks a significant deviation from previously proposed models for human Arf1-dependent regulated secretion, but there have been several indications in published work that the recycling endosomal system could be involved in this process, including in neurons, several of which we had flagged up in the first version of the manuscript, eg. Sugawara et al., Genes Cells, 2009, 14:445-56 [29]; Koles et al., J Biol Chem., 2012, 287:16820-34 [92]; Li et al., J Neurol., 2023, 270:1487-1500 [50].

Referee #1

1. The resolution of the figures is extremely poor, some text was hard to read.

We apologise for this. On reviewing the figures in the submitted pdf, there appears to have been a significant loss of resolution in the jpeg and pdf conversions that were undertaken to produce a single manuscript file, hence the effect also seen with the text in these figures; this is not an issue with the individual figure files that we have now submitted. For the SC images, despite the large size of SC compartments, we are at the limit of resolution for live-cell studies of sub-compartmental biology, if undertaken at sufficiently high throughput to allow robust quantification. The Zoom images merely represent an enlargement of the whole-cell images with no additional microscopic magnification. We believe the clearer figures do illustrate the key points that we are making.

2. On line 129, the authors state: "Knockdown of mfas with two independent RNAs specifically in adult SCs produced large secretory compartments with no DCGs (Figures 1C, 1E and S1A)". I agree with the interpretation but I think a little more accuracy is needed here as it is so important for the manuscript. I suggest something like "produced large secretory compartments with no observable DCGs in the brightfield indicating a lack of protein condensation" or something the authors are comfortable with.

This referee agrees that we can deduce that large DCG protein condensations do not appear to be made in an *mfas* knockdown background through assessment of our bright-field (DIC) images. We have more accurately articulated this point in the text, as suggested (lines 149-151). Furthermore, we have now also included data with another DCG marker, GFP-GPI (Redhai et al., 2016), in response to comments from other referees (see Ref #2, point 1), and these experiments confirm our overall conclusion (lines 169-186 and new Figure EV1H).

3. The manuscript is heavily dependent on microscopy phenotypes. I have no major issue with this, however in this case the quantification becomes absolutely essential and effectively is the data. However, the manuscript is somewhat lacking here. For quantification and statistical analysis, each "n" should equal an independent experimental repeat (as that is the variability that is important). However, I think each cell may resemble the other. Some of the n values are listed as more than 60, and it would strengthen the manuscript if the number of independent experimental repeats is clarified and possibly increased.

This is a good and important point. Thank you. Some time ago, we showed that the level of inter-animal variation was no greater than the level of variation within animals for our exosome studies, but we have not done this for all the DCG biogenesis phenotypic assays, some of which are novel. We have therefore reanalysed these data, following the suggestions of the referee, calculating the mean value for each phenotype in each animal analysed typically for 3 cells. n now represents the number of independent animals imaged and is ≥ 10 for most measurements. In generating these data for each genotype, we independently repeated each experiment at least three times. The changes in our statistical analysis have slightly altered the calculated P values, and in one case (γ -secretase knockdown and abnormal DCG formation [Figure EV3]), the change is no longer significant. Otherwise, the findings are unaltered, reflecting the fact that there is no major inter-animal variation in most of the phenotypes we observe.

4. It was unclear to me what exactly was overexpressed in figure 6 (the abeta peptides and mutants). In the text, it says, "wild-type form of the pathological A β -42 peptide". However, the peptide is extracellular- so I presume it was signal peptide-Abeta peptide (or mutants). I would very much appreciate it if this was clarified. If these Abeta peptides are being expressed cytosolically, then I would suggest this is a misleading reagent and I would recommend this data be removed from the manuscript.

Again, this is a very helpful comment; apologies for not making this clearer. The peptide constructs all carry an ER signal peptide within the coding sequence at their N-terminal end, so they will be secreted following signal peptidase cleavage. They have previously been used in neurodegeneration studies in flies (citations in original manuscript). We have clarified this in the relevant Results section

(lines 473-474) and in the Methods (lines 763-764).

Referee #2

1. Drosophila TGFBI drives DCG assembly in SCs

It is not clear why MFAS was investigated as a potential DCG protein. Other known DCG proteins should be used to show DCG morphology. To further show the selectivity of MFAS in DCG formation, it would be helpful to tag a DCG marker and measure colocalization with MFAS.

Thank you for raising this point, which needed further clarification. We chose the GFP-MFAS marker because it is expressed from the endogenous *mfas* gene locus (so not overexpressed) and it is the only gene trap we have found to date that strongly and exclusively labels the DCGs of SCs in the accessory gland (see also Ref #3, point 7). We also used bright-field DIC throughout our analysis to independently identify DCGs and mini-cores (see Ref #1, point 2). GFP-MFAS co-localises with DIC-detectable protein aggregates in normal DCGs and mini-cores (see Figures 1, 2, etc). DIC imaging had suggested there were only sporadic small puncta detectable inside large non-acidic secretory compartments in *mfas* knockdown SCs.

We accept that it is possible that aggregated material might form in DCG compartments in an MFAS-independent fashion, and that these abnormal aggregates might not be apparent by DIC, though this seems relatively unlikely. Furthermore, we did not investigate the small puncta that were observed by DIC in *mfas* knockdown cells (now, for example, marked with a grey arrowhead in Figure 1C). We have therefore provided additional data using a second published SC DCG (and DCG compartmental membrane) marker, UAS-GFP-GPI (Redhai et al., 2016). These data are presented in new Figure EV1H, and confirm that like MFAS, this marker traffics to DCG structures visible by DIC in normal SCs, in addition to compartment membranes and ILVs. However, in the absence of MFAS, GFP-GPI only concentrates in the small intraluminal puncta observed by DIC. Further analysis of our Rab localisation data in this genetic background indicates that many of these puncta are strongly labelled with Rabs, and are therefore likely to represent clusters of ILVs that we had shown are still made in these cells and would be expected to be marked by GFP-GPI (lines 169-186). We conclude that there is little, if any, protein condensation into DCG-like structures in *mfas* knockdown SCs.

What are these structures that are MFAS-negative and Rab11-positive in the mfas-RNAi condition. It would be interesting to quantify acidified compartments in the mfas-RNAi conditions.

Following *mfas* knockdown, we believe the non-acidic compartments that are Rab11-positive, but lack GFP-MFAS and a DCG, are secretory structures, which have correctly transitioned to Rab11 identity, but cannot form a DCG. We demonstrate that these compartments form internal Rab11-positive puncta (Figures 1 and EV1), which we have previously shown are Rab11-ILVs containing intra-vesicular Rab11 that will be secreted as Rab11-exosomes (Fan et al., 2020; Dar et al., 2021; Marie et al., 2023). Rab11-ILV formation requires the Rab6 to Rab11 transition observed during DCG compartment maturation (Wells et al., 2023), supporting our conclusion that these secretory compartments have matured relatively normally, but cannot condense their DCGs. The new data with GFP-GPI as a marker further consolidates this idea, because in *mfas* knockdown cells, this marker still traffics into these Rab11-compartments, but only concentrates at the compartment's limiting membrane and colocalises with clusters of ILVs (Figure EV1H). We make this point on lines 174-182.

It is a good point that we had not assessed the acidification phenotype. This was partly omitted because the phenotype was introduced later in the manuscript, since we first realised that SCs could generate this phenotype, when we were studying DCG compartments in *GAPDH2* and *Appl* knockdown cells, which still make protein aggregates. In fact, acidification is also increased in an *mfas* knockdown background. We have now included this result in new Figure EV1I with other *mfas* knockdown data, but refer to it later in the text (lines 214-215).

2. GAPDH is required for mini-core fusion in Drosophila SCs

In line 165, the authors claim that, 'In SCs, DCGs are partially coated with chains of clustered ILVs that extend to the compartment's limiting membrane'. ILVs are small vesicles formed during the maturation of early endosomes into late endosomes/MVBs that give rise to exosomes upon fusion with the PM. To call them ILVs an EM analysis is required. In Hassan Dar et al. 2021, the authors characterized these RAB11-positive compartments as part of the endosomal pathway. Here, it is not clear whether these compartments are DCGs or MVBs or even an intermediate. I believe that this should be further investigated with additional markers and EM.

We apologise that we did not explain our previous published findings adequately in the first version of the manuscript. In line with additional suggestions from referees #3 and #4, we have included a fuller explanation of these findings in the Introduction. The general relevance is now supported by the recent Stockhammer et al., Nat Cell Biol paper (new reference 37) on related Arf1-dependent secretory compartments in human cells (see Ref #3, 'overall opinion' for details; in this paper, EM was not employed to determine whether ILVs might also be present). The Rab11-compartments in SCs are DCG secretory compartments with recycling endosomal identity (Redhai et al., 2016) that also produce ILVs. These ILVs are marked by Rab6 and Rab11, as shown through fluorescence microscopy (Fan et al., 2020; Wells et al., 2023); the presence of ILVs was confirmed by EM studies and 3D-super-resolution microscopy in Fan et al., 2020. Therefore, the suggested EM analysis and characterisation of these compartments to confirm the presence of ILVs has already been undertaken.

Fan et al., 2020 also demonstrated that human cells, including HeLa cells, which were employed in Stockhammer et al., 2024, make a specific Rab11-exosome subtype and the data presented in Marie et al., 2023 indicate there are parallels in the specialised mechanisms by which these vesicles are generated in flies and humans (accessory ESCRT-III regulation; some aspects of our Rab11-exosome analysis are reviewed in van Niel G and Théry C, 2020, EMBO J. 39:e105119, and new ref. 35). The introductory text on lines 90-113, and 160-164 now provides more details concerning previously published work in this area.

In line 171, the authors mention that 'the mini-cores appeared to collide less frequently and when they did, fusion usually did not take place'. It would be helpful to quantify the frequency of contacts (movie S3).

Thank you for pointing this out. We have tried to quantify these data, but it has proved challenging for two reasons. First, for wild type cells, there is only one compartment per cell per movie, the one that is generating a new DCG from mini-cores, which we can analyse, so the n number is small (5). Second, and perhaps more importantly, in the *GAPDH2* knockdown, there are many more compartments with mini-cores, but they presumably represent DCG compartments at different

stages of maturation, where the mini-cores have failed to fuse. Many are therefore not directly comparable to the early DCG compartment we are studying in wild type cells.

Having tried to focus in on the least mature DCG compartments in *GAPDH2* knockdown SCs, which are in the process of forming mini-cores (only one, at most, in each SC movie), we found that there was too much variation to be sure that the collision frequency was reduced and it was more difficult to score all collisions with certainty, because the mini-cores usually do not fuse. We have therefore removed the comment on the frequency of collisions, but our data very clearly show that the mini-cores are motile and fail to fuse in the absence of *GAPDH2* (unlike *AppI* knockdown and mutant A β -42 expression), so we have retained that aspect of the phenotypic description (lines 204-206). We do not think that this change affects the main conclusions of the manuscript in any way; we had no explanation for why the mini-cores might collide less frequently than in wild type cells and on reflection, this might be linked to the fact that we cannot really compare like for like.

3. GAPDH co-isolates with human Rab11-exosomes and other AD-associated glycolytic enzyme biomarkers

The whole set of experiments in this paragraph was performed in order to show that GAPDH positive exosomes are also positive in AD-associated markers in HELA cells as previously demonstrated in HCT116 colorectal cancer cells, and further justify the investigation of APP role in DCG biogenesis. In my view, this is not relevant to this study so it should either be removed or significantly reduced as part of the next section (Drosophila APPL regulates formation of large DCGs in SCs).

We have considered this point carefully, particularly since other referees raised issues about including specific aspects of the GAPDH data, eg. Referee #3, point 4 and Referee #4, point 8.

A key and logical reason for discussing *GAPDH2* knockdown in the first place was that it induces a mini-core phenotype in SCs that had been reported in Dar et al., 2021. We immediately thought that the mini-core intermediate stage during normal DCG biogenesis (Figure 1H) was reminiscent of this phenotype, and our previous finding that extravesicular GAPDH clusters ILVs and exosomes (Dar et al., 2021) potentially provided an explanation for why mini-cores do not coalesce as they do in wild type cells. We should have made this clearer when we discussed the phenotype and have now included comments on GAPDH function and the mini-core phenotype in the Introduction (lines 109-113) and Results section (lines 193-198) respectively.

As suggested, we have pared down the proteomics section, though we did not feel that it fitted well into the *AppI* knockdown section. We believe that the set of findings related to GAPDH are important to present, because: i. they provide a link between Rab11-exosomes and AD through the proteomic analysis of the CSF secretome in AD patients versus healthy controls (lines 242-250); ii. as the referee highlights, this justifies the analysis of SC Rab11-compartments in the context of APP function; iii. for the fly phenotype, we later show an important difference between *GAPDH2* knockdown and *AppI* knockdown (reduced mini-core motility in *AppI* knockdown; Figure 3F), which supports our conclusion that membrane:aggregate dissociation is defective in the absence of APPL.

Line 184-185: This is not the definition of exosomes. It is not the consensus that all exosomes are Rab11- positive. Exosome biogenesis could be ESCRT- dependent or independent. It is not as simple as presented.

We apologise that this has led to confusion. We did not intend to suggest that all exosomes are made in Rab11-compartments or are Rab11-positive. In fact, a critical aspect of our previous studies

is that Rab11-exosomes represent a specialised subset of exosomes and are regulated by a specific combination of ESCRTs (Fan et al., 2020; Marie et al., 2023; Mason et al., 2024). This sentence has been removed in cutting back the section, but the distinction between Rab11-exosomes and late endosomal exosomes is explained at greater length in the Introduction (lines 94-100).

Additionally, since there is no evidence that these vesicles are of endosomal origin, please refer to them as Rab11- small EVs.

Fan et al., 2020 provided exhaustive evidence that these vesicles are made within Rab11-positive secretory compartments with recycling endosomal identity, work backed up by subsequent publications (Marie et al., 2023; Wells et al., 2023); the latter reference shows that the Rab6 to Rab11 transition is required for Rab11-exosome formation. These vesicles are, therefore, exosomes, because we are actually visualising the ILVs in a compartment marked by a recycling endosomal marker (eg. Figures 1B, EV1B, EV2B, etc in revised manuscript) and we have also followed their formation in these compartments using real-time imaging (Wells et al., 2023). This background information is now explained in greater depth in the Introduction (lines 90-100) and its general implications have potentially been extended by the observation that human regulated secretory compartments also appear to undertake a switch to recycling endosomal identity (new ref 37).

4. Drosophila APPL regulates formation of large DCGs in SCs

Again here, authors use wrongly the term ILVs and exosomes in RAB11- and RAB6- positive compartments, without demonstrating their endosomal origin. Additionally, I have a concern regarding the size of these so-called exosomes and whether they can be characterized as such with these settings in the microscope. What is the minimum size it could be measured with these settings? Only EM analysis would show the structure corresponding to ILVs/exosomes (size, membrane structure, etc).

These issues were also dealt with extensively in Fan et al., 2020, as discussed in point 3 above. In the revised manuscript, we now discuss the relevant data in this paper in the Introduction (lines 90-100) and an introductory section of the Results (lines 160-164). Fan et al. confirmed that the structures we visualise in Rab11-compartments that are marked by YFP-Rab11 and CFP-Rab6, are exosomes through EM and super-resolution microscopy. Subsequent reviews by leaders in the field have acknowledged the existence of this new subtype of recycling endosome-derived exosomes (eg. van Niel G and Théry C, 2020, EMBO J. 39:e105119, and ref. 35). We have employed the Rab11 and Rab6 markers, visualised using the same microscopy techniques employed in the manuscript, to confirm that genetic manipulations (eg. ESCRT knockdowns), which suppress Rab11-exosome formation and secretion, simultaneously reduce Rab6 and Rab11 puncta in Rab11-compartments (Marie et al., 2023), and to follow ILV biogenesis (Wells et al., 2023). We acknowledge that this type of live-imaging approach would be challenging in other cell types, which have much smaller multivesicular endosomes, but when coupled with the *Drosophila* genetics that we can exploit, this illustrates the unique advantage of working with secondary cells, which has also allowed us to make the observations concerning APP function in the current manuscript.

Wide-field microscopy will only identify fluorescent ILVs or clusters of ILVs as puncta, but in Fan et al., we were able to visualise these vesicles individually by 3D-super-resolution microscopy and show that the majority appeared to be in the standard exosome size range (~30-150 nm diameter; Figure EV1A in that paper) and that they were labelled by specific transmembrane markers (eg. CD63-GFP, Btl-GFP; see also Marie et al., 2023). These findings were also backed up by EM in SCs and

biochemical studies of human Rab11-exosomes, showing that Rab11a is an intravesicular protein (Fan et al., 2020).

Please explain in detail the differences between Appl null and knockdown cells regarding DCG biogenesis.

Thank you for pointing out that this needed further clarification. The major differences between the phenotypes are that *Appl* knockdown cells have a mini-core phenotype in DCG compartments and an elevated number of acidified compartments, while *Appl* null mutants frequently have large membrane-associated DCGs and even higher numbers of acidified compartments. The differences could either be explained by incomplete knockdown, or alternatively and perhaps more likely, compensatory effects or acquisition of repressors in the viable *Appl* mutant, which was generated over 30 years' ago (see Ref #4. Point 11 for further details). We now consider these explanations in the Discussion of the revised manuscript (lines 625-635).

Importantly, the commonalities between the two genotypes, which we present in the manuscript, are the aberrant persistence of limiting membrane:aggregate interactions in mature DCG compartments and an enhanced DCG compartment acidification phenotype when compared to controls (Figures EV2E-G). These phenotypes directly support our model concerning APP function. We have also characterised the *Appl* mutant phenotype further in response to Ref #4, point 11, demonstrating that the aberrant non-acidic DCG compartments in these cells have transitioned to Rab11 identity, as with *Appl* knockdown cells (lines 276-277, new Appendix Figure S2), so there is not a fundamental difference in that aspect of their maturation.

Please explain the presence of this halo formation in DCGs in APP-YFP and APP-YFP, Appl-RNAi cells.

Currently, we do not understand why in some genetic backgrounds that we have analysed, the GFP associated with GFP-MFAS is excluded from the centre of the DCG (scored as GFP-negative centre of $\geq 1 \mu\text{m}$ diameter). One possibility, but certainly not the only one, is that in the case of overexpressed APP, the separation of the extracellular domain of APP from the membrane at the compartment periphery is suppressed or delayed and so the central aggregation of DCG material occurs abnormally and the middle part of the DCG lacks GFP-MFAS. We have briefly considered this phenotype in the Discussion (lines 618-624), also highlighting that this is observed in a β -secretase knockdown. Importantly, whatever the explanation, this does not affect the main conclusions in the manuscript.

In Figure S3C, Rab6-positive structures seem to be increased in APP RNAi cells, but the quantification is not in agreement with the images.

Some of the compartments that are marked by Rab6 in one of the *Appl* knockdown cells (*Appl*-RNAi #2) do appear to have more concentrated intraluminal peripheral Rab6 within them (as we also saw with *GAPDH2* knockdown; Dar et al. 2021), but overall, the proportion of compartments containing these vesicles is unaffected versus controls, the variable displayed in the bar chart (now Figure EV2L). This measurement generally appears to best assess overall levels of ILV/exosome biogenesis in these cells (Marie et al., 2023; Wells et al., 2023), since changes in ILV clustering can alter the CFP-Rab6 and YFP-Rab11 distribution and intra-compartmental signal intensity within specific z-planes inside compartments (see Dar et al., 2021 and this manuscript). This makes it difficult to compare

fluorescence intensities with wild type, particularly when the clusters are peripherally located and difficult to distinguish from the limiting membrane of compartments. We did highlight this clustering phenotype in the original version of the manuscript (lines 269-271).

5. Cleavage of Drosophila APPL accompanies normal DCG formation &

6. Drosophila APPL and APPL cleavage regulate DCG protein aggregation and dissociation of these aggregates from membranes

To show a correlation between cleavage and maturation, RAB11- and RAB6- tagged cells should be used.

This is an interesting, but more challenging, point to address since there is spectral overlap between GFP (APPL marker) and YFP/CFP (Rab11 and Rab6 markers), so we are not able to cleanly undertake this experiment. However, because there is little, if any, GFP-labelled APPL ECD on the limiting membrane of mature DCG compartments, we have been able to show that YFP-Rab11 is localised at the limiting membrane of these compartments, when dt-APPL and YFP-Rab11 are co-expressed (lines 342-346; new Appendix Figure S3B).

It is not possible to use this approach to confirm the identity of immature Rab6-positive compartments that lack a DCG, because at this stage, APPL's GFP-labelled ECD is still at the limiting membrane. However, these compartments are the only large non-acidified compartments that lack a central DCG in secondary cells, and they frequently have a characteristic relatively central location, as in Zoom #1, Figures 4B and 4C (see also Wells et al., 2023 for their detailed characterisation). We have now noted this point in the results section (lines 326-329).

Importantly, other data in the original manuscript already indicated that the compartments containing cleaved APPL are mature DCG compartments, because we independently identified the DCG by DIC (Figures 4B and 4C); we had also previously shown that compartments must take on a Rab11 identity to form a DCG, and when they do this, they form ILVs and a large DCG within less than an hour (Wells et al., 2023).

Fig. 4b should be moved in Fig. 5, or Fig. 4&5 could be merged.

Thank you, that is helpful. Figure 4B has now become Figure 5B.

Fig. S6: Mean intensity would be more informative.

We do use mean intensity and have clarified this on the bar chart.

General comments

a. Throughout the paper it is not clear whether the Rab11- positive compartments are DCGs or MVBs or an intermediate. Maybe the isolation of these compartments and a subsequent proteomic analysis could elucidate the composition of these compartments.

Since secondary cells make up less than 4% of the cells in the accessory gland (and in total, 80 cells per animal), it is unrealistic to isolate these compartments and, even if it was possible, huge numbers of flies would be required for proteomics. The secondary cell system has unique advantages for *ex vivo* cell biological and genetic studies, but is not suitable for biochemical analysis. We have characterised SC DCG compartments through live-cell imaging in detail in Fan et al., 2020,

and Wells et al., 2023, and shown they are both DCG compartments and MVBs (see also Ref #2, point 2 above).

The recent paper by Stockhammer et al, 2024, Nat Cell Biol, doi: 10.1038/s41556-024-01518-4, suggests that human Rab11-positive regulated secretory compartments are also formed via an Arf1-mediated Rab11 transition, as we have shown in secondary cells (Wells et al., 2023; Arf1 is a well-established regulator of regulated secretory compartments in higher organisms, see lines 84-89). The approaches in Stockhammer et al do not have the resolution to determine whether Rab11-ILVs are generated in these compartments; however, they mention this possibility in the context of Wells et al., 2023 in their Discussion. We have shown in Fan et al., 2020 that Rab11-marked exosomes can be formed from endosomal compartments in human cancer cells and proteomic characterisation of these human vesicles has allowed us to identify novel and, in some cases, selective Rab11-exosome regulators, initially through genetic testing of *Drosophila* orthologues in secondary cells (Dar et al., 2021; Marie et al., 2023; this manuscript). We therefore believe that our combined studies of Rab11-exosomes in SCs and human cells are continuing to reveal cell biological parallels, despite the difficulties in using both live-cell cell biological and biochemical approaches in the same cell model. We have expanded on these points in the Introduction and Discussion in the revised manuscript (lines 90-113 and we re-highlight the relevance of Stockhammer et al in lines 575-577).

b. LysoTracker can target mild to strong acidic membranous structures such as lysosomes, endosomes, phagosomes and autophagosomes. To show DCG degradation additional markers are required and maybe a biochemical approach would be more appropriate. Autophagy should be considered as an alternative degradation pathway of deformed DCGs.

These are important points. As discussed above, biochemical approaches are not feasible in this system to identify the identity of the large acidic compartments in SCs. We have previously shown that large acidic compartments have Rab7 identity and suggested they are likely to be lysosomes, partly based also on EM analysis (Fan et al., 2020; Figs. 1 and EV1E from that paper). Late autophagosomes can also carry Rab7 (Hytinen et al., 2013, doi: 10.1016/j.bbamcr.2012.11.018). However, SC acidic compartments increase in size, as normal cells age, and we have shown that they occasionally fuse with DCG compartments (Corrigan et al., 2014), which is likely part of the explanation for their progressive increase in size. Interestingly, we have previously reported that one mature DCG compartment (also labelled by Rab11) recruits Rab7 (Corrigan et al., 2014; Fan et al., 2020). This may well be the compartment that is occasionally targeted for lysosomal degradation in normal cells (Figure EV1G), but we know through live imaging that the compartment is usually secreted, suggesting that these changes may be part of a final quality control step that we are yet to fully characterise.

Regarding the dissociation of the DCG in acidified compartments, as a first step in DCG degradation, this is apparent not just because of the dispersal of GFP-MFAS, but also the loss of condensed material in DIC images, which we have emphasised more clearly in the revised manuscript (lines 207-214 and Figure EV1G). The latter observation is important because GFP-labelled markers are quenched as compartments become acidified, so their subsequent dispersion can be difficult to follow by live imaging, and RFP/mCherry markers, which are not quenched, frequently promote increased generation of acidified compartments in SCs, making them less appropriate for analysing normal degradative events. We have yet to follow the full progression of steps from initial peripheral acidification to complete compartmental acidification in a single movie.

Although we believe the large fully acidified compartments are lysosomal, this reviewer's point that autophagic mechanisms may also be involved in the degradative targeting of DCG compartments remains valid. In other cell types, DCG compartments can be degraded by several autophagic mechanisms, some of them specialised for secretory compartments (Szenci et al., 2023, doi: 10.1242/jcs.260741), so this is a relatively complex question to address. Unfortunately, the standard tool employed to assess autophagic trafficking in fly cells, a GFP/mCherry double-tagged *Atg8a* transgene induces very severe lysosomal trafficking of SC DCG compartments, probably because of the excess trafficking of mCherry-labelled constructs to lysosomes in SCs. It is, therefore, not useful in determining whether autophagy is normally taking place. We have knocked down two major regulators of autophagy in adult SCs, *Atg1* and *Atg8*, using two independent RNAis, and we observe that DCG compartments are made, but there is a very clear build-up of partially acidified compartments.

Therefore, autophagy seems to be involved in aspects of the DCG degradation process in SCs, but does not seem to be required to permit initial acidification of mature DCG compartments that are targeted for degradation. To characterise the specific mechanisms involved will require a much more exhaustive set of experiments, where we test the full range of candidate genes that are known to disrupt different mechanisms regulating DCG compartment breakdown in other cells, look at the effects on ILV production and secretion, etc. We think such an analysis would not add significantly to the main messages in the manuscript and would be best presented as a more complete independent story, including real-time imaging in different knockdown backgrounds. Consequently, we have decided to omit the *Atg* knockdown data (please see also that this is in line with the recommendation of Ref #4, point 8), although they could be added as an Appendix Figure, if it was thought by this referee that they were critical to include.

c. How secretion, mentioned in the discussion section, is addressed?

We demonstrated secretion of DCG material in the manuscript by measuring the levels of GFP-MFAS in the lumen of the gland (now Appendix Figures S9A and S9B). These data very clearly show that GFP-MFAS, exclusively made by SCs, is released into the lumen and if anything, that the level of secretion seems to increase when A β -peptides are expressed in these cells (this does not reach significance for A β -42-Dutch), suggesting that the defective compartments formed in these genetic backgrounds primarily fuse with the apical plasma membrane and are not degraded.

Regarding Rab11-exosome secretion, this specific point was raised by Ref #4, point 7. We have included new data (new Appendix Figures S9C and S9D) using a Btl-GFP marker, which we have used routinely in other publications to measure Rab11-exosome secretion in intact glands during adulthood (Fan et al., 2020; Dar et al., 2021; Marie et al., 2023). We find the number of Btl-GFP puncta secreted into the lumen of the gland when A β -peptides are overexpressed is unchanged when compared to wild type glands (lines 526-531 and new Appendix Figure S9D). We have briefly discussed these data in the revised manuscript (lines 683-684).

d. Please improve the quality/resolution of the images

Thank you and apologies for this issue, which we believe was related to the pdf conversions that were made prior to submission. Please see Ref #1, point 1 above for further explanation. We believe we have now addressed this problem and the individual images submitted are of much higher quality.

e. The Discussion should be more focused and reduced

We have tried to address this point in the revised manuscript. Most notably, we have removed a paragraph on MFAS right at the start and rewritten a section on cell death in SCs, as well as making multiple additional shorter edits, although there have also been some additions in response to comments from this and other referees.

Referee #3

OVERALL OPINION: The biology described is interesting, but the relevance of the findings to Alzheimer's disease is uncertain. This uncertainty stems largely from the unique nature of the secretory organelles in secondary cells, which produce both a dense core granule and exosome-like intraluminal vesicles within the same maturing secretory vesicle. Moreover, while the experiments are largely well-controlled, the interpretation of the data is sometimes questionable, and some conclusions are overstated. Furthermore, the manuscript lacks sufficient broader context to guide general readers, instead relying heavily on familiarity with the corresponding author's previous work. Given these concerns and the broad readership of EMBO Journal, this manuscript may be more suitable for a specialist journal.

We thank this referee for their helpful comments. We appreciate that some aspects of our study rely on our previous work, but this is partly due to the unique advantages of the model we are using, permitting us to visualise dynamic sub-compartmental maturation and aggregation events that are extremely challenging to study in other systems. In our fly work, we have consistently been informed by data from human cell studies, such as EV proteomics analysis and the genetics of DCG biogenesis (Fan et al., 2020; Dar et al., 2021; Marie et al., 2023; Wells et al. 2023; this manuscript), so that we focus on the most evolutionarily conserved biology, where possible.

The newly published paper from Stockhammer et al., 2024, 'ARF1 compartments direct cargo flow via maturation into recycling endosomes', Nat Cell Biol, doi: 10.1038/s41556-024-01518-4, highlighted above in the general points at the beginning of our response, now extends those parallels (discussed in revised Introduction; lines 101-104). It demonstrates that an Arf1-regulated Rab11 transition of Golgi-derived compartments to recycling endosomes is required for human regulated secretory pathway maturation, which, as the authors highlight, mirrors our previous work in flies (Wells et al., 2023). The presence of ILVs in these human compartments is not shown in Stockhammer et al., but as the authors explain, using their live-cell super-resolution imaging approaches, it is not possible to follow the aggregation of secretory cargos, let alone ILVs, particularly because of z-plane movement of the small compartments involved. Therefore, currently we do not know whether dynamic formation of ILVs in DCG compartments might take place in human cells – if it does, there will definitely be far fewer ILVs made in each compartment than in SCs, and they are likely on average to be smaller (cf. Fan et al., 2020, Figure EV1A), because of the much smaller compartment size.

Importantly, the model we propose for APP function and the effects of A β does not require ILVs, since there are some ESCRT knockdowns in SCs, which strongly reduce ILV formation, but do not affect DCGs (Marie et al., 2023); it is merely dependent on Rab11-positive recycling endosomal membranes of secretory compartments, including the limiting membrane of the compartment. These membranes can act as primers for aggregation and then normally separate from aggregates following APP cleavage. Since in human cells, the secretory Rab11-compartments are much smaller versus secondary cells, we would expect the limiting membrane to play a much more major role in the former, though we cannot exclude a normal, but potentially redundant, role for ILVs as well. However, the link between ILVs and APPL in secondary cells and the defects in ILV and DCG interactions induced in these cells by mutant A β -peptide expression (see also Ref #3, point 10 and new Appendix Figure S7) may be relevant to the discovery of A β -peptide-associated exosomes in

amyloid plaques (as we discussed in the original manuscript, now on lines 717-724). Furthermore, most of the work reported in Stockhammer et al., 2024 was undertaken in HeLa cells, the same cell type that we use in our manuscript to identify Rab11-exosome cargos that overlap with the AD secretome, data that indicate a possible role for Rab11-compartments and Rab11-exosomes in disease-relevant secretion. These observations indicate that our study, which further characterises the regulation of Rab11-positive secretory compartments, is potentially relevant to researchers with interests in regulated secretion, protein aggregation, APP function and neurodegeneration in vertebrates as well as invertebrates.

1) Fig. 1C, E: The authors state that knockdown of mfas results in large secretory compartments without DCG formation. Wouldn't this be expected if the mfas-RNAi reagents are working well? Are the authors implying that formation of all DCGs, including those containing other cargoes are affected, or only the MFAS cargo? Imaging another dense core granule cargo, such as another unrelated hit from the initial screen, with mfas-RNAi would help to clarify interpretation of the results. This is critical, since the authors equate defects in mfas distribution to defects in dense core granule biogenesis throughout the manuscript. If they cannot definitively show that all dense core granule biogenesis is affected, the presented findings are only relevant to mfas-containing dense core granules.

This is an important point and is discussed in our responses to Ref #2, point 1 and Ref #1, point 2. In summary, we had used DIC, which allows visualisation of DCGs and mini-cores, and not just the presence and localisation of GFP-MFAS, to demonstrate the absence of or changes in dense-core structures in the original manuscript. However, we now include data with an additional fluorescent DCG marker, GFP-GPI (Redhai et al., 2016), confirming that large DCGs are not formed in an *mfas* knockdown background and the resulting secretory compartments only contain small puncta visible by DIC, which appear to contain clusters of ILVs (lines 169-182; Figure EV1H).

2) In the Fig. 7 summary model, the authors state that initial "clumping" of DCG cargo into minicores depends on the Rab6 to Rab11 transition that occurs during maturation of these secretory organelles. There is no evidence presented in this manuscript to support this statement.

Thank you for this helpful comment. We have discussed more clearly in the Introduction to the revised manuscript (lines 94-97) our published work showing that ILV formation and DCG biogenesis, the latter scored by DIC microscopy, a reliable method for visualising DCGs and mini-cores (point 1 above; see also Dar et al., 2021), requires the Rab6 to Rab11 transition (Wells et al, 2023). This latter paper shows a requirement for *Rab11*, *Arf1*, and *AP-1* subunits through multiple genetic manipulations. The referee is correct that in our new manuscript, we have not shown additional evidence to eliminate the possibility that in the absence of the Rab6 to Rab11 transition, mini-cores are formed that are not visible by DIC, though to date, such structures have not been observed in the range of different genotypes we have assessed using the GFP-MFAS marker. We have reworded the statement in the figure legend to more precisely reflect our current understanding.

The authors then posit that APPL is required for coalescence of minicores into a central large DCG. Given that APPL localizes to the peripheral membrane of these secretory organelles (shown in Fig. 4),

it doesn't make sense for this protein to have a specific role in minicore fusion during DCG formation. How do the authors explain the presence of unfused minicores upon loss or mutation of APPL?

Intact APPL localises both to the limiting membrane of the compartment and to internal ILVs, and is cleaved during the process of forming a central DCG in compartment maturation (Figure 4), an event that we show is required for normal DCG biogenesis (Figure 5). In the presence of non-cleavable APPL, not all aggregated material formed at the limiting membrane of the maturing DCG compartment can dissociate from the membrane (Figure 5). When the non-cleavable APPL Δ sde1 mutant is expressed, the interactions between the ECD of APPL and the aggregates appear particularly stable and the aggregates remain in a peripheral network. We think these data are consistent with APPL having a role in peripheral aggregate assembly and then the involvement of a cleavage-dependent event in releasing these aggregates to form a central DCG.

In the absence of APPL, we believe that the early stages of DCG aggregation are not regulated normally, because APPL can no longer be participating in the association between peripheral aggregates and the compartment's limiting membrane and ILVs. In these circumstances, the APPL-cleavage-dependent release of peripheral aggregates from membranes during formation of the central DCG cannot take place, and the abnormal membrane-associated aggregated material is more frequently retained at the limiting membrane. We have discussed this point further in the Discussion to support the model set out in Figure 7 (lines 598 onwards, including lines 625-635).

We should emphasise that we therefore believe that our *App1* knockdown and overexpression data indicate, perhaps unsurprisingly, that APPL is not the only protein controlling aggregation and membrane dissociation events associated with DCG biogenesis, hence the observation that both of these manipulations can lead to peripheral aggregate accumulation.

Additionally, the "network formation" phenotype described for APPL Δ sde1 seems overstated, since the images shown in Fig. 5A show more of a diffuse "cloud" of DCG cargo vs. a peripheral network.

Apologies, we think this was partly due to a loss of resolution during pdf conversion, but also because we presented the image in the same format as other DCG compartment images, using a z-projection through the compartment. We have now also included multiple images of the same compartment from different z-planes, which we think displays this unique phenotype more clearly. In this genotype, GFP-labelled aggregates are largely excluded from the central region of the compartment, but the peripheral material is localised in intersecting strands rather than mini-cores, which by DIC also appear different to mini-cores when visualised (new Figure 5A, bottom image panels). We have also now analysed YFP-Rab11 and CFP-Rab6 distribution in this genetic background, and find that it is organised in a complex intraluminal peripheral network, presumably reflecting the close association between membranes and aggregates in this genetic background (lines 429-432 and Appendix Figure S4).

3) Interpretation of expression of the APP and A β mutants presented in Figs. 5 and 6 is complicated. The authors directly state that expression of these mutants induces an ectopic "granule acidification" phenotype that is rarely observed in controls. Therefore, how can the authors be sure that any effects on mfas dense core granule formation are specific to expression of the mutant proteins, vs. making the cells generally "sick," thus inducing trafficking defects.

Thank you. This is an important point, which we only partly commented on in the original manuscript; it should have been discussed in more detail. We accept that in any situation where proteins or peptides are expressed, especially if they are known to ultimately induce degenerative phenotypes, it is difficult to completely eliminate the possibility that an undetected defect that ultimately leads to degeneration is affecting other processes like trafficking indirectly. However, we have now discussed the effects of protein/transgene overexpression in the SC system more generally, as summarised below, and included new data addressing the referee's specific point, which supplements evidence that was already presented in the original manuscript. Overall, we believe that these data provide a good argument that the effects of APPL/APP protein and A β -peptide overexpression on DCG biogenesis primarily involve defects in specific aspects of maturation, while other events accompanying maturation, including trafficking steps, still take place. Particularly for A β -expressing SCs, secretory protein aggregation still occurs, but we observe specific defects in DCG biogenesis during time-lapse imaging as soon as immature DCG compartments start to form these aggregates, even though the compartments take on Rab11-identity and appear to lose Rab6, as observed during control DCG compartment maturation (see below; Appendix Figure S6). These findings are not easily explained as a general trafficking disruption resulting from the cells being 'sick', since this disruption would only be indirectly affecting selected events in early DCG biogenesis, several days before A β -expressing cells exhibit more general defects in their morphology (see below).

Regarding the point that overexpression of proteins in the secretory system of SCs might generally induce trafficking defects and compartment acidification, we have overexpressed other proteins associated with the secretory system's limiting membrane, like a GFP-labelled FGFR (Btl-GFP) and YFP-Rab11, and not observed major changes in compartment acidification (Corrigan et al., 2014; Fan et al., 2020). Indeed, the formation of compartments with peripheral acidic microdomains was a novel phenotype, which we first appreciated in our new study when visualising *Appl* knockdown cells. This acidification phenotype is also not observed with overexpression of the additional DCG marker that we now employ in the revised manuscript, GFP-GPI (new Figure EV1H). Therefore, the defects are not just the result of protein overexpression in secretory compartments (now explained on lines 296-298).

To determine whether the general trafficking events associated with DCG compartment maturation occur normally in the APPL and A β overexpression backgrounds, we have assessed the Rab identity of these compartments and shown that compartments with aggregated GFP-MFAS are Rab11-positive and a proportion are Rab6-positive, strongly indicating that the Rab6 to Rab11 transition has taken place (new Appendix Figures S4 and S6).

Importantly, for the A β -peptide mutants, our time-lapse movies of early DCG compartment maturation events in glands from 6-day-old males, which already contain multiple defective DCG compartments that will have been formed over the previous 24-48 hours (Redhai et al., 2016), also suggest that new DCG-forming events are still occurring in A β -peptide-expressing cells. However, these events have a very specific defect, phenocopying *Appl* knockdown, where the mini-cores that are formed and the compartments, which make them, appear to be immobilised in comparison to normal cells (Figures 3F, 6F and 6G). Interestingly, our new analysis of the effects of A β -peptides on processing of double-tagged (dt-)APPL (new Appendix Figure S7) suggests that these peptides are indirectly or directly affecting APPL cleavage, which might provide part of the explanation for the mini-core phenotype. Since the A β -peptides have an ER signal sequence attached to them, they will be localised inside the secretory and endosomal compartments that we are studying, and mostly secreted. Therefore, since we observe specific defects in these compartments as they are formed, it

seems unlikely that these defects are generated by indirect actions of A β -peptides that are mediated by affecting the general health of the cell and then selectively feeding back to the secretory system through a specific process during DCG compartment maturation, particularly since more general effects on cell morphology and viability only appear later (see below).

For overexpression of APPL proteins and A β -peptides, we have also investigated whether some SCs have more general defects in cell morphology over a time course (3, 6 and 12 days post-eclosion; new Appendix Figures S5 and S8). For A β -peptide expression, as for expression of several other transgenes and RNAis in SCs, we do not observe obvious DCG compartment phenotypes at 3 days, presumably because it takes several days for transgene transcript levels to accumulate. However, we find that it is only after 12 days that we detect significant numbers of morphologically abnormal SCs, some of which are almost certainly dead, which still contain GFP, but lack visible secretory compartments by DIC (new Appendix Figure S8). With overexpression of the wild type A β -peptide, which also generates an acidification phenotype and can induce neurodegeneration in other systems, most SCs retain normal general morphology, even after 12 days. They also start to develop a mini-core phenotype, supporting the idea that acidification and defective DCG biogenesis are primary phenotypes and not an indirect effect of 'cell sickness'.

The interpretation of phenotypes induced by APPL constructs requires somewhat more caution, even though we find that the Rab6 to Rab11 transition has taken place in maturing DCG compartments in all genotypes (new Appendix Figure S4). Importantly, APPL-WT does not have significant effects on cell viability even after 12 days of overexpression (Appendix Figure S5). However, expression of the mutant constructs does induce much higher levels of general cellular abnormality at 12 days. The main rationale for studying the APPL mutants was to determine whether uncleaved APPL with and without a functional E1 domain generates any defects in DCG biogenesis or maturation in newly forming DCG compartments, in addition to the increased compartment acidification observed with APPL-WT overexpression (Figure 5). For both mutants, we do see differences. For APPL- Δ sdE1, the defect is highly specific and consistent with stabilised membrane-associated aggregates. Interestingly, severe morphological defects are induced in the majority of SCs, even after 3 days of APPL- Δ sdE1 expression, and the proportion of these cells only gradually increases over the subsequent 9 days. We think this likely reflects different levels of transgene expression in each SC, which are maintained throughout the time course, and indicates that many of the SCs that display the network phenotype at 3 and 6 days are destined to survive at least until day 12.

The APPL- Δ sd DCG compartment phenotype is somewhat less easy to interpret in the context of the more general effects on cell morphology associated with its expression at 12 days, because the primary defect is an enhancement of the acidification phenotype, which might be induced in 'sick' cells that only develop more general morphological defects some days later. However, importantly, this result is consistent with the findings for secretase knockdowns (Figure EV3), where RNAis, not proteins, are being overexpressed, and as discussed above, the secretory compartments in these cells have the expected Rab11 identity, so are in other ways maturing normally.

We have considered these points in the Results sections where we present the new data (lines 401-407, 414-415, 425-447, 507-513) and in the Discussion (lines 598-611, 665-671, etc).

4) The two GAPDH Results sections (lines 154-218) come out of nowhere, since GAPDH was not discussed in the Abstract or Introduction. The logic of examining the role of GAPDH in DCG biogenesis

should be clearly laid out and introduced. Given that GAPDH is not included in the summary model presented in Fig. 7, it is not clear whether this data even belongs in this manuscript.

This is a very helpful comment. As discussed in the response to Ref #2, point 3, we believe that the GAPDH data are important to include, because they provide a link with the mini-core formation observed in wild type cells and to the proteomics data from AD patients. The motility of the mini-cores is also important to contrast with reduced motility in the *App1* knockdown and A β -mutant overexpression phenotypes (Figures 2E, 3G, 6F and 6G). However, we should have included some discussion of our published GAPDH data in the Introduction, rather than leave this to the Results section. We have now done this (lines 109-113). We found it difficult to include GAPDH in the final model schematic, given that the primary focus is on highlighting the roles of APP and A β -mutants, but we have now referred to it in the legend.

5) Fig. 3B-E: The authors use both GFP-mfas and APP-YFP in these experiments. Given that it is not possible to separate GFP and YFP without spectral deconvolution, and the authors show that cleaved APPL localizes to DCGs in Fig. 4, it's not clear that any conclusions can be drawn about the role of human APP in GFP-mfas DCG formation.

This is an important point that we should have explained more clearly. As we discuss early in the results, MFAS is expressed at extremely high levels in secondary cells and so the GFP-MFAS signal is much stronger than the signal produced by APP-YFP – in fact, it is very difficult to detect APP-YFP, even when excitation levels are increased. In addition, the referee is correct to highlight that we show the N-terminal ECD of APPL localises to DCGs. But the APP protein used for rescue in Figure 3 is only tagged at its C-terminal membrane-associated end, which we show for APPL is present on ILVs and the limiting membrane of DCG compartments, and much of it traffics to lysosomes as DCG compartments mature (Figure 4C). The fact that we see no signal at the limiting membrane of DCG compartments in the experiments where we combine GFP-MFAS and APP-YFP confirms that at the gain settings we use to detect GFP, the YFP signal in these compartments is too weak to detect. Furthermore, DIC images also confirm that a large DCG is formed in rescued SCs, independently of GFP-MFAS.

However, to confirm that we are only visualising GFP-MFAS in these experiments, we have undertaken an experiment with APP-YFP expression in the presence and absence of GFP-MFAS (new Appendix Figure S3A). We show that we cannot detect APP-YFP in the DCG or on the DCG compartment limiting membrane, using the YFP detection channel, even with high levels of excitation, which lead to bleed-through into the YFP channel from excitation of both GFP-MFAS and LysoTracker Red (lines 290-293 and new Appendix Figure S3A). We therefore can conclude from the GFP images alone that APP-YFP rescues the DCG defects induced by *App1* knockdown, producing a large GFP-labelled central DCG in most compartments.

6) As stated in the overall summary, this reviewer is concerned that understanding of the findings of this manuscript rely heavily on familiarity with the corresponding author's previous work. At the very least, the Introduction needs to do a better job of contextualizing the secondary cell system relative to more familiar secretory cell types (e.g., pancreatic beta-cells), with which more readers are likely to be familiar.

We apologise for this. We have now revised the Introduction and introductory parts of some results sections, as discussed in several of our points above, better explaining the previous secondary cell

literature and contextualising it, and highlighting specific mammalian cells that generate DCG compartments, as suggested by this referee, eg. see lines 81-84 and 90-113. The recent paper by Stockhammer et al., 2024 supports the argument that SCs share similarities in the control of their regulated secretion with human cells, in ways which were not established in the literature when we first submitted the manuscript.

7) In the gene trap screen described in Figure 1, was mfas the only "hit?" A supplementary table showing all lines screened and a brief summary of the results would be helpful to understand the scope of the screen. Other hits from the screen could also be used to strengthen the conclusions of other experiments (see point 1 above).

We have added a Supplementary Table, Appendix Table S1, to highlight the lines that we initially screened some years ago, when a limited number of gene traps were available. Of the small number of gene traps screened, *GFP-mfas* was one of two GFP fusions that expressed selectively in the relatively non-abundant secondary cells of the accessory gland. The other was Angiotensin converting enzyme (ANCE), which was the first SC DCG marker identified through immunostaining studies (Rylett et al., 2007, J Exp Biol. 210:3601-6. doi: 10.1242/jeb.009035). We have found that cell morphology is much better preserved with live cell imaging, so have not used the ANCE antibody used by those authors for some time in our studies (but see Redhai et al., 2016 for an example). The ANCE gene trap expresses at far lower levels than GFP-MFAS. It was only selected because we already knew it marked SC DCGs. It was not a good tool for the studies described here, especially for detecting mini-cores. However, we have addressed this referee's point 1 by using another GFP-labelled DCG marker (see Ref #3, point 1 above).

MINOR CONCERNS

8) Lines 287-289/Fig. 4C: Is Zoom #4 labeled properly? This image does not appear to contain "peripheral" RFP/ICD.

This image (now Figure 4B, Zoom #4) shows a putative acidified compartment with a partially granular appearance by DIC in which the DCG is dispersed and the contents are degrading. This is broadly equivalent to the lower Zoom images for each genotype in Figure 3B, where we can more easily recognise these compartments by staining their peripheral acidic microdomains with LysoTracker; we observe that GFP-MFAS is absent or completely quenched in these compartments. With dt-APPL, the peripheral mRFP-labelled APPL ICD domain is consistently internalised and therefore accumulates throughout the lumen in these compartments, as was highlighted in the text, line 349-351. Lysosomal compartments in these cells also contain the internalised RFP-labelled ICD (large bright red compartments in Figure 4B).

When the ICD is labelled by GFP, the internalised GFP is absent or only weakly fluorescent inside this class of compartments, because of quenching. In compartment Zoom #4, Figure 4C (low and high magnification), low levels of internal GFP fluorescence are observed and, in this case, the compartment can be recognised by its acidic microdomains (LipidTox-positive).

In the same paragraph where the Figure 4B, Zoom #4 compartment was discussed, we also mentioned compartments 'labelled peripherally by the ICD contained diffuse GFP inside them, consistent with early stages of DCG compartment acidification' and incorrectly referred to Figure 4B,

Zoom #4 as the example of that. An actual example of this type of compartment was not marked in the original version of the manuscript, but it has now been labelled with a white arrowhead (lines 347-349). These compartments presumably represent the very earliest stages of DCG compartment acidification, before internalisation of the ICD.

9) The placement of Fig. 4B is misleading, considering that the data using these constructs is largely discussed in Fig. 5, not Fig. 4D next to which 4B is placed.

Thank you, that is a helpful point. We have moved this panel to Figure 5B.

10) The title is misleading and inappropriate. There is no evidence in the manuscript to support the conclusion that A β inhibits the normal function of APP. Additionally, the authors are not examining true recycling endosomes; instead, they use the secretory organelles in secondary cells which are essentially a unique hybrid of secretory granules and recycling endosomes.

We have considered this point carefully. The title does not claim that A β inhibits APP function, only that it inhibits protein aggregation and dissociation, which is itself APP-regulated. In addition, we have now included data showing that mutant A β -peptides do interfere with the normal processing of double-tagged APPL at the limiting membrane and on ILVs (new Appendix Figure S7). These effects could be indirect, but they suggest a complex impact of mutant A β -peptide expression on APPL, protein aggregation and aggregate dissociation.

Furthermore, we think the work of Stockhammer et al. now raises issues about the definition of a 'true' recycling endosome, and this is reflected in the title of their paper: 'ARF1 compartments direct cargo flow via maturation into recycling endosomes'. The compartments we study definitely have recycling endosomal Rab11 identity, as do the maturing secretory compartments in Stockhammer et al. In many studies where Rab11-positive 'recycling endosomes' are analysed, it has not been excluded through live-cell imaging analysis or other approaches that some of these compartments might have originated as secretory compartments from the Golgi.

Given these points, our strong preference is to slightly modify the title to: 'A β disrupts APP-regulated protein aggregation and dissociation from recycling endosomal membranes', but if the referee believes that this is still misleading, an alternative, which does not provide any distinction between the pathological and physiological roles of A β versus APP respectively is: 'APP and A β modulate protein aggregation and dissociation from recycling endosomal membranes'.

11) The placement of bars and asterisks to denote statistical significance is confusing in several panels throughout the manuscript (Figs. 3F, 5B-E, S5E, 6C-E).

Thank you for pointing this out. We have changed the formatting for these graphs.

ADDITIONAL SUGGESTIONS

12) Lines 128-131: Are the authors equating dominant mutations in human TGFBI with RNAi knockdown of fly mfas? Without knowing the nature of the human mutations, it is not clear that such a parallel can be drawn.

No, in this section we were not trying to link the effects of dominant mutations in human TGFBI with *mfas* knockdown, but had raised this point to highlight that mutant forms of human TGFBI also have aggregating properties. None of the work we present in the manuscript is linked to the human TGFBI mutant phenotypes. However, we agree with the referee that the positioning of this sentence may be confusing, so we have deleted it, but retained a brief consideration of this point in the Discussion (lines 570-571).

13) Movies: It is very difficult to correlate the movies with the still images shown in the figures. Adding an arrow in the movies that points to the specific DCG analyzed in the figures would be helpful.

Thank you for this helpful suggestion. We have done this.

14) Line 194/Fig. S2: Why are levels of CD63 expected to be reduced? Is this based on a previous study? If so, that study should be referenced.

This is based on the Fan et al., 2020 study [now ref 32], which was referenced in the previous sentence. We have rewritten and shortened this part of the results, quoting the reference when CD63 is mentioned (lines 223-228)

Referee #4

Overall, Referee #4 appears to broadly accept that the evidence we present provides new insights into the potential role of APP in the biogenesis and maturation of secretory compartments and how A β -peptides might interfere with the same processes. However, they have some concerns regarding interpretation, some of which we have addressed experimentally, while in other cases, we believe the data we originally presented need further explanation or clarification.

- Specific major concerns essential to be addressed to support the conclusions

1) Figure 1A provides the scheme of DCG formation, described in the introduction chapter.

Silencing of TGFBI according to Figure 1 and Figure S1 neither impaired YFP-Rab11 nor CFP-Rab6 vesicular formation in flies expressing fluorescently labeled Rab6 or Rab11 separately.

However, transition Rab6/Rab11 vesicles are an indicator of DCG biogenesis, enabling the conclusion that DCG biogenesis is not altered by TGFBI silencing.

Could the authors provide evidence of dually labeled Rab6 and Rab11 vesicles to demonstrate that all steps of DCG biogenesis are preserved in the TGFBI silencing paradigm?

This is an interesting, but difficult, experiment to undertake in any quantitative way. For reasons that we do not understand, when we co-express these two fluorescent Rabs (CFP-Rab6 and YFP-Rab11) from their endogenous loci, they appear to affect each other's expression in many SCs, so we rarely see cells with both good Rab6 and good Rab11 staining (we imaged those relatively rare cells in the Wells et al., 2023 paper, but used independent lines for quantification in that paper, which we also do in this manuscript).

However, we do not think that this double-labelling experiment is required for four reasons:

i. We have only identified one route by which the exceptionally large, Rab11-positive non-acidic DCG compartments are made in secondary cells, which involves formation of large Rab6-positive compartments on the *trans*-Golgi face and then their transition to Rab11 identity (Wells et al., 2023). Stockhammer et al., 2024, Nat Cell Biol, doi: 10.1038/s41556-024-01518-4 suggests this process is conserved for much smaller human secretory compartments. We think it is extremely unlikely that *mfas* knockdown would induce a new mechanism for generating very large Rab11-positive compartments in SCs, which is independent of this Golgi route;

ii. We also think it is unlikely that double-labelling of the ~10 compartments that might be expected to contain a DCG in *mfas* knockdown secondary cells will tell us more than single labelling, particularly given the variability in expression observed with this combination, explained above. In any cells that are clearly marked for both Rabs, using the latter approach, there will inevitably be substantial overlap, since our experiments have already shown that about 8 compartments are labelled by each marker, which correlates with our previous data, and indicates that about 5 or 6 'mature DCG compartments' are co-labelled (see Wells et al., 2023), though these lack a DCG in *mfas* knockdown SCs. We appreciate that if there was not a significant overlap between Rab6- and Rab11-positive compartments, it would be difficult to deduce that the latter had matured from the former. But it is difficult to envisage how these jointly labelled compartments would be made in the absence of a Rab6 to Rab11 transition. We have used separate Rab6 and Rab11 lines previously to analyse DCG compartment maturation and identified genotypes that failed to undertake the Rab6 to Rab11 transition and form a DCG, as determined by DIC (Wells et al., 2023), so we think this is a valid approach;

iii. In many SCs, at least one 'immature' non-acidic precursor DCG compartment that lacks a DCG is typically centrally located and Rab6-labelled, but usually only has diffuse Rab11 structures outside it. Such compartments are visible in the *mfas* knockdown cells in Figures 1D, EV1B and EV1D, suggesting that this step, which is the one that would be most clearly distinguished by double-labelling, is retained.

iv. Furthermore, we not only show that Rab6 and Rab11 compartments are still present in *mfas* knockdown cells, we also show that Rab6- and Rab11-positive ILVs are formed in these compartments. Since we have previously shown that this requires a Rab6 to Rab11 transition to take place (Wells et al., 2023), this strongly suggests that the transition must have occurred, especially for Rab6-positive ILVs to form.

We have expanded on some of these points in the first Results section (lines 151-168).

2) Per se, the presented results enable us to conclude that Rab6 and Rab11 vesicles are observed, but it cannot be concluded that Rab11 vesicles are DCGs lacking TGFBI aggregates. Alternatively, can the author provide additional results to demonstrate that the fusion of Rab6 to Rab11 or recruitment of Rab11 to Rab6 vesicles is preserved in TGFBI silenced cells?

We may be misunderstanding the first statement here (see also point 12 below), but we assume Ref #4 refers to Rab6- and Rab11-compartments, when discussing Rab6 and Rab11 vesicles (and not Rab6- and Rab11-ILVs). As discussed above, our data indicate that jointly labelled Rab6/Rab11 compartments must be formed in *mfas* knockdown SCs, which would be difficult to envisage in the absence of a Rab6 to Rab11 transition, and that these compartments display a Rab11-dependent activity observed in normal DCG compartments, the production of Rab6- and Rab11-positive ILVs.

3) As TGFBI forms amyloid protein aggregate, are these aggregates stained by Thioflavin S or T similarly to other secreted hormone aggregates? Labeling of matured DCG using a different indicator of maturation would strengthen the conclusion of the lack of DCG in the late maturation step.

Testing for amyloid formation in the DCG is an interesting suggestion, although not all DCGs are believed to form amyloid during aggregation and for human TGFBI, only mutant protein appears to form amyloid. We have tried to use amyloid stains on accessory glands under several genetic conditions, but this has not been successful, even when we express known amyloidogenic proteins. We think this is at least partly explained by poor stain penetration through the outer muscle layer, since we often observe penetration issues with other dyes and antibodies.

The analysis using an alternative cargo marker (see Ref #2, point 1 and Ref #1, point 2; lines 169-186) confirms that a mature DCG is absent after *mfas* knockdown and that the small puncta visible by DIC in this background are likely to primarily contain clusters of ILVs, supporting our original conclusion.

4) Page 7, line 151. The authors conclude that TGFBI drives rapid DCG protein aggregation in SCs during compartment maturation.

The limited number of intraluminal Rab11 vesicles supports the conclusion that TGFBI silencing represses DCG maturation before mini-core formation. The authors should further argue how their results support their conclusion.

We are not clear about the referee's point here. As the referee states, our experiments show that neither DCGs nor smaller peripheral mini-cores are formed in an *mfas* knockdown cell, using DIC, and now, a GFP-tagged DCG marker, GFP-GPI, to detect aggregate formation. YFP-Rab11 marks the ILVs within these compartments, which are frequently associated with mini-cores (Dar et al., 2021 and this manuscript), but these ILVs can exist independently of aggregates and indeed, appear to form shortly before DCG biogenesis (Wells et al., 2023). We do not see an obvious difference in the proportion of Rab11-compartments with Rab6- or Rab11-positive ILVs after *mfas* knockdown (Figures 1D, 1G, EV1B, EV1C and EV1F). Since ILV formation seems to precede aggregation (Wells et al., 2023), this is what we might have expected. Changes in ILV biogenesis can affect DCG biogenesis in SCs, but the relationship between ILVs and protein aggregates is complex and, in some genotypes, reductions in ILV numbers are not accompanied by obvious changes in DCG morphology (Marie et al., 2023). Therefore, we do not believe that we can use our data concerning Rab11-ILVs in *mfas* knockdown SCs to support our conclusions concerning the aggregation properties of MFAS.

5) GAPDH is a glycolytic enzyme necessary to produce ATP in the glycolysis cycle. Glycolysis provides energy to key cellular mechanisms such as microtubule-dependent vesicular transport, sperm flagella motion... Lack of GAPDH represses mini-core fusion into large TGFBI mature dense core intraluminal vesicles. According to the authors, energy provided by glycolysis may be necessary for mini-core fusion, as supported by several glycolytic enzymes in Rab11 isolated exosomes. The link between GAPDH, exosomes, and amyloid deposits provides the rationale to question the potential role of APP in DCG biogenesis and DCG-derived exosome secretion.

We do not argue in the manuscript that the effects of *GAPDH2* knockdown are linked to metabolic changes in secondary cells. Nor do we know whether glycolytic enzymes associated with ILVs might play a metabolic role in controlling DCG protein aggregation.

Using both knockdown and overexpression, we previously showed that extra-vesicular GAPDH is involved in ILV and exosome clustering both in fly and human cells, apparently acting as a vesicle cross-linker; we provided evidence that this activity in SC Rab11-compartments affects DCG

biogenesis (Dar et al., 2021). The ILVs in *GAPDH2* knockdown SCs are concentrated around peripheral mini-cores detected by DIC (Dar et al., 2021). In this new manuscript, we observe that the mini-cores do not fuse when they collide in *GAPDH2* knockdown SCs, perhaps because this fusion requires adhesion between associated ILVs.

Notably in flies, there are two *GAPDH* genes, which may have some redundant roles (Sun et al., 1988, Mol Cell Biol. 8:5200-5), so *GAPDH2* knockdown may only have limited effects on metabolism.

6) Several cellular vesicular compartments produce extracellular vesicles together merged as exosomes. For instance, exosomes can be produced from multivesicular bodies or herein by DCG vesicles.

It is still determined why the authors performed the proteomic analysis of Rab11 exosomes in an autophagy-related paradigm. Rab11 facilitates the crosstalk between autophagy and the endosomal pathway (Szatmári et al., 2014), and Rab6 promotes insulin receptor and cathepsin trafficking to regulate autophagy induction (Ayala et al., 2018), as referenced in the authors' paper published in 2023.

We consider the role of autophagy in our response to Ref #2, point b, and now have generated data showing that blocking autophagy increases the number of DCG compartments that have become acidified at their periphery; however, DCG compartments are still formed. We have not checked for Rab11-ILV formation as yet, though we suspect it is still taking place. As we discuss in our response to Ref #2, point b, and in line with Ref #4's point 8 below, we think this new phenotype will require significant further analysis to interpret fully, and we do not believe that this directly adds to the APP functional analysis, so we have not included our autophagy data in the manuscript.

Regarding the point about both Rab11's and Rab6's interactions and functions in autophagy and the possible identity of the Rab11- and Rab6-labelled compartments, the recent paper by Stockhammer et al., 2024, now suggests a general mechanism by which regulated secretory compartments are formed via a transition to Rab11 identity. Although autophagy is involved in the quality control of mature DCG compartments (Szenci et al., 2023, doi: 10.1242/jcs.260741), we do not believe there is currently evidence that it is required for the initial maturation or the actual secretion of regulated secretory compartments, although our recent observations following inhibition of the autophagy pathway in SCs merit further investigation to determine autophagy's precise downstream roles.

We developed a nutrient-depletion/mild mTORC1 suppression protocol for enhanced Rab11-exosome secretion from human cancer cells in Fan et al., 2020, and demonstrated a change in endosomal flux from the late endosomal to the recycling endosomal pathway under these conditions in HCT116 cells. These data indicated that enhanced recycling endosomal trafficking plays an important role in the increased levels of Rab11-exosome secretion in this scenario. The finding that orthologues of human Rab11a-exosome cargos are regulators of fly Rab11-exosome and DCG formation (eg. in Marie et al., 2023, Dar et al., 2021, this new manuscript and other unpublished data), and the observation that DCG compartments are still apparently formed normally when autophagy is suppressed in SCs, also supports our hypothesis that non-autophagic mechanisms are involved in the generation of these compartments and the ILVs and DCGs within them. However, this does not exclude that autophagy is involved in a downstream event associated with DCG compartment quality control.

7) Their results show that APP and APLP2 are reduced in Rab11 purified exosomes (Table 1) while

Rab11 exosome secretion increases. How do authors interpret these results if APPL and APP contribute to DCG maturation? Does the reduced expression of APP promote mini-core granules secretion as exosomes? Authors should determine whether DCG Rab11-derived exosomes remain secreted or not. Herein, the results are puzzling and questioning the rationale of the present work. The rationale between GAPDH, autophagy, Rab11, DCG, APP, and Alzheimer's is not straightforward.

We are somewhat unclear about the point being made here. In our response, we first describe our findings below (see also point 12 below). The protein aggregates that make DCGs are largely formed from extravesicular proteins arriving from the secretory and recycling endosomal pathways. In the fly model, the Rab11-ILVs that become Rab11-exosomes appear to associate with aggregated material, and part of that association involves APP-like molecules, which are normally then cleaved, so the extracellular domain of APP remains associated with the DCG (Figure 4). The intracellular domain remains attached to membranes (compartmental and ILVs) and most of it traffics to acidic compartments, where it is internalised (Figure 4). We have previously confirmed in human cells that Rab11 is inside Rab11-exosomes (Fan et al., 2020), as expected. At least under normal conditions in secondary cells, most aggregated material in the large DCG is not directly associated with ILVs in Rab11-compartments, which lie only at the periphery of the DCG, and when the DCG is dissipated upon secretion, most GFP-MFAS must be separated from Rab11-exosomes (compare GFP-MFAS [DCG material; relatively homogeneous] and Btl-GFP [Rab11-exosomes; punctate] distribution in accessory gland lumen in new Appendix Figure S9). When *Appl* is knocked down or mutant A β -peptides are expressed, the dissociation of aggregates from the compartmental limiting membrane and ILVs is suppressed. The peripheral mini-cores formed still appear to be secreted and it is possible that more aggregated material remains associated with the Rab11-exosomes that are also released, though we do not have an assay at present to assess this. However, mini-cores are not secreted ‘as exosomes’ – these are distinct entities.

In Table 1 (now Appendix Table S2), the referee correctly notes that APP and APLP2 are both reduced in human Rab11-exosome-enriched EV preparations versus late endosomal exosome-enriched preparations, suggesting that these proteins may be more abundant on extracellular vesicles that are not Rab11-exosomes. But this certainly does not exclude that they are also present on Rab11-exosomes (as accepted by this referee; point 9). In human cells, there is extensive published evidence that APP is trafficked and cleaved within the late endosomal pathway, so it is quite possible that APP protein and its cleavage products are present at high levels in association with ILVs within these compartments. This may also happen in secondary cells; indeed, we show in Figure 4B that the APPL-mRFP intracellular domain appears to be trafficked at high levels to the lumen of acidic compartments in this cell model. We have previously shown these compartments contain many ILVs by EM and live cell imaging (Corrigan et al., 2014). The GFP marker on dt-APPL is quenched in acidic compartments, so it is difficult to determine whether any ECD is present. Importantly, we have previously shown that in SCs, the vast majority of secreted ILVs (exosomes) appears to be generated in Rab11-compartments, not late endosomes (Corrigan et al., 2014; Fan et al., 2020; Marie et al., 2023). Therefore, although APP might be present at high levels inside late endosomal and lysosomal compartments in human and fly cells (and this is also supported by an extensive literature), we do not think this affects our interpretation of the SC data for secretory DCG compartments, where we have clearly shown APPL associates with Rab11-ILVs and DCG compartment membranes. Furthermore, the fact that human Rab11-exosomes carry a different set of cargos to late endosomal exosomes (Marie et al., 2023) suggests that APP-associated ILVs from these two compartments are likely to have different composition, which may be relevant to the novel APP functions we have identified in secretory compartments with Rab11-identity.

In relation to whether Rab11-exosome secretion is affected in genetic backgrounds where mini-cores are formed, the assay we have developed for Rab11-exosome secretion involves counting fluorescent puncta in the accessory gland lumen marked by the exosome marker Breathless-GFP (Fan et al., 2020; Dar et al., 2021; Marie et al., 2023). This assay cannot distinguish individual exosomes from clusters, which often form around mini-cores, but it has routinely detected major changes in SC exosome secretion (see references above).

In response to Ref #2, point c, we have now presented new data using this assay in genetic backgrounds where A β -peptides are expressed in secondary cells (new Appendix Figures S9C and S9D). The number of Btl-GFP puncta seems unchanged when any of the three A β -peptides are overexpressed versus controls (lines 526-531 and new Appendix Figure S9D).

8) I would rather suggest that authors suppress results with GAPDH and autophagy directly, which could be addressed in different works.

We consider the roles of GAPDH2 and autophagy in Ref #4, points 5 and 6 above, and explain in the latter why we have omitted data on autophagy, as suggested by this referee.

The referee also proposes omitting the GAPDH data, potentially because it adds a major new topic of metabolism to the manuscript. We do not believe that the role of GAPDH2 is primarily metabolic (see Dar et al., 2021), so we have not omitted it, but clarified how GAPDH fits into our model (outlined further in Ref #2. Point 3 and Ref #3, point 4) in the revised manuscript in the Introduction, Results and Discussion (lines 109-113, 204-206 and 563-565 respectively). We think inclusion of our GAPDH studies is important in linking the manuscript to our previous work and because GAPDH's effects on compartment/mini-core motility contrasts with some of the new mechanisms we have identified (motility versus immotile compartments and mini-cores).

9) The rationale for the potential contribution of APP to DCG is supported by the presence of APP and APLP2 in Rab11 exosomes, likely produced through a DCG-dependent pathway. Then, is APPL found in DCG? Does APP contribute to DCG biogenesis and maturation? And does APP human mutation impair DCG?

As this reviewer states, the identification of APP-like molecules associated with Rab11-exosomes only supports a direct contribution of APP to DCG biogenesis, if APP is also associated with DCGs. A core finding in the manuscript is that the APPL extracellular domain (ECD) is found in DCGs (Figure 4B). We also show that *Appl* knockdown affects DCG biogenesis and maturation, and that human APP overexpression can rescue the DCG morphology defects (Figure 3). And then we show that overexpression of mutant forms of fly APPL and human A β -peptides (Figure 5 and 6) disrupt DCG biogenesis. We think these points address the referee's questions, unless they are also suggesting that we express mutant forms of APP that mirror the mutant forms of APPL, which we have already tested in SCs. We do not think that such experiments would add to our mechanistic understanding, given that we already know that human APP can rescue the *Appl* knockdown mini-core phenotype, even though it is known that it is not processed as efficiently as APPL in *Drosophila* (ref. 51).

10) Page 10: Silencing of APPL using siRNA leads to a "mini-core" phenotype, like GAPDH2 knockdown. APP has been described as driving down the regulation of Oxphos and enhancing glycolysis through the non-amyloidogenic metabolism pathway (Lopez Sanchez et al., 2017). Would the loss of APPL repress the glycolytic cycle and thus mimic the GAPDH2 knockdown?

App1 knockdown produces a mini-core phenotype, but it is different from *GAPDH2* knockdown, because the defective DCG compartments and the mini-cores within them display highly reduced motility. As with *GAPDH2* knockdown, we cannot eliminate a possible effect of *App1* knockdown on metabolism or other cellular processes, which might ultimately have indirect effects on DCG compartment maturation. However, the effects of APPL and A β mutants on DCG biogenesis, particularly the formation of a peripheral protein aggregate network following APPL- Δ sdE1 expression, strongly suggest a more direct effect of this protein on DCG formation, made more likely because of APPL's localisation in DCG compartments, the presence of its extracellular domain in DCGs and its intracellular domain on Rab11-membranes, and the correlation between DCG biogenesis and APPL cleavage. As discussed above (point 5), we believe *GAPDH2* also plays a direct, non-metabolic extravesicular role in DCG aggregation within DCG compartments, based on our previous work on its role in ILV/exosome clustering (Dar et al., 2021). We therefore do not think similarities between *App1* and *GAPDH2* knockdown phenotypes are linked to metabolism.

11) The APPLd mutant Figure S3D shows a phenotype different from that of APPL silencing, and precisely, the "mini-core" DCG phenotype is not observed. How can authors explain this discrepancy? DCG is observed in APPLd drosophila as observed in Figure S3A. Rab6 and Rab11 DCG are not shown with the APPLd. For a complete overview of two complementary models of APP loss of function, Rab6 and Rab11 vesicles should be shown APPLd drosophila.

This is a good point and one that we should have tried to address more clearly in the manuscript. First, we should emphasise the similarities between the two phenotypes, which directly relate to our model. In both mutant and knockdown, protein aggregates more frequently remain associated with the limiting membrane of the DCG compartments and a significantly larger proportion of these compartments are targeted to lysosomes.

There seem to be at least three possible explanations for the differences that we see: i. in the *App1^d* mutant, the DCG compartments have not matured normally to Rab11 identity; ii. the *App1* knockdown is incomplete and the knockdown phenotypes we observe represent a partial loss of function; iii. the homozygous *App1^d* null allele, which was generated over thirty years ago, has acquired suppressors or compensatory mechanisms, which modify the loss-of-function phenotype. We attempted a simple outcross to try to remove from the mutant line any suppressors that are distant from the *App1* gene, but this did not drastically affect the phenotype. Addressing this specific explanation further would require an extended outcrossing experiment. We do not believe this is necessary given the underlying similarities in the two phenotypes, and the relationship of these phenotypes to our model for APP function. Furthermore, we are confident that the *App1* knockdown phenotype is reflecting the effect of reducing *App1* levels because we have demonstrated the mini-core phenotype with two independent RNAis and rescued it with a human APP construct.

We have, however, undertaken the experiments helpfully suggested by this referee with CFP-Rab6 and YFP-Rab11 (though not in combination; see Ref #4, point 1) in an *App1^d* mutant background to provide a more complete overview of the two complementary models (lines 276-277 and new Appendix Figure S2). This work shows that mature, but abnormal, DCG compartments in the *App1^d* mutant have Rab11 identity and some of them are also Rab6-labelled, as expected. The issues raised by this referee here are now further discussed in lines 628-635.

12) Previous studies have shown that APP, carboxy-terminal fragments, including AICD (amyloid

intracellular domain) and Abeta peptides, are found in exosomes. According to the present study exosomes arising from DCG contain principally the N-terminal domain of APP and potentially Abeta peptides.

We think that we have not explained our findings sufficiently well and there has been a misunderstanding here, which may be linked to points 2 and 7 above. The DCG is a protein aggregate, at least partly surrounded by intra-luminal vesicles, as we have shown in Fan et al., 2020 and subsequent publications, eg. Marie et al., 2023; Wells et al., 2023. It contains the N-terminal extracellular domain of APPL, demonstrated in Figure 4B where it is marked by GFP. It is the C-terminal, membrane-associated Intracellular domain that is associated with ILVs (and therefore Rab11-exosomes). This is difficult to see in Figure 4B, because RFP-labelled proteins in secondary cells are preferentially targeted to the lysosome (Fan et al., 2020) and are at low levels inside Rab11-compartments, though we do show some weak RFP labelling inside the DCG compartment at the surface of the DCG in Zoom #3. The APPL construct with GFP at the ICD C-terminus (Figure 4C), which is less extensively trafficked to the lysosome, clearly associates with membranes around the DCG (Zoom #2; yellow arrowheads), presumably equivalent to the ILVs that lie at the boundary of the DCG (Figure 1A). Our findings are therefore consistent with the literature and with the ideas highlighted by this referee.

We think that the text in the current manuscript does describe these ideas accurately, including the model in Figure 7, but we have stated this more explicitly in the description of Figure 4C (lines 340-342 and 358-360) to clarify the important point that the ICD is associated with ILVs/exosomes.

13) Page 12 and Page 45 are missing methodological details.

We have elaborated on the methods in the Materials and Methods (lines 825-829) and to a more limited extent, in the text of the results section (lines 321-324). It is possible that these inclusions of more methodological details will complement and help in clarifying point 12 above.

14) The authors showed that APPL silencing induces a "mini-core" phenotype. The first rescue experiments in Figure 3S should be shown.

The images for this rescue experiment are shown in Figure 3B (fourth row of images), which clearly show a suppression of the mini-core phenotype.

15) In Figure 4, APPL fluorescent constructs are under a GAL4-UAS expression system, which supposes overexpression of APPL fluorescent constructs together with the endogenous expression of APPL. Control experiments should include the overexpression of APPL UAS-GAL4 in Rab11, Rab6, and MFAS drosophila to demonstrate that overexpression of non-fluorescent APP overexpression does not impair or modify the DCG biogenesis and maturation.

In discussing Figure 4, we had highlighted that overexpressing double-tagged-APPL does disrupt overall compartmental organisation (line 324-326), but that we could still identify the different stages of DCG compartment maturation morphologically, ie. non-acidic compartments without DCGs and non-acidic compartments with DCG structures (by DIC [and APPL-ECD presence in Figure 4B]). Other overexpressed APPL and APP constructs, eg., APPL-WT, APP-YFP, also affect late stages of DCG compartment maturation, primarily leading to an increased level of targeting to the lysosome

(Figures 3 and 5; also mentioned in the original manuscript, line 293 onwards and 389-393), though less severely than double-tagged APPL (dt-APPL), perhaps because the RFP tag on dt-APPL enhances the trafficking of the RFP-labelled APPL-ICD to lysosomes (see point 12 above). Large DCGs are still formed with APPL-WT (Figure 5A; we think this is one of the experiments suggested by the reviewer in this point), so the Rab6 to Rab11 transition required for this process (Wells et al., 2023) is almost certainly taking place (see Ref #4, point 1 above and see next paragraph).

We undertook the Rab6 and Rab11 labelling experiments for *mfas* knockdown (Figures 1 and EV1) because DCGs are not being formed in mature compartments, and in the *App^l* knockdown backgrounds to confirm that the mini-core phenotype induced by reducing APPL is not merely explained by a failure of DCG compartments to mature. As discussed in Ref #4, point 11, we have now checked that all DCG compartments have Rab11 identity in the *App^l* mutant, and that some of these compartments are also Rab6-positive. In line with the referee's suggestion, we have undertaken the same analysis with SCs overexpressing APPL-WT (and also mutant APPL constructs, and dt-APPL [new Appendix Figure S3]). We find that DCG compartments have the expected identities (new Appendix Figure S4, lines 394-397 and 425-429). Our data therefore indicate that overexpression of APPL does not induce any obvious disruption of the Rab6 to Rab11 transition, and so this does not appear to be the explanation for the higher level of lysosomal targeting of DCG compartments in this genetic background (see also point 16 below). Indeed, additional experiments testing the effects of 12 days of APPL-WT expression over 12 days of adulthood (new Appendix Figure S5) suggest that these cells continue to make DCG compartments for many days with no effect on their general morphology.

16) Otherwise, experiments with APP fluorescent constructs should be performed in the APPL silenced paradigm first to show that the "mini-core" phenotype is rescued and determine the N-terminal and C-terminal contribution of APP to DCG biogenesis and maturation.

We have shown that APP-YFP rescues the *App^l* knockdown mini-core phenotype (Figure 3), demonstrating that this protein can undertake APPL's protein aggregation functions in the absence of normal APPL (but note that the levels of APP-YFP protein expression are ultimately too low to detect in SCs (new Appendix Figure S3A)). We do not think it is necessary to also show rescue with a dt-human APP protein. We have not used dt-APP to visualise APP processing in this study, because it is thought to be less efficiently cleaved at the β -secretase site in flies (ref. 51) and would therefore add an additional layer of complexity to our analysis. It seems that the referee is suggesting additional dt-APP experiments as an alternative to the experiments in point 15, which we have now performed. We wonder whether the referee suggested the dt-APP experiments should be undertaken, because of the issues they raised in point 12, concerning potential conflicts with the human APP literature. We hope we have clarified above that our work using APPL is consistent with the literature.

We should also add that we have confirmed that the mature DCG compartments in SCs overexpressing dt-APPL are Rab11-positive (new Appendix Figure S3B), in response to Ref#2, points 5/6, which at least partly addresses Ref#4's query concerning the effects of dt-APPL on DCG compartment biogenesis. We have also demonstrated using this double-tagged molecule that the APPL-ECD is cleaved off and associates with DCGs, and that some APPL-ICD molecules remain associated with Rab11-compartment limiting membranes and ILVs after cleavage, while much of it traffics to lysosomal compartments (Figure 4). We believe these experiments allow us to provide an explanation for the phenotypes observed when non-cleavable APPL constructs are expressed. We do not think that visualising the dt-APP molecule in an *App^l* knockdown background will provide

additional insights into the trafficking of the ECD and ICD during DCG compartment maturation, beyond what we have found with dt-APPL.

17) Moreover, deciding whether APP is addressed to DCG via Rab6 vesicle or Rab11 appears essential. Authors should, therefore, consider showing the APP double fluorescent construct and the APP construct with either a fused fluorescent protein at the N- or C-terminus of the APP. Moreover, acidification could be related to the overexpression of APPL constructs.

We do not fully understand the point that the referee is making here and the experiment they are suggesting: we have shown the localisation of dt-APPL and C-terminal-tagged APPL-GFP in Figures 4B and 4C (see point 12 above). We would need to label these molecules differently, eg. CFP/YFP/RFP, if we were going to study a single-tagged and double-tagged APPL/APP simultaneously and do not think this would provide further information. With the exception of the YFP-Rab11 experiment in Appendix Figures S3B, the same would apply if we wanted to determine the Rab identity of compartments containing the dt-APPL protein.

We are not sure whether the referee would like us to determine whether APPL is trafficked into maturing DCG compartments through the large Golgi-derived Rab6-compartments or smaller Rab11-compartments that interact with these Rab6-compartments (Wells et al., 2023). It is challenging to provide an unequivocal answer to this question. In normal cells, the immature Rab6-positive DCG compartments are non-acidic, do not contain a DCG and are often localised near the centre of secondary cells; they go on to form a DCG with Rab11-compartment input and often remain relatively central within the cell during the early phase of that process (Wells et al., 2023).

When dt-APPL is expressed, we can identify relatively central compartments that lack a large DCG, but in almost all cases, they contain some membrane-associated internal, ECD-labelled aggregated material, visible by DIC, as well as putative membrane-associated uncleaved APPL, eg. Box 1 in Figures 4B and 4C, as we reported in the original manuscript. Since aggregates are beginning to form, we would interpret this in normal cells as an early phase in the Rab6 to Rab11 transition. However, we cannot completely exclude the possibility that overexpressed dt-APPL is inducing premature protein aggregation prior to the Rab6 to Rab11 transition being initiated. Rab6 compartments are made from smaller Rab1/Rab6-positive compartments emerging from the *trans*-Golgi (Wells et al., 2023). We have not been able to identify such compartments containing dt-APPL, which again suggests that the APPL may primarily be arriving via the Rab11-recycling endosomal pathway. We have briefly discussed this in the revised manuscript (lines 333-336, 363-372), making it clear that we cannot exclude some input from the Golgi-derived Rab6-compartments; we also do not think that the experiment suggested by the referee would shed light on this.

We do agree that overexpression of wild type APPL may be inducing excessive acidification of more mature DCG compartments in response to high levels of this protein, although the levels of peripheral APPL ICD on compartments appear relatively low (or even absent) compared to less mature compartments. We have overexpressed other proteins in the secretory pathway in this and other studies and had not detected a similar phenotype, so we think this is not a general problem with overexpression (now explained on lines 296-298). For APPL expression, many stages of DCG compartment maturation do seem to occur relatively normally because the mature DCG compartments are Rab11-positive (eg. see new Appendix Figures S3B and S4) and a large central DCG is formed in most cases. We have used overexpression of non-mutated APP and APPL proteins either to visualise APPL trafficking (where we need to overexpress to visualise the protein in DCG compartments) or for rescue (where we would need to use two secondary cell-specific expression systems to employ reduced APP expression levels while maintaining knockdown) or to compare with

mutant APPL constructs (where we see clear additional mutant-specific phenotypes). We have discussed these issues further in the revised manuscript (lines 394-397, 425-447), but we do not think they affect our major conclusions using these constructs, which mostly relate to their trafficking, cleavage and roles in protein aggregation during early stages of DCG compartment biogenesis.

18) According to the literature, Kuzbanian/ADAM10 drosophila homologous localizes to the cell plasma membrane principally. BACE1 has been described as localizing to axonal Rab11 endosomes (Burggia-Prevot et al., 2014). In the present study, cleavage of APPL (Page 14) appears essential for DCG since lack of APPL N-terminal cleavage leads to an acidification of DCG as acidified vesicles are neither Rab11 nor Rab6. Drosophila expressing the APPL defective for either alpha or beta-cleavage should be shown. These experiments are essential to support the conclusion that overexpression of these APPL cleavage defective constructs leads to a preferential routing of DCG to lysosome compartments and that Rab6 and Rab11 fusion are early DCG biogenesis steps that are not impaired.

We have expressed a form of APPL that lacks the α - and β -cleavage sites, which are closely apposed (APPL- Δ sd). This construct (ref. 53) has been used previously by several groups to analyse the function of non-cleavable APPL. Particularly in light of the results from our secretase knockdown experiments (Ref#4, point 20; Figure EV3), we do not think that we need to independently test APPL proteins lacking either the α -site or the β -site, especially as it is known that removing one cleavage site can affect cleavage at the other (see also point 20). Regarding the referee's helpful point about BACE1 and Rab11 endosomes, it is interesting to note that of the different secretase knockdowns, β -secretase knockdown produces the strongest DCG compartment phenotype in SCs with a defect in DCG formation (GFP-negative centre; see Ref #2, point 4); we have now included a reference (Das et al., 2016) to the work on BACE1, APP and Rab11 endosomes in our discussion (lines 615-618).

19) The overexpression of the APPL constructs remains a question.

See point 17 above.

20) The role of APP cleavage and DCG biogenesis and maturation should be addressed by secretase silencing, such as Kuzbanian and dbase.

These experiments were presented, and they have now moved to Figure EV3. We believe this also partly addresses the referee's point 18.

21) Previous studies have shown that exosomes contain APP, carboxy-terminal fragments, including AICD (amyloid intracellular domain) and Abeta peptides. According to the present study, exosomes arising from DCG contain principally the N-terminal domain of APP and potentially Abeta peptides. The APP C-terminal fragments should not be observed since, in Figure 4, mature RFP is not visualized in mature DCG but localized to acidic compartments. This would, therefore, make a difference between multivesicular body-derived exosomes and those derived from DCG.

We think this point relates to the issues discussed in point 12 above. We find ILVs/exosomes contain the intracellular (C-terminal) domain of APPL or may carry full-length APPL with the extracellular domain presumably on the extravesicular side where it can be cleaved off. We do not present evidence that the extracellular domain in isolation is carried by ILVs. Our findings are therefore entirely consistent with reports in other systems and make sense topologically. The schematic model, Fig. 7, illustrates this topology. The N-terminal domain of APPL is found in DCGs, but this structure is not vesicular and there is no evidence from our studies that there are membranes inside

the DCG, only interactions with ILVs at the periphery of DCGs (and mini-cores), as explained in Figure 1A and previously shown in Fan et al., 2020 with the Breathless-GFP and CD63-GFP markers. We have tried to explain this point more explicitly on lines 358-360 and 370-372. There may be some confusion, because when mini-cores are formed, ILVs associate around their periphery and can only be distinguished from the protein aggregates by fluorescence and DIC microscopy when the mini-cores are particularly large (eg. see *GAPDH2* knockdown SC in Fig. 3 of Dar et al., 2021; this arrangement of ILVs and mini-cores is now briefly explained early in the results on lines 195-198).

22) Moreover, the localization of GFP in "mini-core" and "dense-core" granules suppose that core structures are not delineated by a lipid membrane and are opposed to exosomes produced from multivesicular bodies. In multivesicular bodies, exosomes are produced following membrane invagination by the ESCRT pathway and leading to an inverted orientation of APP. The C-terminus is inside the vesicle, while the N-terminus of APP is inside. In Figure 4, the mini-core and dense-core contain the N-terminus of APP but not the C-terminus (no RFP labeling).

Again, this relates to point 12. DCGs and mini-cores are not vesicles, they contain extravesicular proteins and potentially extracellular domains of membrane proteins, but they will only interact with vesicles at their periphery. ILVs are on the surface of the DCG (Fan et al., 2020; Marie et al., 2023) and mini-cores (Dar et al., 2021), and this is where the GFP-labelled APPL-ICD is found (Figure 4C). This is illustrated in the model in Fig. 7.

23) Minor concerns that should be addressed

Page 7, line 144 from 152: DCG biogenesis and maturation observed by video microscopy should be placed after line 127, page 6.

Thank you for this helpful suggestion, we have reordered the text.

24) If the referee understood well, mini-core intraluminal vesicles fuse to form the dense core intraluminal matured vesicles. They are then supposedly an intermediate step, occurring after Rab6 / Rab 11 vesicle fusion and DCG matured vesicles. Could the authors clearly state, based on their observations, the kinetic of DCG biogenesis and maturation, including the mini-core vesicles?

This indicates that we needed to provide a clearer description of the system in the Introduction, which we have now done, so that it is emphasised that vesicles are not components of the DCG, but interact with the DCG. This aspect is also covered in point 12 above and Ref #3, point 6.

25) Figure 4B: Representation of APPL, APPLdeltaSD, and APPLdeltasdE1 constructs should be moved to Figure 5. nAPPLc-GFP construct should be represented. Is the nAPPLc-GFP in Figure 4D like the APPL-GFP in Figure 4A?

Thank you, that is helpful. We have moved this panel to Figure 5B and made the labelling of constructs in Figure 4 consistent.

26) Could the author homogenize the nomenclature used? Use TGFBI or MFAS everywhere, please.

We have done this and use MFAS in most cases, except where we specifically refer to the human molecule.

27) The present work interests the scientific committee of cell biology and Alzheimer's disease. Only a

limited number of scientists are experts in cell vesicle trafficking and DCG biogenesis, maturation, and secretion. Could authors make a slight effort to make their manuscript more accessible to a general audience?

Please see point 24 above and also one of the opening paragraphs of our overall response, where we discuss the broad relevance. We have tried to more clearly explain these aspects in a revised Introduction and Discussion.

Dear Clive,

Thank you again for submitting your revised manuscript (EMBOJ-2024-118705R1) to The EMBO Journal for our consideration, and for your patience during peer review. Your manuscript has now been seen by the four original referees who had previously assessed the first version of your manuscript, and we have received their comments, which you can find below. I am very glad to say that the referees find the revised manuscript strengthened and improved, and they mention that the initially raised concerns have been adequately and sufficiently addressed. In light of this input and the recommendations of the referees, I am very pleased to inform you that your manuscript has been in principle accepted for publication in The EMBO Journal. Congratulations on an excellent study.

Before we can proceed with formal acceptance and publication of the manuscript, there is a minor remaining comment (of referee #2) that we kindly request you to address in a final version of your manuscript: the referee suggests that the used term "exosomes" be changed to "extracellular vesicles (EVs)" given that their status as exosomes has not been formally justified by full characterization.

From the editorial side, there are also a few changes and corrections we need you to make in the final version of your manuscript:

- The references format is not compatible with our journal's style (please note that the names of the first 10 co-authors should be listed for each reference, followed by "et al." in cases of papers with more than 10 co-authors; references should be listed in alphabetical order) - please find more information on the required references format in our guide below and update your references accordingly: <https://www.embopress.org/page/journal/14602075/authorguide#referencesformat>.

- Thank you for providing reviewer access to your deposited datasets. The confidential reviewer tokens can now be removed from your "Data availability" statement. Please make sure that all datasets will be publicly available at the time of publication, and provide/update as necessary the databases, dataset identifies, and the specific links (URLs) to all deposited datasets in the "Data availability" statement of your revised manuscript.

- Please change the heading of your "Declaration of interests" statement to "Disclosure and competing interests statement".

- Please note that Figure callouts should be listed sequentially; we also noticed that callouts for Fig. 5D are missing. Please make sure that all Figure panels are called out sequentially in your revised manuscript.

- Please complete your Author Checklist by selecting the appropriate response to each question from the drop-down menu.

- The title page of the Appendix PDF file should begin with "Appendix for" followed by the manuscript title and a Table of Contents with the page numbers of the listed items.

- Please note that EMBO press papers are accompanied online by:

- A) a short (2 sentences) summary of the findings and their significance,

- B) 2-5 short bullet points highlighting the key results, and

- C) a synopsis image in .jpg or .png format that is exactly 550 pixels wide and 300-600 pixels high (the height is variable). Please note that the text needs to be legible at the final size.

Please upload this information along with your revised manuscript (the text for A and B should be provided in a separate Word file).

- During our standard Figure checks, we detected similarity between cells shown in Figure 4C (Lys). We kindly request you to check these images carefully and revise them if necessary, or clarify.

- During our routine pre-acceptance checks, our data editors have raised the following queries regarding Figures, data, and legends. Please make sure that the following requests are fully addressed in the final version of your manuscript: "Please provide the exact p values in the legends of Figures 1E, 2B-D; 3D-F; 5C-G; 6C-E; EV1 I; EV2 E-H; EV3 C-E; EV4 B; EV5 B."

- Movies (with their corresponding callouts throughout the manuscript) should be renamed to "Movie EV1-EV6"; their legends should be removed from the main manuscript file and zipped instead individually with each corresponding movie file.

- The order of the manuscript sections must be corrected as follows: Title page - Abstract and Keywords - Introduction - Results - Discussion - Methods - Data Availability - Acknowledgements - Disclosure and Competing Interests Statement - References - Figure Legends - main Tables (if there are any) - Expanded View Figure Legends.

Please also note that as part of the EMBO publications' Transparent Editorial Process, The EMBO Journal publishes online a

Peer Review File along with each accepted manuscript. This File will be published in conjunction with your paper and will include the referee reports, your point-by-point response and all pertinent correspondence relating to the manuscript. You can opt out of this by letting the editorial office know (contact@embojournal.org). If you do opt out, the Peer Review File link will point to the following statement: "No Peer Review File is available with this article, as the authors have chosen not to make the review process public in this case."

We look forward to seeing a final version of your manuscript as soon as possible. Please let us know if you have any questions and use this link to submit your revision: <https://emboj.msubmit.net/cgi-bin/main.plex>.

Best wishes,

Ioannis

Referee #1:

The authors have addressed all my concerns to a satisfactory manner and I recommend the manuscript be accepted.

Referee #2:

The authors have adequately addressed the comments made by this referee in the revised version of the manuscript that has been substantially improved. Some of the points were not addressed experimentally but the authors sufficiently explained the difficulties they faced.

I have one comment that needs to be addressed.

Since the authors haven't characterized 'exosomes' (exosome isolation, biochemical characterization, proteomics, EM, NTA etc according to the MISEV guidelines) and their origin is not purely endosomal/MVB, I strongly suggest changing the term 'exosome' to 'Extracellular Vesicles-EVs'.

Referee #3:

The authors have satisfactorily addressed this reviewer's major and minor concerns.

Referee #4:

The authors have addressed all major and minor concerns, performed the essential additional experiments requested, and substantially edited the manuscript, including modified and new figures. The review is grateful to the authors for their answers and comments and has no additional major or minor concerns with the revised manuscript. This completely novel APP-related function toward dense granule genesis is herein well-demonstrated. I would recommend to accept this manuscript for publication in Embo Journal.

Below, we include a rebuttal to the request made by Referee #2:

'there is a minor remaining comment (of referee #2) that we kindly request you to address in a final version of your manuscript: the referee suggests that the used term "exosomes" be changed to "extracellular vesicles (EVs)" given that their status as exosomes has not been formally justified by full characterization.'

We strongly disagree with the amendment to the terminology (from 'Rab11-exosome'/'exosome' to 'EV') suggested by Referee #2, because this would be at odds with the current literature and we think would lead to confusion in the field. Indeed, justification for the use of the term 'exosome' was included in our initial response (parts of our response to Referee #2, points 3 and 4). We believe that using the correct and accepted nomenclature for these and any other types of EV is a really important point in the EV field, and we have detailed our arguments for retaining the use of the word 'exosome' below.

Rebuttal of 'exosome' to 'extracellular vesicle (EV)' change

We make our arguments on the basis of the following seven points:

1. Referee #2 refers to the MISEV guidelines to argue their case, but this position paper specifically refers to our exosome studies – The latest version of the MISEV guidelines (MISEV2023) supports our position. Indeed, it highlights '*exosome subtype biogenesis*' (Table 4) as a specific strength of our fly secondary cell (SC) model. Point 6 below, provides further support from MISEV2023 for our use of the term 'exosome'.

2. Our previous publications with the SC model, used 'Rab11-exosome' or 'exosome' throughout as the description of the secreted versions of the intraluminal vesicles (ILVs) made in SC Rab11-positive secretory compartments, including the original detailed characterisation of this specific exosome subtype in *EMBO J*, eg [all cited in our manuscript, EMBOJ-2024-118705R1].:

Corrigan L, Redhai S, Leiblich A, Fan SJ, Perera SM, Patel R, Gandy C, Wainwright SM, Morris JF, Hamdy F, Goberdhan DC, Wilson C. BMP-regulated exosomes from *Drosophila* male reproductive glands reprogram female behavior. *J Cell Biol.* 2014 206:671-88.

Published prior to the characterisation of the Rab11 exosome signature.

Fan SJ, Kroeger B, Marie PP, Bridges EM, Mason JD, McCormick K, Zois CE, Sheldon H, Khalid Alham N, Johnson E, Ellis M, Stefana MI, Mendes CC, Wainwright SM, Cunningham C, Hamdy FC, Morris JF, Harris AL, Wilson C, Goberdhan DC. Glutamine deprivation alters the origin and function of cancer cell exosomes. *EMBO J.* 2020 39:e103009.

Marie PP, Fan SJ, Mason J, Wells A, Mendes CC, Wainwright SM, Scott S, Fischer R, Harris AL, Wilson C, Goberdhan DCI. Accessory ESCRT-III proteins are conserved and selective regulators of Rab11a-exosome formation. *J Extracell Vesicles.* 2023 12:e12311.

Wells A, Mendes CC, Castellanos F, Mountain P, Wright T, Wainwright SM, Stefana MI, Harris AL, Goberdhan DCI, Wilson C. A Rab6 to Rab11 transition is required for dense-core granule and exosome biogenesis in *Drosophila* secondary cells. *PLoS Genet.* 2023 19:e1010979.

3. The use of 'exosome' was accepted by the multiple reviewers of these different papers – This is because we had visualised membrane and membrane-associated markers (and also ILVs by super-resolution microscopy and EM) inside multivesicular endosomes labelled by the recycling endosomal marker Rab11. We also demonstrated that these compartments release their contents from the

apical plasma membrane of SCs leading to the marked ILVs being secreted into the lumen of the accessory gland.

4. We have also visualised ILV biogenesis in compartments coated with an established endosomal marker (Rab11) in real-time (Wells et al., 2023) and showed these ILVs are secreted (Fan et al., 2020) – To our knowledge, this is the only real-time demonstration in any living tissue that a specific vesicle that will be secreted as an exosome, actually *originates* in the endosomal compartments in which they are observed.

5. We have identified and used specific conserved regulators of Rab11-exosome biogenesis to validate the functions of these vesicles (Marie et al., 2023) – We demonstrated that accessory ESCRT-III proteins affect Rab11-exosome biogenesis, but not late endosomal exosome biogenesis. This work validates the YFP-Rab11 and Btl-GFP markers employed in our current manuscript (EMBOJ-2024-118705R1), with CFP-Rab6 described as an additional marker in Wells et al., 2023. Importantly, the Marie et al paper was published in the *Journal of Extracellular Vesicles*, the open-access scientific journal of the International Society for Extracellular Vesicles (ISEV), which also coordinates the assembly of the MISEV guidelines. This journal has specific guidelines concerning use of terms like exosome and in writing Marie et al., 2023, we adhered to those guidelines. None of the referees raised issues about the nomenclature.

6. Professor Goberdhan was part of the five-member MISEV2023 organising committee – The five committee members drafted and co-ordinated MISEV2023 (Goberdhan was also a corresponding author). This was written with a core principle that demonstrating the presence and roles for exosomes, EVs, etc in different models often requires the use of different approaches:

Welsh JA, Goberdhan DCI, O'Driscoll L, et al. Minimal information for studies of extracellular vesicles (MISEV2023): From basic to advanced approaches. *J Extracell Vesicles*. 2024 Feb;13(2):e12404.

MISEV2023 included a new section on EV analysis *in vivo*, which emphasised the advantages and limitations of *in vivo* systems, the latter including EV separation and the techniques that follow from that, which Referee #2 refers to. Notably, Fan et al., 2020 also included characterisation of human Rab11-exosomes, including the analyses highlighted by Referee #2, to support the complementary findings from SCs. Marie et al., 2023 and this new manuscript have used the human Rab11-exosome proteome to identify specific regulators and interactors that we have initially confirmed as functionally important in fly SCs (see also Dar et al., 2021), strongly suggesting that we are studying equivalent types of vesicle in humans and flies.

Key points in MISEV2023, which strongly argue that we are using the appropriate term, 'Rab11-exosomes', to refer to the secreted vesicles from SCs are:

a. page 4: *'MISEV IS NOT: A one-size-fits-all blueprint, a comprehensive checklist of 'dos and don'ts', or a substitute for careful expert judgement. There is no technique or platform that is absolutely required or prohibited by MISEV.....Chosen techniques and targets should be fit for purpose, appropriate for the experimental system, contributing to overall MISEV compliance, and properly reported. Importantly, no research group has access to all techniques and platforms.'* – this statement and other points made in this section of the position paper were deliberately included to make it clear that there are no specific techniques absolutely required to demonstrate vesicles are exosomes. The citing in this position paper of our previous work as examples of exosome studies (see 'c' below) is consistent with that, ie. Corrigan et al., 2014; Fan et al., 2020; Marie et al., 2023.

b. Table 2, page 6: *definition of exosome: 'Biogenesis-related term indicating origin from the endosomal system.'* – one might argue that our live-imaging approach with recycling endosomal Rab markers (Wells et al., 2023) has shown this more clearly than most other studies in the field.

c. Table 4, page 40: *fly secondary cell model: 'Large MVBs: exosome subtype biogenesis'* – the use of this model to study exosome biology in an *in vivo* system is exclusively highlighted in this Table.

7. Referee #2 also argues that the origin of Rab11-exosomes is not 'purely endosomal/MVB' – This requirement is, however, too stringent for even the classic site of exosome production, late endosomes, which receive inputs from the *trans*-Golgi, and sometimes from autophagic compartments. The production of Rab11-exosomes in Rab11-positive recycling endosomes, formed after fusion with Rab6-positive secretory compartments is another example of compartmental membrane mixing prior to exosome biogenesis. Indeed, we are, currently, assembling a review that deals with the idea that exosomes are formed from endosomal compartments receiving inputs from multiple different non-endosomal compartments and organelles, and the implications of this for exosome structure and function.

In summary, our view is that renaming fly SC Rab11-exosomes as EVs will cause considerable confusion in the field, since it will contradict the information in the MISEV2023 guidelines and our previous publications. There are also recent reviews by experts in the EV field, which discuss the significance of identifying exosomes in different types of endosomal compartments, which specifically cite our work, eg.

van Niel G, Théry C. Extracellular vesicles: eat glutamine and spit acidic bubbles. *EMBO J.* 2020 Aug 17;39(16):e105119.

Commentary for Fan et al., 2020 *EMBO J.* 2020 39:e103009 refers to, '*a novel subpopulation of exosomes enriched in Rab11a.....*' (last sentence of abstract).

Dixon AC, Dawson TR, Di Vizio D, Weaver AM. Context-specific regulation of extracellular vesicle biogenesis and cargo selection. *Nat Rev Mol Cell Biol.* 2023 24:454-476.

Review quoted in our manuscript (EMBOJ-2024-118705R1), states, '*these exosomes are formed within RAB11-positive recycling endosomes*' (bottom paragraph, page 458).

We therefore kindly request that the Rab11-exosome nomenclature we have used is maintained.

Dear Clive,

Congratulations on an excellent study! I am very pleased to inform you that your manuscript has been accepted for publication in The EMBO Journal. Thank you very much for comprehensively addressing the initially raised referees' concerns and the editorial and formatting requests.

If you have any questions, please do not hesitate to contact the Editorial Office. Thank you for your contribution to The EMBO Journal. Working with you has been a pleasure.

Best regards,

Ioannis
